# Boron isotopes in boninites document rapid changes in slab inputs during subduction initiation

Hong-Yan Li [1,2,3✉], Xiang Li[1,4✉], Jeffrey G. Ryan [5✉], Chao Zhang [6] & Yi-Gang Xu[1,2,3]

How subduction-related magmatism starts at convergent plate margins is still poorly understood. Here we show that boron isotope variations in early-formed boninites from the Izu-Bonin arc, combined with radiogenic isotopes and elemental ratios document rapid (~0.5 to 1 Myr) changes in the sources and makeup of slab inputs as subduction begins. Heterogeneous hornblende-granulite facies melts from ocean crust gabbros ± basalts fluxed early melting to generate low silica boninites. Hydrous fluids from slab sediments and basalts later fluxed the low silica boninites mantle source to produce high silica boninites. Our results suggest that initially the uppermost parts of the slab were accreted near the nascent trench, perhaps related to early low-angle subduction. The rapid changes in slab inputs recorded in the boninites entail a steepening subduction angle and cooling of the plate interface, allowing for subduction of slab sediment and basalt, and generating hydrous fluids at lower slab temperatures.

[1] State Key Laboratory of Isotope Geochemistry, Guangzhou Institute of Geochemistry, Chinese Academy of Sciences, Guangzhou 510640, China. [2] CAS Center for Excellence in Deep Earth Science, Guangzhou 510640, China. [3] Southern Marine Science and Engineering Guangdong Laboratory (Guangzhou), Guangzhou 511458, China. [4] University of Chinese Academy of Sciences, Beijing 100049, China. [5] School of Geosciences, University of South Florida, Tampa, FL 33620, USA. [6] State Key Laboratory of Continental Dynamics, Department of Geology, Northwest University, Xi'an 710069, China. ✉email: hongyanli@gig.ac.cn; lixiang@gig.ac.cn; ryan@usf.edu

Despite over fifty years of studies into the workings of plate tectonics, the phenomenon of subduction initiation is still not well understood[1–3]. How slabs start to subduct, whether via shallow convergence[4] or vertical foundering[5,6], to the point that they begin to interact with the mantle continues to be debated. The Izu-Bonin-Mariana (IBM; Fig. 1) convergent plate margin is a unique natural laboratory for the study of subduction initiation, with well-preserved forearc igneous sequences that represent the first eruptive products of subduction in this region. International Ocean Discovery Program (IODP) Expedition 352 recovered a representative suite of the earliest subduction-related volcanic sequences of the Izu-Bonin subduction system[7–9] (Fig. 1). Drillsites nearer the trench (U1440 and U1441) recovered forearc basalts (FAB), while two sites ~15 km inboard from the trench (U1439 and U1442) recovered boninites and their high-Mg andesite (HMA) differentiates. Radiometric dates for FAB are 51.9–51.3 Myr, while boninite dating suggest eruption very shortly thereafter, at ~51.3–50.3 Myr[7]. On the *JOIDES Resolution*, the Expedition 352 science team used a hand-held portable XRF (pXRF) instrument to track chemical variations in the FAB and boninite sections as part of the core logging process, collecting over 2000 individual measurements[10]. Recent high precision shore-based XRF analyses[11,12] supplemented by the shipboard pXRF[10] and ICP-AES[13] analyses created high resolution chemostratigraphies for the drilled holes[10–13]. In the boninite holes (U1439A, U1439C and U1442A), lavas < ~250 meters below the seafloor (mbsf) are dominated by high silica boninites (HSB) and their HMA differentiates, while lavas >250 mbsf are dominated by low silica boninites (LSB) and HMA, with uncommon appearances of HSB[12], consistent with HSB intrusions through LSB strata.

Boninites are a ubiquitous early IBM volcanic product, which requires that a chemically highly depleted, shallow mantle source be fluxed by the slab[14]. Expedition 352 boninites all have elevated $SiO_2$ and MgO, and low rare earth element (REE) contents, consistent with melting of highly depleted mantle[4,14]. The HSB have higher $SiO_2$ and lower $TiO_2$ at given MgO than the LSB[9,12], pointing to more depleted mantle[15]. The IBM boninites show fluid-mobile element (FME) enrichments broadly similar to those in arc lavas, but are also enriched in key fluid-immobile species (e.g., Zr and Hf) that suggest the involvement of slab-derived melts at relatively shallow depths[16].

Trace element and radiogenic isotope results indicate little or no slab influence on the mantle sources of the earliest erupted Izu-Bonin FAB[4,11,17]. LSB lavas specifically and boninites more generally have been interpreted as reflecting slab melt contributions from Pacific plate crust[4,16]. Slab sediment-derived isotopic signatures only become evident in the later erupted HSB[4]. However, the specific constituents of the oceanic crust and lithosphere (altered basalts or gabbros, or lithospheric serpentinite) that contribute to boninite genesis, and the respective roles of slab-derived fluids and/or melts in the genesis of the HSB

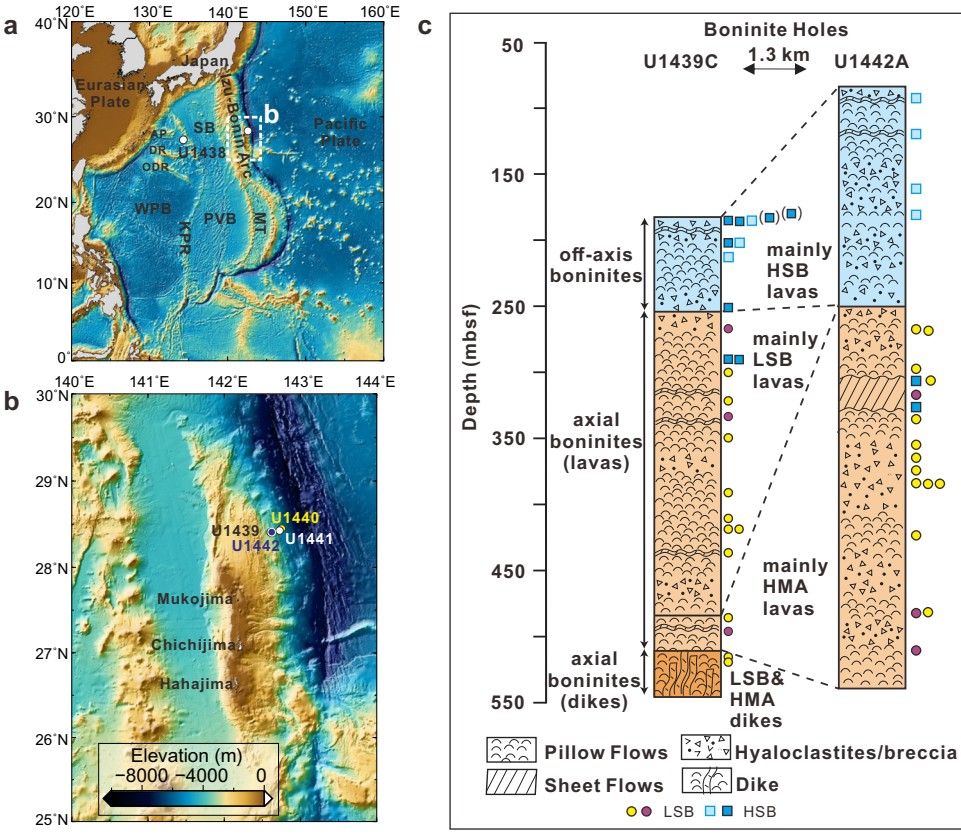

**Fig. 1 Regional map of the Izu-Bonin-Mariana convergent plate margin with sampling locations. a, b** Locations of the International Ocean Discovery Program Expedition 352 drill sites in the Izu-Bonin forearc, Western Pacific. U1439 and U1442 are boninite sites, while U1440 and U1441 are forearc basalt sites. **c** Simplified logs for the two boninite holes (U1439C and U1442A), recovering lavas and dykes mainly of high-silica (HSB) and low-silica (LSB) boninites and their differentiation products, high magnesium andesites (HMA). The depths in the cores for each sample examined this study and HSB samples from Li et al.[4] (light blue squares) are plotted alongside the core logs. The depth of two HSB samples from hole U1439A (In brackets) are also plotted alongside hole U1439C. For the purposes of this study HMA differentiates are not distinguished from their parental HSB or LSB. SB: Shikoku Basin; PVB: Parece Vela Basin; MT: Mariana Trough; KPR: Kyushu Palau Ridge; WPB: West Philippine Basin; AP: Amami Plateau; DR: Daito Ridge; ODR: Oki-Daito Ridge; mbsf: meters below the seafloor.

versus the LSB are still not well understood. These questions are directly relevant to how forearc magmatism reflects the geometry of the downgoing slab and its thermal evolution during the subduction initiation. Of particular importance in this regard is constraining the slab-to-mantle inputs during subduction initiation, and the mechanisms governing them. The extremely depleted nature of boninite mantle sources means that even minor inputs from the downgoing plate should produce strong geochemical signals. Boron (B), as an endmember trace element in terms of its extreme mobility off slabs, with clear abundance and isotopic differences among different slab constituents[18,19], has great potential for tracking these earliest slab-to-mantle material exchanges.

Here, with new high precision B abundance and B isotope results, combined with data for other trace elements and with Sr-Nd-Hf isotopes for Expedition 352 boninites, we show that rapid temporal changes occur in the slab inputs to the sources of IBM boninitic magmas, from melts of lower crustal gabbros to fluids derived from upper crustal basalts and sediments, which provides insights into the thermal evolution of the slab during the earliest stages of subduction.

## Results

**Boron elemental and isotopic compositions of the boninites.** Boron concentrations in Expedition 352 boninites vary by boninite type, with mean LSB at ≈4 µg/g B, and mean HSB at ≈12 µg/g B (Supplementary Data 1). While the B concentration ranges of HSB and LSB show some overlap, there are clear distinctions with respect to MgO and TiO$_2$, suggesting multiple magmatic sequences with distinct B abundances. Differences in the correlations of B with Ba and other trace elements suggest LSB versus HSB source differences (Fig. 2). As B, Ba, and Nb are all strongly incompatible during mantle melting and magma chamber crystallization, B/Nb and Ba/Nb can characterize to a first order the degree of B and Ba enrichment in the mantle source. δ$^{11}$B in Expedition 352 boninites ranges from −1.6‰ to +8.3‰, with clear differences among sub-suites when linked to B/Nb and Ba/Nb (Fig. 3a, b). The LSB encompass the full range of boninite δ$^{11}$B variation, and show variable B/Nb and Ba/Nb. By contrast, δ$^{11}$B in the HSB cluster tightly between −0.2‰ and +1.8‰, and have uniformly higher B/Nb and Ba/Nb. All HSB, both those recovered at <250 mbsf, and the few that crosscut LSB strata at >250 mbsf, show closely similar geochemical characteristics, pointing to similar subduction inputs. The LSB appear to break into high and low Ba (Ba/Nb) subgroups, based on correlations between Ba and Nb, Hf, Zr, and Th (Supplementary Fig. S1 and Fig. 4a). No obvious stratigraphic relationships are evident among the LSB subgroups (Fig. 1c). The high-Ba/Nb LSB subgroup has an overall higher δ$^{11}$B (+0.5‰ to +8.3‰) than do low-Ba/Nb LSB (−1.6‰ to +4.1‰). The δ$^{11}$B of Expedition 352 boninites overall are lower

than the δ$^{11}$B range reported for Izu-Bonin-Mariana arc volcanics (δ$^{11}$B: +3‰ to +12‰)[20–22]. In detail, the LSBs, in particular those with elevated Ba/Nb, show considerable δ$^{11}$B overlap with Izu-Bonin volcanic front lavas, while the low Ba/Nb LSBs and all HSBs are distinctly lower at Ba/Nb ratios 2-20 times lower than are seen in the arc. While the LSB range to higher δ$^{11}$B, they have lower and more uniform $^{87}$Sr/$^{86}$Sr (0.7032–0.7038) than the HSB (0.7035–0.7048; Fig. 3c).

**Boron systematics in subduction versus subduction initiation settings.** The model for B and B isotope systematics in arc magmatism is distinct among those of lithophile trace elements in that B is uniquely mobile in hydrous slab-derived fluids at low temperatures. Thus, slab-related reservoirs produced at low temperatures, specifically serpentinites derived ultimately from reactions with seawater (at 4.5 ppm B and δ$^{11}$B ≈ +39.5‰) play an outsized role in B cycling during subduction, serving as both the dominant B reservoir, and the "best fit" source for the isotopically heavy δ$^{11}$B signatures seen in many arcs[18,19,23]. The involvement of serpentinites in arc magmatism is possible because in mature subduction systems downgoing plates often have sufficiently cool thermal structures to permit the deep subduction of serpentinite along the plate interface and within the uppermost portions of slab crust and lithosphere, leading ultimately to B enriched, high δ$^{11}$B eruptive products in arcs. By contrast, in subduction systems where downgoing plates are hotter and, usually, younger (e.g., the Cascades, Mexico, Italy), lavas are much less B-enriched, and often preserve low δ$^{11}$B signatures suggesting little to no involvement of serpentinite in the slab-derived component[18].

Subduction initiation as recorded in the IBM system[24] reflects uniquely high temperature, low pressure conditions[25], as initial slab foundering led to extension, asthenospheric upwelling and melting, resulting first in FAB magmatism and then boninite magmatism, all occurring in close proximity to the slab edge, such that shallow, hot mantle depleted by FAB melting was re-melted due to fluxing by slab-derived inputs[4,12,13,16]. Hot mantle conditions near the sinking edge of the downgoing plate are likely responsible for the generation of melts in the slab crust, suggested to occur under hornblende-bearing granulite facies conditions (900–950 °C)[16,26]. Serpentine minerals break down at <700 °C under low-pressure conditions, and serpentinite decomposition will release water, which can cause ocean crust to melt at low temperatures (<750 °C)[26]. Therefore, slab-hosted serpentinites (either crustal or lithospheric) must have been scarce for the slab to reach higher temperatures before melting to fertilize the boninite mantle source. It is unlikely that serpentinites could develop in a nascent subduction interface, given the high temperature conditions. If they ever existed, they would be likely to break down well before slab melts could be generated.

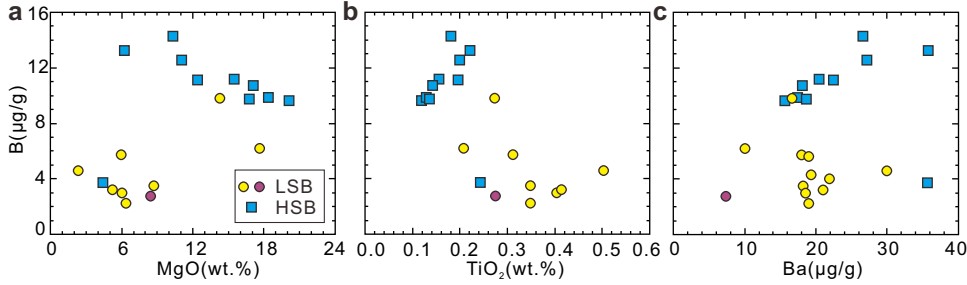

**Fig. 2 Boron elemental variations of the Expedition 352 boninites.** Plots of B vs. **a** MgO, **b** TiO$_2$ and **c** Ba for the Expedition 352 boninites. LSB: low-silica boninite; HSB: high-silica boninite.

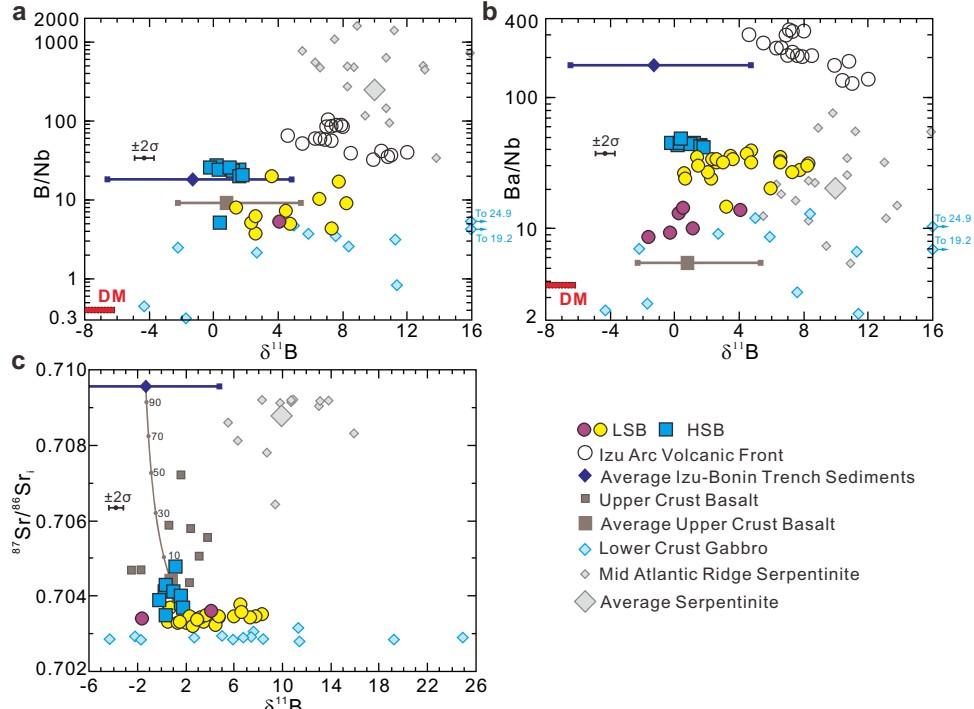

**Fig. 3 Boron and strontium isotopes and trace element systematics of the Expedition 352 boninites.** Plots of **a** B/Nb, **b** Ba/Nb and **c** $^{87}Sr/^{86}Sr_i$ vs. $\delta^{11}B$ for the International Ocean Discovery Program Expedition 352 boninites. Error bars ($2\sigma = 0.6‰$) for boninite $\delta^{11}B$ represent average whole chemical process external reproducibility based on duplicate analyses of different digestion of the reference materials and natural samples[62]. Depleted mid-ocean ridge basalt mantle (DM[32,49,64]), Mid-Atlantic Ridge serpentinites[65], Izu-Bonin trench sediment[28,38,39], upper crust basalts (Deep Sea Drilling Project Holes 417A, 417D and 418A)[28,66], lower crust gabbros (Ocean Drilling Program Hole 735B)[28,54], and Izu arc (Ocean Drilling Program Hole 782A tephra)[21] data are shown for comparison. Bulk mixing curve between Izu-Bonin trench sediment and upper crust basalt in **c** is calculated per their chemical compositions, summarized in Supplementary Table 1. Numbers on the mixing curves denote percentages. Two low-silica boninite (LSB) subgroups are distinguished by their distinct Ba versus Nb, Zr, Hf, and Th correlations (Supplementary Fig. 1). HSB: high-silica boninite.

**Slab sources for boninite boron in subduction initiation settings.** Although $\delta^{11}B$ data for ocean crust are limited, there nonetheless appear to be distinctions in $\delta^{11}B$ with crustal layer[27–29]: oceanic Layer 2a basalts and diabases vary from −2.5‰ to +5.4‰, while gabbroic Layer 3 rocks scatter from −4‰ to +25‰, with consistently lower B/Nb and overall higher $\delta^{11}B$ than Layer 2a altered basalts[28] (Fig. 3). Similar patterns have also been observed in ophiolitic crustal sections[30,31]. Layer 2a and Layer 3 are also distinguishable in terms of Sr isotopes, with Layer 3 rocks preserving consistently lower $^{87}Sr/^{86}Sr$ than Layer 2a rocks. The combined B and Sr isotopic systematics of oceanic Layer 3 rocks are consistent with overall lower extents of seawater alteration than seen in Layer 2a, leading to lower $^{87}Sr/^{86}Sr$ and lower B contents; and with exchanges occurring at overall higher temperatures than in Layer 2a, consistent with greater depth in the crust, resulting in overall higher $\delta^{11}B$. The $\delta^{11}B$ variability in Layer 3 data points to considerable local variation in the extent of seawater/rock exchange deeper in the ocean crust, and to the extreme sensitivity of B isotopes to high B, high $\delta^{11}B$ inputs (seawater at 4.5 ppm B and $\delta^{11}B \approx +39.5‰$, as compared to fresh Layer 3 rocks at <<1 ppm B and $\delta^{11}B \approx -7‰$[19,32]).

Ocean crust metamorphic processing during subduction modifies its mobile element abundances and $\delta^{11}B$. In mature subduction zones, the $\delta^{11}B$ of the altered oceanic crust near the slab-mantle interface decreases due to early losses of high $\delta^{11}B$ boron to the forearc mantle at shallow depth, as indicated by $\delta^{11}B$ of −6 ± 4‰ of mafic blueschist clasts metamorphosed at ~19 km depths and 200–350 °C, recovered from serpentinite muds in the Mariana forearc[33]. However, during subduction initiation, the downgoing tip of the slab reaches much higher temperatures,

upwards of ~900 °C at 1 GPa based on recent models[25]. B, Rb, K, and Ba concentrations are high in amphibolites in the metamorphic sole of the Oman ophiolite, suggesting exchanges with FME-rich fluids during prograde slab metamorphism[34]. The Oman amphibolites have elevated $\delta^{11}B$, between −2.3‰ to +10.8‰, averaging +3.75‰[35]. Exchanges with deeply derived amphibolite-facies fluids appear to explain high $\delta^{11}B$ signatures and elevated B/Nb, Ba/Nb and Sr/Nd in the Oman amphibolites, and this high temperature phenomenon may be characteristic of subduction initiation generally[34]. It is also possible that the breakdown of lithospheric serpentinite underlying downgoing plate crust could enrich crustal rocks with high $\delta^{11}B$ boron in similar ways, though serpentinites underlying the oceanic crust may be comparatively less enriched in B than crustal amphibolites, given the strong uptake of seawater-derived B during alteration of the oceanic crust[27,36]. Both amphibolite and serpentinite fluids will heat up when rising, increasing extraction of FMEs from the crust[34,37]. So generally, Layer 2a and Layer 3 ocean crustal materials can both provide heavy B during subduction initiation.

Modern marine sediments are enriched of B, with Izu-Bonin trench sediments averaging 94.2 µg/g B[38], and the majority of analyzed samples ranging between 70 and 130 µg/g[39]. Sediment $\delta^{11}B$ ranges from −6.6 to +4.8‰ with significant differences among constituents, e.g., continental detritus are very low, at −13 to −8‰ while biogenic carbonates vary from +8.0 to +26.2‰[39]. The majority (>70%) of sediment-hosted B will likely be removed from sediments early in subduction by a range of fluid release phenomena, driven by mechanical compaction, diagenesis and prograde metamorphism, all at pressures <1 GPa and

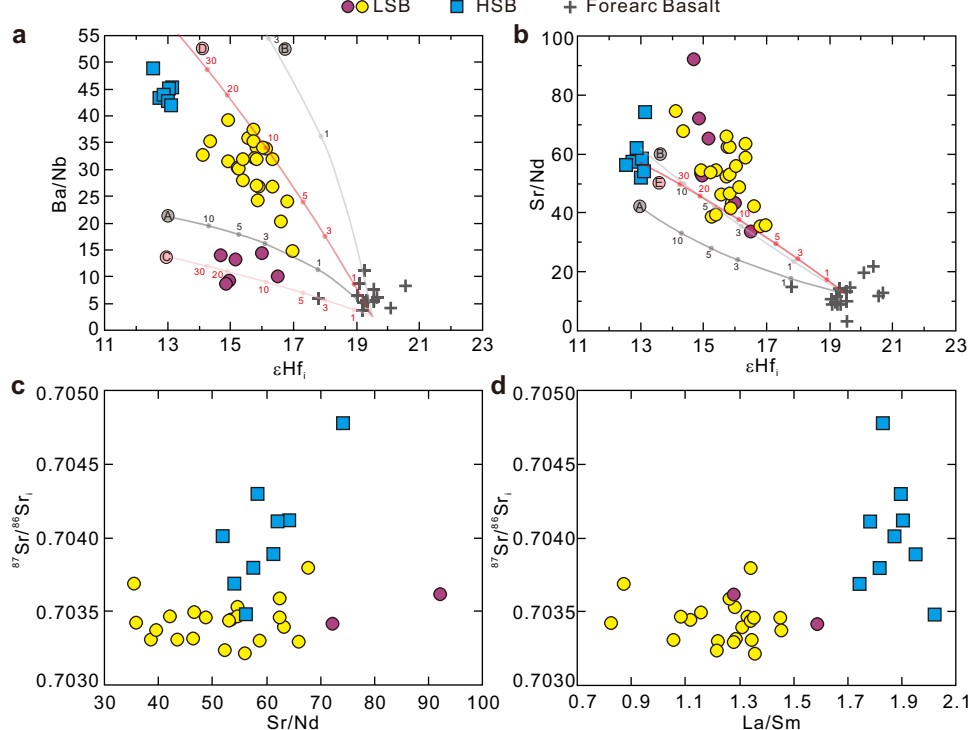

**Fig. 4 Radiogenic isotopes and trace element systematics of the Expedition 352 boninites.** Plots of **a** Ba/Nb and **b** Sr/Nd versus εHf$_i$ and $^{87}Sr/^{86}Sr_i$ versus **c** Sr/Nd and **d** La/Sm for International Ocean Discovery Program Expedition 352 forearc basalt and boninites. The mixing curves in **a** and **b** are between the depleted mid-ocean ridge basalt mantle[49,64] and the calculated granulite melts (F = ~10%). **A**: Basaltic protolith low Ba/Nb granulite melts; **B**: Basaltic protolith high Ba/Nb granulite melts; **C**: Gabbroic protolith low Ba/Nb granulite melts; **D**: Gabbroic protolith high Ba/Nb granulite melts; **E**: Melts of gabbroic protolith granulite with its original Sr/Nd. The numbers on the mixing curves represent the mass percentage of the slab melt. The forearc basalt data are from Li et al.[4]. LSB: low-silica boninite; HSB: high-silica boninite.

temperature <350 °C in mature subduction zones[40–44]. Subducted sediments at ~0.5 GPa and ~300 °C should thus have lower B abundances (3.4 to 57.4 µg/g) and lower δ[11]B (−9‰ to −11.9‰)[45] than seafloor sediments. Similarly low B contents (3.6 to 24.1 µg/g) are observed in subducted sediments from the Oman metamorphic sole[34]. Therefore, subducted sediments should generate isotopically light B inputs during subduction initiation.

Expedition 352 volcanic rocks show Hf, Nd and Pb isotopic evidence for a shift from predominantly Indian Ocean-related mantle sources in early-erupted FAB to sources with a Pacific Ocean provenance in the boninites, consistent with slab-derived inputs from Pacific plate crust and lithosphere[4]. Our data patterns in Fig. 3 are consistent with these arguments, but provide more detail as to the likely source contributors. A key feature of our boninite data is the absence of any positive correlation between B enrichment and δ[11]B: the HSB, which have the highest B/Nb ratios, are among the lowest δ[11]B lavas, while the LSB show variable B/Nb ratios and δ[11]B signatures, ranging from δ[11]B lower than any HSB to δ[11]B > +8‰. These systematics argue against a controlling role for a high B, high δ[11]B slab constituent such as serpentinite. In Fig. 3, our boninite data lie within the fields for Layer 2a and 3 ocean crustal rocks, and are largely distinct from the fields for serpentinites, and for Izu-Bonin arc lavas. The LSB show on average higher δ[11]B at lower B enrichments and lower $^{87}Sr/^{86}Sr$, consistent with the signatures of Layer 3 crustal rocks. The LSB mantle source can best be explained by the involvement of contributions from subducted Layer 3 gabbros ± less altered Layer 2a basalts, in consideration of metamorphism effects during subduction initiation. By contrast, HSB mantle sources appear to be more consistent with

contributions from subducted sediments and/or altered Layer 2a basalt.

**Basalt/gabbro melt inputs to the low silica boninite mantle source.** Pearce et al.[16] and Li et al.[4] proposed melt inputs from subducting ocean crust with hornblende largely presented as residual mineral phase to explain decreases in εHf$_i$ and enrichments of Zr and Hf in the LSB, as εHf$_i$ correlates positively with measures of Hf enrichment (e.g., Sm/Hf and Ti/Hf). As shown in Fig. 4a, b, Ba/Nb and Sr/Nd in the low- and high-Ba LSB subgroups show inverse correlations with εHf$_i$. The subgroups have distinctly different Ba/Nb and overlapping, but different δ[11]B ranges, and they are not discriminated by Sr/Nd ratios. The patterns in Fig. 4 suggest that the systematics of B, Ba, Sr and Hf are broadly similar during the formation of the LSB. As Hf is uniformly immobile in hydrous fluids, correlations among these tracers mean that B, Ba, Sr, and Hf must have been added to LSB sources via the same non-fluid slab input mechanism. Neither altered basalts nor sediments are a satisfactory source for this LSB slab component, as both have much higher $^{87}Sr/^{86}Sr$ (Fig. 3c), as well as more enriched Pb isotopes[4]. Less altered basalt and/or gabbro-derived amphibolites from deeper in the ocean crust are the most reasonable slab constituent to contribute to the LSB mantle source.

That our LSB data in Fig. 4a includes low- and high-Ba/Nb subgroups likely points to the occurrence of multiple fluxing slab input events, each with different Ba/Nb signatures and different δ[11]B (Fig. 5). While the range in δ[11]B in the low and high Ba/Nb subgroups shows substantial overlap, lower δ[11]B in the low Ba/Nb samples is consistent with inputs from basalts/gabbros that have

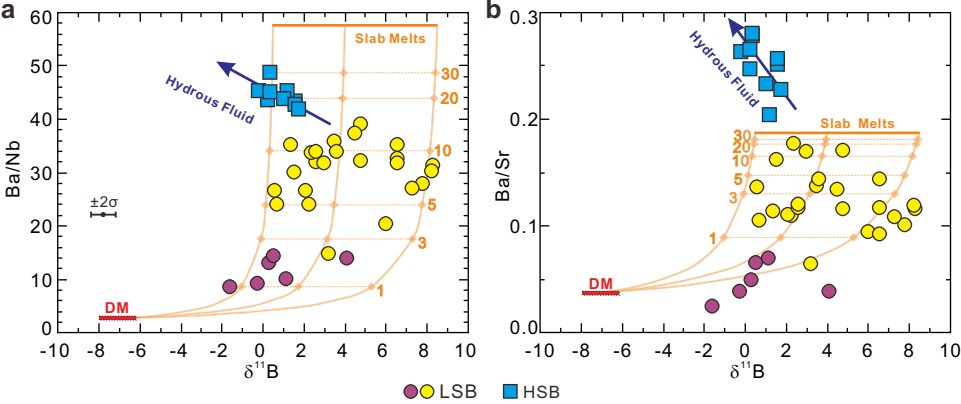

**Fig. 5 Boron isotopes and trace element systematics of the Expedition 352 boninites compared to mixing trajectories between the calculated slab melt/fluid and the depleted mantle.** Plots of **a** Ba/Nb and **b** Ba/Sr vs. $\delta^{11}$B for the International Ocean Discovery Program Expedition 352 boninites. Error bars ($2\sigma = 0.6‰$) are for boninite $\delta^{11}$B. The mixing curves are between the depleted mid-ocean ridge basalt mantle (DM[32,49,64]) and the calculated gabbroic protolith granulite melts. The numbers on the mixing curves represent the mass percentage of the slab melt. The $\delta^{11}$B of the gabbroic protolith granulite melts are assumed to be from +0.5‰ to +8.5‰. LSB: low-silica boninite; HSB: high-silica boninite.

suffered less metasomatism before melting. We have calculated trace element abundances for mineral-melt assemblages of representative amphibolite samples from the slab-mantle interface beneath the Oman ophiolite, one with high and one with low Ba/Nb (samples WT28A-5 and WT32[34]), under hornblende-bearing granulite facies of 900 °C and 0.9 GPa. This temperature is slightly higher than the reported peak metamorphic temperature of ~825 °C beneath the Oman ophiolite[46] but is consistent with recent modeling by Zhou and Wada[25]. The chosen pressure is comparable to the highest equilibrium pressure estimates for LSB genesis (0.4–0.9 GPa[12]). The calculations were performed using Perple_X[47]: detailed input parameters and outputs are provided in Supplementary Information. B-Ba-Nb-Sr-Nd-Hf abundances in the hornblende-bearing granulite melts (F = ~10%) were calculated from estimated mineral-melt assemblages, and the Nb, Nd, and Hf abundances of the "super composite" composition of ODP Site 801[48]. B, Ba, and Sr compositions were scaled to the B/Nb, Ba/Nb, and Sr/Nd of representative Oman amphibolites (Supplementary Table S1). Detailed mineral-melt partition coefficients for the elements of interest are listed in Supplementary Table S2. Mixing arrays between the hornblende-bearing granulite melts and the depleted mantle (depleted DMM of Workman and Hart[49]) were also calculated assuming $\varepsilon$Hf = 13 for the granulite melts and $\varepsilon$Hf = 19.5 for the depleted mantle (Fig. 4a, b). Our calculations indicate that 1–10 wt.% melt inputs from basaltic protolith granulites with high and low Ba/Nb can explain the Ba/Nb - $\varepsilon$Hf variations for the two LSB subgroups, but do not explain the Sr/Nd versus $\varepsilon$Hf variations, as Sr/Nd is too low (Fig. 4b).

We have also calculated melt compositions derived from hornblende-bearing granulite with gabbroic protoliths, using the average gabbro composition of the Atlantis Bank Massif (IODP Hole U1473A, SW Indian Ridge[50]; Supplementary Table S1), scaled to the B/Nb and Ba/Nb variation seen in Oman metamorphic sole amphibolites. Inputs of melts (3–20 wt.%) from granulites with oceanic gabbro protoliths generally explain the Ba/Nb versus $\varepsilon$Hf variations of the two suboroups of LSB (Fig. 4a). More depleted mantle compositions require smaller percentage melt inputs[12]. Sr/Nd ratios of gabbro-derived granulites are difficult to estimate. Assuming Sr/Nd ratios similar to their protoliths (average ~48[50]), the generated melts will have Sr/Nd comparable to that of basalt-derived high Ba/Nb granulite melts (Fig. 4b). Thus, melt inputs from gabbroic protolith

granulites, modified by metamorphic fluids with elevated Ba/Nb and Sr/Nd, may best explain the chemistry of the LSB.

Figure 5 compares our LSB data to calculated mixing arrays between depleted mantle and hydrous melts of lower crust gabbros under hornblende-bearing granulite facies conditions. As the LSB at Ba/Nb >30 have a wide range of $\delta^{11}$B, B isotope heterogeneity is necessary in the crustal melt inputs. A hornblende granulite melt component with $\delta^{11}$B between +0.5‰ and +8.5‰ best fit the Ba/Nb and Ba/Sr versus $\delta^{11}$B patterns of the high Ba/Nb subgroup LSB (Fig. 5). At 900 °C, crustal melting could induce at most 5‰ of $\delta^{11}$B fractionation between the melt and residual slab, assuming isotope fractionation factors similar to those for fluids and solids, though the strong B speciation bias in melts would reduce this effect[51–53]. Our calculations suggest the melt might at most be ~2‰ higher in $\delta^{11}$B than its slab source, given B elemental and isotopic mass balance.

Slab materials with $\delta^{11}$B between −1.5‰ and +6.5‰ best fit the LSB data in this study. The high and variable $\delta^{11}$B in granulite melts documented by our LSB results may partly be inherited from oceanic gabbros before subduction (e.g., ODP Hole 735B oceanic gabbros have high and variable $\delta^{11}$B from −4.3‰ to +24.9‰[28,54]), and this variability may later be reduced and partly homogenized by fluid losses before melting, and metasomatism by amphibolite fluid from deeper in the crust and/or fluids derived from lithospheric serpentinites. The initial melts appear to come from gabbros, which are deeper than basalts and sediments in the slab. Melting or dehydration of shallower slab materials are not evident from the LSB boron and radiogenic isotope data, indicating that the top of the slab was possibly scraped off during subduction initiation, exposing the deeper gabbros ± diabases to the hot mantle wedge.

**Basalt and sediment fluid inputs to the high silica boninite mantle source.** The data arrays for the HSB in Fig. 3 differ from those of the LSB in that $\delta^{11}$B is much less variable, ranging only from −0.2‰ to +1.8‰, more similar to values for average altered basalts, and to a lesser extent marine sediments, than to many of the LSB. In Fig. 3 the HSB data are coincident with Layer 2a and 3 basalts/gabbros, albeit at overall higher B/Nb than either of these constituents, or of marine sediments. In Fig. 4a, the HSB appear to plot as an extension of a trend of increasing Ba enrichment with declining $\varepsilon$Hf$_i$, defined by the high Ba/Nb LSB

and the Expedition 352 FAB. However, other mobile elements in the HSB show different systematics (Fig. 4c, d; Supplementary Fig. S2), pointing to the involvement of component(s) other than those responsible for LSB genesis.

The HSB differ from the LSB in that their petrogenesis appears to involve the mixing of at least two distinguishable slab components. One slab endmember is a high Ba slab-melt, such as is evident in the high Ba/Nb LSB. $\varepsilon Hf_i$ declines as Ba becomes more enriched, suggesting a maximum Ba/Nb for this component where $\varepsilon Hf_i$ reaches its lowest, most "Pacific-like" value. Sr/Nd shows a similar, if more scattered, pattern of increase relative to $\varepsilon Hf_i$ (Fig. 4b), but plotted versus $^{87}Sr/^{86}Sr_i$ the HSB trend to higher Sr/Nd and Sr isotopic ratios from a minimum Sr/Nd of ~50, the median value of our LSB data (Fig. 4c). Increases in B/Nb and Ba/Nb in the HSB are associated with decreases in $\delta^{11}B$ and $\varepsilon Nd_i$, and with increases in $^{87}Sr/^{86}Sr_i$, suggesting coherent behavior among B, Ba, Sr and Nd during HSB formation (Fig. 3 and Supplementary Fig. S2). $^{87}Sr/^{86}Sr_i$ increases in the HSB occur at nearly constant to declining La/Sm, while La/Sm varies markedly in the LSB at near-constant $^{87}Sr/^{86}Sr_i$, consistent with variable melt inputs from a crustal source (Fig. 4d). $\varepsilon Hf_i$ is uniformly low in the HSB, irrespective of the B/Nb, Ba/Nb, or Sr and Nd isotopic variations. This apparent decoupling of Hf isotopes from other tracers, and the patterns seen in mobile element enrichments and in La/Sm suggest that additional B, Ba, Sr, and REE inputs to HSB mantle sources may have occurred via hydrous fluids, in which Hf would be immobile[55,56]. Hydrous fluid inputs will decrease Hf/Nd, and greater fluid inputs will result in higher degrees of melting, that would be reflected in decreases in Hf/Ti and La/Yb (Supplementary Fig. S2). Slab-derived fluids will also have higher B/Nb than their protoliths, consistent with the comparatively high HSB values in Fig. 3a. $\varepsilon Hf_i$ in the HSB thus appears to reflect the same slab contributor seen in the LSB (i.e., melts from less altered basalt/gabbro), which was likely pervasive in the LSB residuum that was re-melted to produce the HSB. The other, more fluid-mobile tracers reflect new, likely fluid-mediated inputs from different slab contributors.

Bulk mixing calculations based on the HSB data patterns in Fig. 3 indicate that the highest $^{87}Sr/^{86}Sr_i$ HSB sample requires >10% sediment involvement (Fig. 3c). At >5 wt.% sediments, Pb isotopes in the HSB become sediment dominated[4]. Amphibolite fluids (based on the Oman amphibolites) derived from a subducted slab with 5–10 wt.% sediments should have Ba/Sr from 0.26 to 0.32[34], similar to the highest values in the HSB (Fig. 5b). Therefore, an estimate of 5–10 wt.% sediments involved in the slab inputs to the HSB appears reasonable.

In Fig. 6, Cs/La is negatively correlated with $\varepsilon Nd_i$ in the HSB. As such, the highest HSB Cs/La ratio most closely reflects the slab fluid signature (Fig. 6). We have calculated the dehydration temperature of the slab, based on experimental results on marine sediments from Hermann and Rubatto[57]. Our peak temperature estimates are between 780 °C and 840 °C, assuming that altered basalts and sediments experienced no Cs loss before additions to the HSB mantle source; as such our temperature estimates are likely maximum possible values. These estimates are similar to reported solidus temperature for slab sediments (i.e., <775 ± 25 °C at 2 GPa, or 810 ± 15 °C at 3 GPa)[58], pointing to overall lower temperature conditions at the plate interface during HSB generation. At ~800 °C, slab dehydration could induce a maximum of ~6‰ of $\delta^{11}B$ fractionation between the fluid and residual slab, assuming near-neutral fluid pH values[51,52]. Given the lowest HSB $\delta^{11}B$ value of ~ −0.2‰, the dehydrated slab may have had $\delta^{11}B$ as low as −6‰. Bulk mixing calculations suggest that a mixture of 5–10 wt.% subducted sediments (B = 31.5 µg/g; $\delta^{11}B$ = −11.2‰[45]) and 90–95 wt.% of Oman-like amphibolites (B = 33.3 µg/g; $\delta^{11}B$ = +3.75‰[34,35]) would have $\delta^{11}B$ between

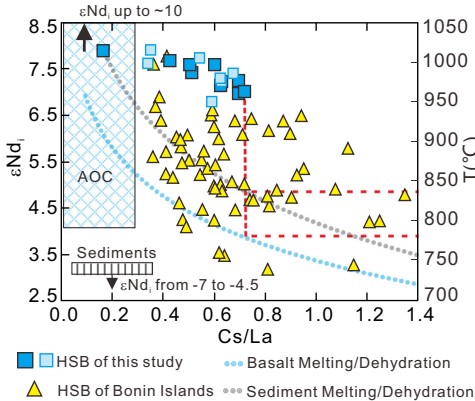

**Fig. 6 Elemental ratios and neodymium isotopes characterize slab makeup and temperature during Expedition 352 high-silica boninite generation.** $\varepsilon Nd_i$ (left y-axis) vs. Cs/La diagram for the Expedition 352 and Bonin Islands[60] high-silica boninite (HSB), Izu-Bonin trench sediments[38] and altered basalts[48,67]. The temperature (°C; right y-axis) versus Cs/La paths for sediment (light gray dotted line; calculated after Hermann and Rubatto[57] at 2.5–4.5 GPa) and basalt (light blue dotted line; calculated after Kessel et al.[68] at 4 GPa with Ocean Drilling Program Site 801 "Super-Component" as starting material) dehydration/melting are also plotted for comparison. The red dashed lines are used to estimate the slab dehydration/melting temperature during HSB generation of this study according their highest Cs/La ratio. AOC: altered oceanic crust.

+3.2‰ and +2.3‰. These values are higher than our estimated slab component (< −0.2‰, probably as low as −6‰), and may point to an overestimate of the B and $\delta^{11}B$ in subducted basalts, indicating the subducted basalts may not have been metasomatized by amphibolite- or serpentinite-derived fluids during HSB generation.

## Discussion

Dating of Izu-Bonin FAB and boninites indicate that early IBM magmatism evolved from FAB (51.9–51.3 Myr) to boninite (51.3–50.3 Myr), and thus from ≈0% to significant additions of slab-derived materials in <2 Myr[7]. Early radiogenic isotope[4] and elemental studies[12] have suggested that LSB mantle sources include melt contributions from oceanic crust, while HSB mantle sources involving contributions from subducting sediments. Our new data for Expedition 352 boninites record details about these early exchanges, which appear to provide insights into physical constraints on the slab input processes and document rapid thermal evolution of the slab during subduction initiation. Higher and more heterogeneous $\delta^{11}B$ in the LSB offer evidence the first slab additions came from less altered oceanic crust, most likely lower crustal gabbros. Sm/Hf, Ti/Hf[4,16] and Ba/Nb ratios in the LSB place melting temperatures on the slab at 900–950 °C. This indicates early heating of the nascent slab due to interactions with upwelling hot mantle. Slab inputs from the upper portions of Pacific plate crust and sediments appear to have begun later, as reflected in the compositions of the overlying HSB. Their higher B and lower, more uniform $\delta^{11}B$ are consistent with slab basalt + sediment contributions most likely via hydrous fluids generated at ~800 °C. This change indicates the start of cooling of the slab-mantle interface via refrigeration by early subducted crust and lithosphere. While a small role for B from dehydrated lithospheric serpentinites in the LSB is not entirely precluded by our results, it is clear that unlike in mature volcanic arcs, serpentinite does not play a controlling role in the petrogenesis of either boninite subtype.

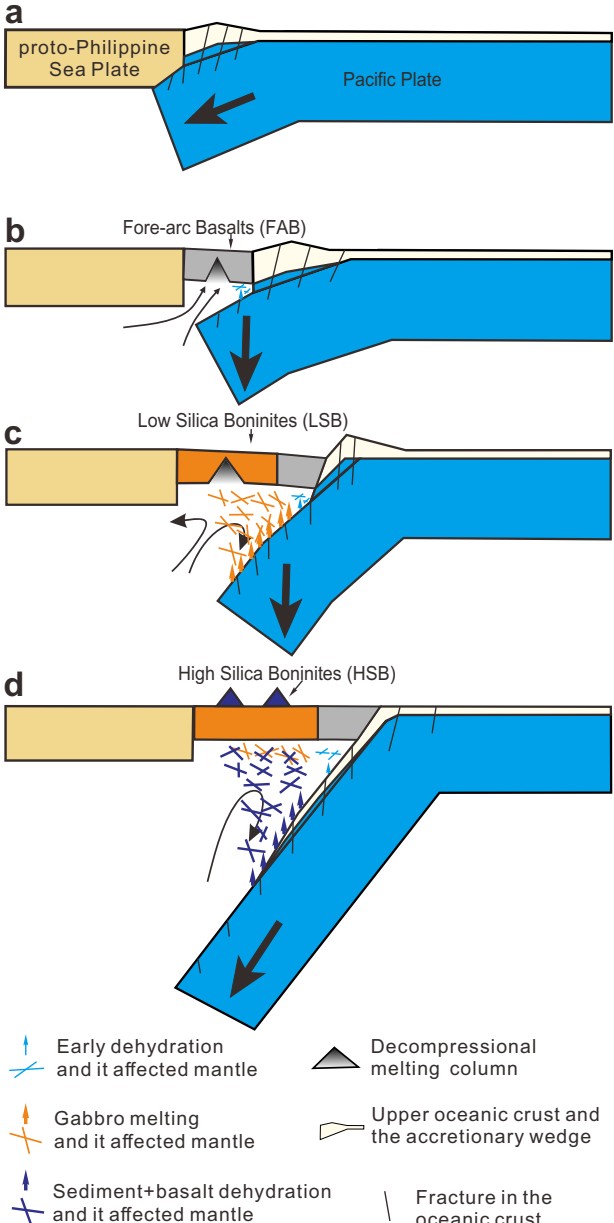

**Fig. 7 Cartoons depicting subduction initiation and the slab dehydration/melting processes.** The model starts at a hypothetical transform fault at the boundary between the proto-Philippine Sea Plate and Pacific plate[69]. **a** Shallow angle convergence of the Pacific plate beneath the proto-Philippine Sea Plate resulted in accretion of Pacific sediments and upper crust. The lower crust experienced higher temperature metasomatism resulting in higher overall $\delta^{11}$B. **b** Decompression melting in response to slab rollback, producing forearc basalt (FAB). **c** low-silica boninite (LSB) genesis, depleted FAB mantle source metasomatized by lower crust melting components. **d** high-silica boninite (HSB) genesis, with an ultra-depleted mantle wedge after LSB extraction further metasomatized by hydrous fluids from a mixed ocean crust source + sediment.

The comparatively rapid temporal changes in slab inputs reflected in the Expedition 352 boninites allow us to place tighter constraints on how the downgoing plate and mantle interact during subduction initiation (Fig. 7). Slow and shallow early convergence apparently disrupted the upper portions of Pacific plate crust, producing a proto-accretionary wedge that included basaltic crustal rocks as well as sediments. Early slab subsidence

led to mantle upwelling that triggered the eruption of voluminous FAB, which in their later stages show evidence for some minor slab influence[17]. The subducting portion of the downgoing plate metamorphosed due to burial and contact with hot upwelling mantle, liberating melts and fluids from its leading edge. As subduction-driven mantle convection had not started, a trapped domain of depleted, once-melted mantle (FAB residual mantle) developed near the slab interface, and the fluxing by heterogeneous melts from the gabbroic section of the Pacific slab modified this trapped mantle and triggered LSB magmatism. Increasing slab subsidence rates ultimately resulted in whole-plate subduction, allowing cooler slab crust and lithosphere to penetrate deeper into the mantle. The cooling of the subduction interface likely precluded further crustal melting, but permitted the generation of hydrous fluids from subducting basaltic crust and sediments that further fluxed this melt-modified residual mantle, and resulted in HSB magmatism. As the forearc gradually cooled, the locus of HSB volcanism appears to have moved westward, away from the trench, after ~50 Myr, consistent with younger radiometric dates on HSB from the Bonin Islands (~48–46 Myr) and other localities inboard of the Expedition 352 drillsites[59–61]. Increased sediment subduction combined with a relatively cooler subduction interface leads to the relatively higher Cs/La and lower $\epsilon Nd_i$ signatures of Bonin Islands HSB (Fig. 6). The geometry and character of HSB magmatism, in which slab fluids trigger mantle melting in a predominantly non-extensional environment, are similar to what is encountered in volcanic arcs[13]. HSB thus appear to represent "last gasp" forearc magmatic activity, following which the start of subduction-related mantle convection leads to the development of a cooler, stable forearc. Slab cooling leads to low temperature hydrous fluid releases, hydrating the forearc mantle wedge and ultimately producing the "subduction channel" of disaggregated and serpentinized upper and lower plate materials along the plate interface, which is dragged down by the subducting slab to become a high $\delta^{11}$B component in arc volcanic rocks. Ultimately deeper melting of fertile mantle wedge materials well inboard from the trench, triggered via inputs from a deeper slab, produce calc-alkaline arc lavas in the Izu-Bonin system after 44 Myr[60].

## Methods

**Sample selection and preparation.** Samples examined in this study were selected from the working halves of cores for Holes U1439A, U1439C, and U1442A, to represent both the natural chemical variability of the IODP Expedition 352 boninites and to reflect overall core stratigraphy. Hole 1439C best reflects the lithologic variability of Expedition 352 boninite and associated differentiates, comprising nine volcanic units and one basal intrusive unit. The stratigraphy of Hole U1442A is similar to Hole U1439C but simpler, comprising only four volcanic units. Only one volcanic unit (all HSB) was sampled in the three boninite cores recovered from Hole U1439A[12]. Therefore, in this study we have investigated Hole U1439C in the greatest detail. The HSB from <250 mbsf in all three holes are relatively fresh and show similar geochemical variations[4,12]. Two U1439A samples were selected to supplement the HSB samples from Hole U1439C. Four HSB samples, representing intrusive dikes through LSB sequences from >250 mbsf in Holes U1439C and U1442A, were identified for analysis via our pXRF chemostratigraphy[10–12], for comparison with the shallower HSB samples. Fresh LSB were only rarely recovered during Expedition 352, so LSB samples from Hole U1442A were selected to supplement those from Hole U1439C. Our samples included fresh boninites selected specifically for this work, and LSB drawn from larger 'POOL' samples, that had been chosen for coordinated post-cruise measurement. The LSB and HSB sample subsets both included some boninitic differentiates, properly termed high-magnesium andesites (HMA). For simplicity of description, we do not discriminate the HMA differentiates from their respective/parental HSB or LSB. The major and trace element abundances and Sr-Nd-Pb-Hf isotopic ratios of the 'POOL' LSB samples are published in Li et al.[4].

Below is a brief summary of the analytical methods and quality controls, which include B isotopes for the 'POOL' samples and trace elements and Sr-Nd-Hf-B isotopes for the 'Non-POOL' samples. All 'POOL' samples and a subset of 'Non-POOL' coarse grained samples were leached with 6 M HCl before sample digestion for B isotope analyses[62] to remove seawater-derived B. Leaching experiments done on our fresh samples resulted in no change in their $\delta^{11}$B, indicating that these

### Legend (Fig. 7)

Early dehydration and it affected mantle

Gabbro melting and it affected mantle

Sediment+basalt dehydration and it affected mantle

Decompressional melting column

Upper oceanic crust and the accretionary wedge

Fracture in the oceanic crust

samples have not been affected by seawater alteration. All $\delta^{11}B$ measurements reported for the 'POOL' samples were conducted on leached powders.

**Trace elements**. Trace element analyses were performed at Guizhou Tongwei Analytical Technology Co., Ltd. on a Thermal X series 2 ICP-MS equipped with a Cetac ASX-510 AutoSampler. After digestion, samples were dissolved in 3 ml of a 2 M $HNO_3$ stock solution that was then diluted to 4000:1 in 2% $HNO_3$, and spiked with 12ppb $^6Li$, 6ppb $^{61}Ni$, Rh, In and Re, and 4.5ppb $^{235}U$ internal standards. The USGS reference material W-2a was used as reference standard and BIR-1, BHVO-2 and several other reference samples were crosschecked. Instrument drift and mass bias were corrected using these internal spikes and external monitors. Based on results for rock standard BIR-1a, the analytical precision for the rare earth elements (REE) and most of the other species analyzed is ±1–5%.

**Sr-Nd-Hf isotopes**. Sr-Nd-Hf isotopes were analyzed at the Guangzhou Institute of Geochemistry, Chinese Academy of Sciences (GIG-CAS). Sr isotopic ratios were measured on a Thermo Triton TIMS, and $^{87}Sr/^{86}Sr$ was corrected for instrumental mass fractionation by normalizing to $^{88}Sr/^{86}Sr = 8.375209$. The Sr isotope results are reported relative to SRM 987 of $^{87}Sr/^{86}S = 0.710248$. Rock standards BHVO-2 ($^{87}Sr/^{86}Sr = 0.703506 \pm 5$, 2SE), AGV-2 ($0.703976 \pm 7$), JB-3 ($0.703432 \pm 7$), and W-2A ($0.706952 \pm 7$) were prepared and measured along with the unknowns to monitor the quality of Sr analyses. We analyzed Nd-Hf isotope ratios on a Finnegan Neptune MC-ICPMS. Neodymium and Hf isotope ratios were monitored and corrected for mass bias using the values of $^{146}Nd/^{144}Nd = 0.7219$ and $^{179}Hf/^{177}Hf = 0.7325$, respectively. The Nd isotopic ratios are reported relative to $^{143}Nd/^{144}Nd$ of JNdi-1 = 0.512115. The rock standard BHVO-2 was repeatedly analyzed with chemical treatment in separated aliquot for each analysis, yielding a results of $^{143}Nd/^{144}Nd = 0.512986 \pm 4$ (1 SD, $n = 2$). Analyses of rock standard JB-3 and W-2A gave $^{143}Nd/^{144}Nd = 0.513043 \pm 5$ and $0.512529 \pm 5$ (SE), respectively. The Hf isotopic ratios are reported relative to $^{176}Hf/^{177}Hf$ of JMC 14374 = 0.282189 (corresponding to JMC475 of 0.282158). Analysis of rock standard BHVO-2 and BCR-2 yield $^{176}Hf/^{177}Hf = 0.283097 \pm 4$ and $0.282858 \pm 3$ (SE), respectively.

**B and B isotopes**. B and B isotopes were analyzed at GIG-CAS[62,63]. B concentrations for the 'Non-POOL' sample was measured on a Varian Vista Pro ICP-AES, equipped with an HF-resistant Teflon spray chamber and an $Al_2O_3$ injector. B was measured using the 249.772 nm spectral line. B-5, JB-2, JB-3 and JR-2 were chemically prepared with the samples and used as external standards for calibrating B concentrations. The analytical precision for our B concentration measurements was generally better than 5% (RSD). B isotope measurements were performed using the Finnegan Neptune MC-ICPMS in sample-standard-bracketing (SSB) mode. NIST SRM 951 dissolved in B-free Milli-Q deionized water was used as the bracketing standard, and the results of measured samples were expressed as $\delta^{11}B$ relative to SRM 951[62]. The internal precision for $\delta^{11}B$ was better than ±0.05‰ (1SE), and external precision for $\delta^{11}B$ is better than ±0.40‰ (1 SD) based on our long-term results for SRM 951. The standard reference samples B-5, B-6, JB-2, AGV-2, and JR-2 were repeatedly prepared and analyzed along our unknowns to monitor the quality of the B isotope measurements. Measured $\delta^{11}B$ values for the reference samples were: AGV-2: −4.36 ± 0.68‰ (2 SD, $n = 3$); B-5: −4.71 ± 0.49‰ (2 SD, $n = 9$); B-6: −2.86 ± 0.62‰ (2 SD, $n = 9$); JB-2: +7.29 ± 0.60‰ (2 SD, $n = 9$); JB-3: 6.74 ± 0.09‰ (2 SD, $n = 2$); and JR-2: 3.10 ± 0.77 ‰ (2 SD, $n = 11$).

## Data availability

The authors declare that the data generated or analyzed during this study are included in this published article and its Supplementary Information files.

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

## Acknowledgements

All authors thank the International Ocean Discovery Program (IODP) for samples. H.Y.L. and J.G.R. appreciate the contributions of the members of the IODP Exp. 352 Scientific Party as well as the officers and crew, and scientific support staff, on the *JOIDES Resolution*. H.Y.L., X.L., and Y.G.X. were supported by the National Natural Science Foundation of China (NSFC Project 41922020), the Strategic Priority Research Program of the Chinese Academy of Sciences (Grant No. XDB42020201, XDB18000000) and the Key Special Project for Introduced Talents Team of Southern Marine Science and Engineering Guangdong Laboratory (Guangzhou) (GML2019ZD0202). J.G.R. thanks the USSSP for initial Expedition funding, NSF for grant OCE-1558855, and USF for a UNI-Nexus travel and collaboration award. H.Y.L. thanks Le Zhang, Jinlong Ma, Xuefei Chen and Gangjian Wei for their contributions to the analyses and running of the GIG-CAS laboratories. This is contribution No. IS-3123 from GIGCAS.

## Author contributions

H.Y.L. and J.G.R. conceived and designed the study, comprehensively interpreted the geochemical data and drafted the primary manuscript. X.L. conducted the geochemical analyses. C.Z. conducted the pseudosection calculations. Y.G.X. contributed to improving the interpretation of the data and the writing of the manuscript.

## Competing interests

The authors declare no competing interests.
