## [Peer Review File · Nature Communications]

Boron isotopes in boninites document rapid changes in slab inputs during subduction initiationReviewers' comments:

Reviewer #1 (Remarks to the Author):

Review of Boron isotopes track the heating and cooling of the Izu-Bonin slab during subduction initiation by Hong-Yan Li, Xiang Li, Jeffrey G. Ryan and Yi-Gang Xu.

Review by George Cooper

This manuscript presents boron isotope data from Izu-Bonin Mariana boninites recovered during IODP Expedition 352. Boron isotopes provide a novel means to trace the involvement of changing fluid sources during subduction initiation and to my knowledge, this is the first boron study of boninites. The paper proposes that the heating and cooling of the downgoing plate during subduction initiation is reflected in the B isotope data and that the B isotopes place new constraints on the initial slab inputs. Whilst I agree with most of the interpretation, I found the discussion of the figures and data hard to follow (particularly evident in the discussion surrounding Fig. 4) and I think that this could be significantly expanded upon.

A number of the supplementary figures could be moved into the main text to help explain and expand the discussion. Fig. 5 could be moved into the supplementary or dropped. It is currently difficult to discern what unique evidence boron isotopes bring to the subduction initiation story. However, this may become clearer after some restructuring of the manuscript.

There is little mention of the involvement of serpentine within the downgoing plate. Are there any estimates as to how serpentinised the earliest subducted Pacific plate would be? Would you expect all serpentine dehydration to occur at shallow levels due to the higher initial temperatures? In which case, the initial fore-arc basalts may have a heavy B signature that is lost before the LSB begins. It would be worth discussing these points.

Below I make a number of suggestions and comments on specific parts of the manuscript.

Lines 48-52: This sentence seems out of place and could be moved to the end of the introduction. The introduction is short and could be expanded to discuss the work that has already been done on these and/or similar samples (e.g. Li et al., 2019) and what new insights can potentially be gained from boron isotopes. In addition, I think that the IBM subduction initiation model could be discussed in detail here.

Line 69: You refer to boron abundances in Fig. 2, however these are not presented in Fig. 2 so it is difficult to follow the logic of the discussion.

Line 70: 'clear correlations with MgO and TiO₂' Describe how the B/Nb correlates with these?

Line 78: Are the IBM arc volcanics mentioned here also plotted in Fig. 2? The volcanics currently in Fig. 2 seem to range from ~+4‰ to +12‰. Please clarify.

Line 97: More information on Hf isotope systematics is needed here to explain to the reader what we can learn from Fig. 3.

Lines 108 to 111: Confusing sentence which refers to Nb concentrations in highly depleted mantle whilst discussing Abyssal serpentinites. Please clarify. This discussion would seem better placed in previous paragraphs where Fig. 2 is described.

Line 115: Again, describe what the correlations are. Negative correlations?

Line 126: You describe mixing curves based on B/Nb concentrations, but it is only Ba/Nb concentrations that are plotted in Fig. 4. This makes interpretation of Fig. 4 very confusing and hard to follow the logic.

Line 130: Mixing lines between DM and the highest Ba/Nb LSBs are described. However, the

mixing lines do not appear to go close to the highest Ba/Nb samples. Please explain further.

Line 135: Can you explain what is meant by 'a better match to the pattern'. In both cases, almost all of the data plots outside of these mixing trajectories!

Line 172: An alternative explanation is that the sampling of cores only represents a small snapshot of the total range of compositions in HSB lavas.

Line 189-214: It is not apparent that any of these conclusions were drawn from the B isotope data presented in this manuscript. Currently, it appears like the boron story was fitted to a previously established subduction initiation model. While I agree that the new data is exciting and a worthwhile contribution, the authors need to clearly state what NEW evidence boron isotopes bring to the subduction initiation model. I would consider moving most of this discussion to the introduction to explain what we currently know about subduction initiation, before going on to present the new data. I believe this would provide a logical flow to the manuscript and highlight the importance of the new data.

Figures:

Fig. 2: Uncertainties on the data should be plotted and/or stated within the figure caption. In panel (a) there is a reduced dataset presented. Is there a reason for having B isotope data without B concentrations? Were these the leached samples? This needs explaining within the text or the methods section.

Fig. 3: Uncertainties on the data should be plotted and/or stated within the figure caption.

Fig. 4: Label panel (a) with 'undehydrated' and (b) with '60% B loss from slab' so that it is easier to follow the different plots. This figure needs a lot more information (ideally in a key) so it can be interpreted. What does the blue and red star represent? What are the three coloured shaded areas? Are the mixing lines you describe the thick grey lines? In the figure caption (line 472), fluid compositions for 5% dehydration are discussed with no reference to the Figure - is this the green arrow?

Fig. 5: I am not sure it is necessary to have this cartoon. This same schematic is in most Izu-Bonin subduction initiation papers, so could simply be cited within the text. Currently, this figure gives the impression that the data is being fitted into an already established model.

Figure S2: There is no key or explanation as to what the cross symbols represent.

Reviewer #2 (Remarks to the Author):

This is an interesting paper, with new data for B and B-isotopes in recently drilled rocks from the IBM forearc subduction initiation sequence. As it stands, the paper needs some work to be appreciated by a geoscience audience. Major flaws are: 1) the absence of context, i.e., how the paper addresses a compelling issue in Earth sciences, and 2) that results and implications are mired with some details, while other information is lacking. The paper should be revised to improve organization and clarity.

Here are some suggestions:

- 1) Introduce the big picture. What exactly has been discovered about subduction? My take is that there are distinct early and late fluids. Why is it important to know this? Coverage of the importance of the IBM system and the special behavior of B as an element is good, but an early introduction to the behavior of B isotopes is needed (i.e., what is covered on lines 236-251).
- 2) Some details about what is selected for study is needed up front: a) Why are there two groups of LSB? b) What are their stratigraphic/petrographic differences or are they distinguished only by geochemistry? c) Why aren't HMA mentioned even though they form a large part of the section in Figure 1c? d) where are the FABs in the section?
- 3) Why does the variation in Hf-isotopes prove that there is input of Hf from the slab? Why couldn't mantle sources have variable Hf isotopic composition?

4) Label the difference in the models plotted in Figs. 4 a and b clearly. Why is the early extraction of B significant?

For Nature Communications, the conclusions should be clear and significant. What are they and which are unique to this study? Two fluids involved in early slab dehydration? Early fluids different from the "typical" ones involved in arc magmatism? Slab cooling during early subduction?

Summarize the evidence that supports each conclusion. Overall, it seems that the B-isotope data are not fully conclusive; instead they are being built into a self-consistent model shown in Figure 5. If this can be improved the paper will be more influential.

Some details:

Fig. 4 caption and discussion: Is the process modelled: 1) mixing of components (fluids/melts) of serpentinite and AOC with the mantle, followed by melting, or 2) mixing of the components with melts of the mantle? It would be good to back this model up with an array of other trace elements.

Fig. 2a – Why is there only one purple LSB dot?

Fig. 3b – Why are there no FABs on this plot?

Fig. 5a What is "IZA" on this diagram?

Reviewer #3 (Remarks to the Author):

The authors provide new B and d11B data on early arc boninites, and attempt to interpret those with respect to slab temperature. Unfortunately, the study is far from being convincing. I find it difficult and confusing to read. There is not strong convincing case made how the data could help to infer slab T evolution. I made a couple of major comments, and a few comments linked to the text, which I hope help understanding why I found the manuscript lacking.

1. The use of terms 'fluid' vs. 'melt' is very confusing. For example, in the abstract it reads: "Correlations between boninite B isotopes, Ba/Nb, and ϵHf indicate that both dehydration and melting of subducted ocean crust under amphibolite facies conditions produced the slab fluid flux that triggered melting in the LSB mantle wedge". (There are more examples in the manuscript). How can slab melting produce a fluid flux only? These term need to be distinguished stringently.

2. You argue that high d11B and low B points to an early 'hot' slab. How does that compare to the active 'hot-slab' Cascades arc where both d11B and B are low? You are that high B and low d11B indicates a cooler slab? How does this tally with the active 'cool-slab' Izu Bonin arc where both B and d11B are high?

I wonder if you recast your data from the boninite into a global compilation, how they would appear?

3. There is no quantitative attempt on characterized the alleged change in slab T. It's all based on relative observation and inferences. There are no constraints.

What is the temperature significance of the B and B isotope signature? Your conclusions are based on comparative behavior and distribution of B and B isotope during subduction, and ideas that have been forward on how to explain diversity in different arcs (not shown). It is a long leap to recast these observations (not even quantitative models) into inferring slab T variation during early subduction.

4. The time-progressive models is not really shown. The time interval studied with boninites (1.5 Myr duration) is short compared to the overall length of the arc. Why do you expect/explain drastic slab T change within this short period that produced slab-influenced boninite series? It would be more instructive to have added B and d11F from FAB and post-boninite series, and see where the alleged slab T changes correlate with chemistry.

Besides, do you have evidence that the HSB are really significantly younger than the LSB? Is there are time lapse between them relevant to the time scales needed to transmit slab processes to arcs?

Comments linked to the text.

L67: first you say that B in LSB and HSB are distinct by a factor of 3, and then you speak of

overlap? Confusing.

L70-72: are you suggesting that the high MgO, low-TiO₂ (younger) HSB are produced by higher extent of melting despite their higher B/Nb relative to the LSB? Or why do you discuss the extent of melting here?

L82: here you speak of slab melts - why then of fluids in the abstract?

L87: Reference missing after period.

L88: in Fig. 2, AOC is at +4 d11B, here you write d11B = -7.

L101: Here you do have slab melts. Does that mean that the Ba and B are transported in different slab fluids? Confusing.

L111: You introduce here serpentinite a B source, which were never mentioned in the abstract.

L113: Does that mean, that gabbros could also be a B source? Or are they excluded?

L132: since the subducting slab loses B from the start, there is no wonder that B is lost from slab? How does the estimate of 60% B loss compare to other studies?

L129-139: confusing, incomprehensible -- how do the low d11B HSB fit into this modeling? What do you mean by 'our data' in L136? The LSB?

L140: I assume that this is the model to explain the LSB? Or both LSB and HSB?

L150: but the eHf of the HSB is very low. You explained earlier that this low eHf was a melt?

L161-163: I think you need to be clear whether you have one or several components from slab.

L218-220: That you should show by a global compilation.

L232: Reference missing after period.

Figure 4: why there are two panels? What is the difference?

Figure S1: Legend/caption missing - what are the symbols?

Figure S2: what are the crosses?

DETAILED RESPONSES TO THE REVIEWERS

Overall response to reviewers:

In preparing our response to reviewers, we compiled additional data on boron and strontium isotopic variability in the oceanic crust. We have included a new plot highlighting the B-Sr isotope relationships for our samples (Figure 4). This diagram helps confirm that the first subduction inputs in the Izu-Bonin system arise from melting of the lower oceanic crust, and rules out any significant role for slab serpentinites. These new findings greatly enrich our understanding on subduction initiation. Our results favor a more actively convergent model for IBM subduction, as opposed to a purely passive vertical foundering of the Pacific plate at the beginning of subduction (i.e., Stern and Bloomer, 1992), such that the upper portions of the ocean crust are accreted initially, and do not contribute to magmatism. We have modified the manuscript to include discussion of how the Pacific slab enters the nascent subduction zone. We have changed the title to **‘Boron isotope evidence for upper crust accretion and a cooling slab during subduction initiation’** and modified the ‘introduction’ and ‘discussion’ sections accordingly.

The evidence for early rapid heating and subsequent cooling of the downgoing plate during subduction initiation is solid. In the revision, the amphibolite-facies crustal melt (5% melting) component assumes oceanic gabbros with $\delta^{11}\text{B}$ from 1‰ to 8.5‰ as starting compositions, well within the observed range for Layer 3 ocean ridge gabbros. Involvement of slab serpentinite and an early dehydration event on the slab (60% B loss) are no longer necessary. The mixing arrays between depleted mantle and slab melts are now simpler, clearer and more realistic in the revision.

Reviewer #1: (Remarks to the Author):

Review of Boron isotopes track the heating and cooling of the Izu-Bonin slab during subduction initiation by Hong-Yan Li, Xiang Li, Jeffrey G. Ryan and Yi-Gang Xu.

Review by George Cooper

This manuscript presents boron isotope data from Izu-Bonin Mariana boninites recovered during IODP Expedition 352. Boron isotopes provide a novel means to trace the involvement of changing fluid sources during subduction initiation and to my knowledge, this is the first boron study of boninites. The paper proposes that the heating and cooling of the downgoing plate during subduction initiation is reflected in the B isotope data and that the B isotopes place new constraints on the initial slab inputs. Whilst I agree with most of the interpretation, I found the discussion of the figures and data hard to follow (particularly evident in the discussion surrounding Fig. 4) and I think that this could be significantly expanded upon.

A number of the supplementary figures could be moved into the main text to help explain and expand the discussion. Fig. 5 could be moved into the supplementary or dropped. It is currently difficult to discern what unique evidence boron isotopes bring to the subduction initiation story. However, this may become clearer after some restructuring of the manuscript.

We have added the new $^{87}\text{Sr}/^{86}\text{Sr}$ versus $\delta^{11}\text{B}$ (Figure 4) diagram, the ϵNd versus B/Nb diagrams (Figure 8; from the appendix), and the ϵHf versus Nb/Ti diagram (Figure 6; from the appendix) to the main body of our revised text to expand our discussion. B-Sr isotope correlations confirm that the first crust to melt is the gabbroic lower crust during the subduction initiation. We have modified our model cartoon (Figure 9) accordingly in the revision. The new cartoon indicates a more detailed dehydration/melting process on the nascent subducted slab, which will help the readers to catch the main idea of the paper. Therefore we select to preserve this cartoon in the main text.

There is little mention of the involvement of serpentine within the downgoing plate. Are there any estimates as to how serpentinitised the earliest subducted Pacific plate would be? Would you expect all serpentinite dehydration to occur at shallow levels due to the higher initial temperatures? In which case, the initial fore-arc basalts may have a heavy B signature that is lost before the LSB begins. It would be worth discussing these points.

In our original submission, we attempted to use the average altered ocean crust (AOC) value of Smith et al. (1995) ($\delta^{11}\text{B}=+3.7\text{‰}$) to model the slab melt component. This was not ideal, as altered oceanic crust is highly heterogeneous in terms of both boron and other tracers (Smith et al., 1995), and because we had LSB significantly higher than average AOC, which led us to consider serpentinite as a component. In our revised submission, the new $^{87}\text{Sr}/^{86}\text{Sr}$ versus $\delta^{11}\text{B}$ diagram (Figure 4) clearly shows that the best fit to our data is a slab melt input from lower oceanic crust, which removes the need to consider serpentinite involvement to explain the B isotopes. $\delta^{11}\text{B}$ of the lower oceanic crust is highly heterogeneous, based on published data (Smith et al., 1995; McCaig et al., 2018), which is consistent with the variability in $\delta^{11}\text{B}$ we see in the LSB. As well, lower crustal gabbros are considerably lower in boron than serpentinites, more in line with the overall low B concentrations of the LSB. In the revision, we calculate the amphibolite crustal melt ($F = 5\%$) component using gabbro with $\delta^{11}\text{B}$ that varies between $+1\text{‰}$ and $+8.5\text{‰}$ as starting compositions (Figure 7).

The generation of fore-arc basalts (FAB) is controlled by extension-related decompression, and in later stages potentially a fluid-flux. The FAB sampled during Expedition 352 do not show a clear fluid contribution to their mantle sources. However, Reagan et al. (2010) found some evidence for the fluid involvement in lavas transitional between FAB and Boninite. Our Exp. 352 recovered samples didn't find any transitional rocks between FAB and boninite. We would expect such transitional rocks would have heavier B isotopes than MORB. Whether FAB might record heavier B isotopes would depend on how 'Early fluids' might interact with FAB mantles, and what the origins of such fluids might be.

Below I make a number of suggestions and comments on specific parts of the manuscript.

Lines 48-52: This sentence seems out of place and could be moved to the end of the introduction. The introduction is short and could be expanded to discuss the work that has already been done on these and/or similar samples (e.g. Li et al., 2019) and what

new insights can potentially be gained from boron isotopes. In addition, I think that the IBM subduction initiation model could be discussed in detail here.

This sentence has been moved to the end of the introduction (line 72-74).

We have re-organized our 'introduction section' in response to the comments from reviewers 1 and 2 to better present the 'big picture' and the importance of B isotope study. We also added more information related to the published radiogenic isotope results (line 64-69), and about subduction initiation models for the IBM system (line 35-40).

Line 69: You refer to boron abundances in Fig. 2, however these are not presented in Fig. 2 so it is difficult to follow the logic of the discussion.

Re-wording (line 78-81):

While B concentration ranges in HSB and LSB overlap, clear distinctions are evident with respect to MgO, TiO₂, and Ba, suggesting different magmatic sub-suites with distinct B abundances (Figure 2).

Line 70: 'clear correlations with MgO and TiO₂' Describe how the B/Nb correlates with these?

This sentence has been deleted in the revision.

Line 78: Are the IBM arc volcanics mentioned here also plotted in Fig. 2? The volcanics currently in Fig. 2 seem to range from ~+4‰ to +12‰. Please clarify.

The Mariana arc and Izu arc tephra range in $\delta^{11}\text{B}$ from ~+3‰ to +12‰ (line 92).

Line 97: More information on Hf isotope systematics is needed here to explain to the reader what we can learn from Fig. 3.

We have added this information (line 132-134):

Li et al. (2019) proposed amphibolite facies melting of subducting ocean crust to explain decreases in ϵ_{Hf} and enrichments of Zr and Hf in boninites, as the ϵ_{Hf} is

positively correlated with factors indicating the degree of Hf enrichment (e.g., Sm/Hf and Ti/Hf).

Lines 108 to 111: Confusing sentence which refers to Nb concentrations in highly depleted mantle whilst discussing Abyssal serpentinites. Please clarify. This discussion would seem better placed in previous paragraphs where Fig. 2 is described.

The concentration information is deleted and the sentence now describes Fig. 3 (line 106-108).

Line 115: Again, describe what the correlations are. Negative correlations?

Negative correlations. Modified.

Line 126: You describe mixing curves based on B/Nb concentrations, but it is only Ba/Nb concentrations that are plotted in Fig. 4. This makes interpretation of Fig. 4 very confusing and hard to follow the logic.

We changed this sentence to be more simply descriptive (line 159-160):

Mixing curves between the depleted mantle (DM) and the low F melts of oceanic gabbro are strongly hyperbolic (Figure 7a).

Line 130: Mixing lines between DM and the highest Ba/Nb LSBs are described. However, the mixing lines do not appear to go close to the highest Ba/Nb samples. Please explain further.

The intent of the mixing calculations is to generate a family of curves that can explain the observed B isotopic variability. We have modified this passage accordingly.

Line 135: Can you explain what is meant by 'a better match to the pattern'. In both cases, almost all of the data plots outside of these mixing trajectories!

As noted above, our discovery of clear B-Sr isotopic correlations improved our models for LSB source mixing, so our specific mixing calculations are different in the revised draft, as noted above. Also, in Figure 7b we compare our boninite data to calculated

mixing arrays between a moderately B-enriched mantle (M1; $\delta^{11}\text{B}=-4\%$, B/Nb is 8 times of the DM) and the hydrous melt. This model yields mixing trajectories between M1 and slab melt compositions that are a better match to the pattern of our data, arguing for the idea of some minor, fluid-related enrichment of the LSB source mantle before the addition of slab melts. All of this information is in the revised text.

Line 172: An alternative explanation is that the sampling of cores only represents a small snapshot of the total range of compositions in HSB lavas.

Given that our boninite samples represent the sum of two different sample sets, both of which were selected to represent the observed stratigraphic and chemostratigraphic diversity of the two boninite holes, it is unlikely that a large degree of geochemical heterogeneity is as yet undiscovered in the HSB, as these samples are comparatively much more uniform in their overall compositions than the LSB. We think it is more likely that the limited HSB range in B isotopes reflects an extremely depleted mantle source overprinted by late-stage hydrous slab fluids, leading to relatively uniform B isotope compositions. Both B/Nb and $\delta^{11}\text{B}$ in the HSB correlate with the Nd isotopes (e.g., Figure 8), which means both B and Nd were mobilized during late stage dehydration. The lowest ϵNd for an HSB in this study is lower than the published range in Li et al. (2019 EPSL). Based on B-Nd isotopes, we would expect no large B isotopic variations in the HSB.

Line 189-214: It is not apparent that any of these conclusions were drawn from the B isotope data presented in this manuscript. Currently, it appears like the boron story was fitted to a previously established subduction initiation model. While I agree that the new data is exciting and a worthwhile contribution, the authors need to clearly state what NEW evidence boron isotopes bring to the subduction initiation model. I would consider moving most of this discussion to the introduction to explain what we currently know about subduction initiation, before going on to present the new data. I believe this would provide a logical flow to the manuscript and highlight the importance of the new data.

As noted above in the ‘**Overall response to reviewers**’, the revision process has clarified the unique constraints provided by the B isotopes on subduction initiation in the Izu-Bonin system. B-Sr isotopic constraints demonstrate that neither sediments nor the upper oceanic crust contributes to the subduction input to the LSB. The amphibolite facies crustal melting suggested by the radiogenic isotopes are shown by B isotope systematics to arise from lower crustal gabbros. These observations indicate that upper ocean crustal materials must be accreted in the earliest stages of subduction, arguing for a different physical model for subduction initiation than that proposed for the IBM system by Stern and Bloomer (1992). Slab signatures in the HSB require fluid inputs from both sediment and upper oceanic crust, requiring a change in subduction geometry and dynamics and a cooler downgoing plate in the latest stages of boninite magmatism. We have modified Figure 9, our physical model for the process, in the revision.

Figures:

Fig. 2: Uncertainties on the data should be plotted and/or stated within the figure caption. In panel (a) there is a reduced dataset presented. Is there a reason for having B isotope data without B concentrations? Were these the leached samples? This needs explaining within the text or the methods section.

Fig. 3: Uncertainties on the data should be plotted and/or stated within the figure caption. We have added data uncertainties in the Figures. Error bars ($\pm 2\sigma$; 0.6‰) for $\delta^{11}\text{B}$ represent average whole chemical process external reproducibility based on duplicate analyses of different digestion of the reference materials and natural samples (Li et al., 2019 Chemical Geology). And we also added information why only part of the samples have the B concentration data in the ‘methods section’. Not all the boninites are suitable for B concentration study because of the alteration issue. Only the fresh, glassy samples were analyzed for B concentrations. We developed a leaching method to remove seawater alteration effects on the B isotopic signatures of modestly altered samples (see Li et al., 2019 Chemical Geology), to ensure that our B isotope results reflect the samples’ primary compositions. However, obtaining primary B concentration data for altered samples is not possible.

Fig. 4: Label panel (a) with 'undehydrated' and (b) with '60% B loss from slab' so that it is easier to follow the different plots. This figure needs a lot more information (ideally in a key) so it can be interpreted. What does the blue and red star represent? What are the three coloured shaded areas? Are the mixing lines you describe the thick grey lines? In the figure caption (line 472), fluid compositions for 5% dehydration are discussed with no reference to the Figure - is this the green arrow?

These diagrams (Figure 7) were extensively modified in our revision. 'Early dehydration' is not necessary as ocean crustal gabbros show a wide range of $\delta^{11}\text{B}$. In both (a) and (b) we recalculated the mixing curves for new endmembers, and we have updated this information in the text and figure captions.

Fig. 5: I am not sure it is necessary to have this cartoon. This same schematic is in most Izu-Bonin subduction initiation papers, so could simply be cited within the text. Currently, this figure gives the impression that the data is being fitted into an already established model.

See **reply to the main concern**. We think that as modified the physical model expressed in the cartoon will be very helpful to the readers. .

Figure S2: There is no key or explanation as to what the cross symbols represent.

We have fixed the legend as indicated.

Reviewer #2 (Remarks to the Author):

This is an interesting paper, with new data for B and B-isotopes in recently drilled rocks from the IBM forearc subduction initiation sequence. As it stands, the paper needs some work to be appreciated by a geoscience audience. Major flaws are: 1) the absence of context, i.e., how the paper addresses a compelling issue in Earth sciences, and 2) that results and implications are mired with some details, while other information is lacking. The paper should be revised to improve organization and clarity.

Please see the '**Overall response to reviewers**'. Our revisions provide clearer context

and relevance, and a better, clearer use of existing data to highlight the new geochemical and tectonic/dynamic constraints that the B isotopes in boninites provide on subduction initiation.

Here are some suggestions:

1) Introduce the big picture. What exactly has been discovered about subduction? My take is that there are distinct early and late fluids. Why is it important to know this? Coverage of the importance of the IBM system and the special behavior of B as an element is good, but an early introduction to the behavior of B isotopes is needed (i.e., what is covered on lines 236-251).

The introduction section has been significantly re-organized per the reviewer's suggestions.

2) Some details about what is selected for study is needed up front: a) Why are there two groups of LSB? b) What are their stratigraphic/petrographic differences or are they distinguished only by geochemistry? c) Why aren't HMA mentioned even though they form a large part of the section in Figure 1c? d) where are the FABs in the section?

We have added information aimed at addressing the reviewer's questions above.

a) and b) The LSB separate into high and low Ba subgroups based on correlations between Ba versus Nb, Hf, Zr and Th (Supplementary Fig. S1; line 81-82). Both LSB groups include fresh samples as well as samples that have been leached. Low Ba/Nb signatures appear to be characteristic of some of the LSBs.

c) The HMA are LSB differentiation products and are isotopically indistinguishable from them. We have added this information to the text (line 54-55) and to the caption for Figure 1.

d) FABs were recovered at Holes U1440 and U1441 (information added in the caption for Figure 1), and therefore are not shown in Figure 1c.

3) Why does the variation in Hf-isotopes prove that there is input of Hf from the slab?

Why couldn't mantle sources have variable Hf isotopic composition?

A detailed discussion of the variation in Hf isotopes can be found in Li et al. (2019 EPSL). We have provided more information as context in the 'introduction' (line 63-67) and 'discussion' (line 132-134) sections.

4) Label the difference in the models plotted in Figs. 4 a and b clearly. Why is the early extraction of B significant?

These diagrams and their caption are substantially revised, as noted above, addressing the reviewer's concern here.

For Nature Communications, the conclusions should be clear and significant. What are they and which are unique to this study? Two fluids involved in early slab dehydration? Early fluids different from the "typical" ones involved in arc magmatism? Slab cooling during early subduction? Summarize the evidence that supports each conclusion. Overall, it seems that the B-isotope data are not fully conclusive; instead they are being built into a self-consistent model shown in Figure 5. If this can be improved the paper will be more influential.

The key conclusions are much more clearly stated now. LSB B isotopes require melts of lower oceanic crust as the primary slab input. Sediments and the basaltic upper ocean crust are accreted during the initial stages of subduction. HSBs require fluid inputs from slab sediments and basalt, which means the geometry of subduction changes from LSB to HSB petrogenesis. We significantly modified our introduction, discussion, and implications sections in the revised manuscript.

Some details:

Fig. 4 caption and discussion: Is the process modelled: 1) mixing of components (fluids/melts) of serpentinite and AOC with the mantle, followed by melting, or 2) mixing of the components with melts of the mantle? It would be good to back this model up with an array of other trace elements.

Figure 7 (originally 4) has been substantially modified, as described above. This information has been added to the Figure caption and text.

Our choice of B, Ba, and Nb as the elements to focus on in our modeling is because they have similarly small partition coefficients during mantle melting, and distinctively different partitioning during fluid-rock interactions and amphibolite melt generation. Other elements do not give us the same leverage on the different melting and solid-fluid exchange events we are trying to model.

Fig. 2a – Why is there only one purple LSB dot?

Only a subset of our boninites were suitable for B concentration study due to seawater-related alteration, which is a particular issue for this element. As noted, we conducted leaching procedures on samples where alteration was a concern to ensure good, primary B isotope measurements. Only the very freshest samples could be analyzed for primary B concentrations. Among those LSB samples with low Ba/Nb, only one was fresh enough to give a reliable B abundance measurement. It was prepared and measured several times, and its Ba/Nb is the same within error.

Fig. 3b – Why are there no FABs on this plot?

The FABs are all significantly altered (Shervais et al., 2018 G3). Therefore, reliable for B isotope or abundance analyses are not possible.

Fig. 5a What is “IZA” on this diagram?

The model starts at a hypothetical transform fault at the Izanagi-Pacific plate boundary (Li et al., 2019). The IZA is Izanagi plate while the PAC is the Pacific plate. We have added this information in the revised manuscript.

Reviewer #3 (Remarks to the Author):

The authors provide new B and d_{11B} data on early arc boninites, and attempt to interpret those with respect to slab temperature. Unfortunately, the study is far from

being convincing. I find it difficult and confusing to read. There is not strong convincing case made how the data could help to infer slab T evolution. I made a couple of major comments, and a few comments linked to the text, which I hope help understanding why I found the manuscript lacking.

1. The use of terms ‘fluid’ vs. ‘melt’ is very confusing. For example, in the abstract it reads: “Correlations between boninite B isotopes, Ba/Nb, and ϵHf indicate that both dehydration and melting of subducted ocean crust under amphibolite facies conditions produced the slab fluid flux that triggered melting in the LSB mantle wedge”. (There are more examples in the manuscript). How can slab melting produce a fluid flux only? These term need to be distinguished stringently.

We agree with the reviewer and have extensively rewritten the manuscript to more clearly distinguish between slab fluids and slab melts.

Correlations between Ba/Nb, ϵHf_i , $^{87}\text{Sr}/^{86}\text{Sr}_i$ and $\delta^{11}\text{B}$ indicate that melting of subducted lower ocean crust under amphibolite facies conditions produced the slab flux that triggered melting in the LSB mantle wedge.

2. You argue that high d11B and low B points to an early ‘hot’ slab. How does that compare to the active ‘hot-slab’ Cascades arc where both d11B and B are low? You are that high B and low d11B indicates a cooler slab? How does this tally with the active ‘cool-slab’ Izu Bonin arc where both B and d11B are high?

I wonder if you recast your data from the boninite into a global compilation, how they would appear?

The revised manuscript is much clearer on these issues. The positive correlations between boron concentration and B isotopes in arcs globally arises from the mixing of two different slab-derived components, one a high B, high $\delta^{11}\text{B}$ component consistent with serpentinite and a second, low B, low $\delta^{11}\text{B}$ component consistent with a deep sediment melt (see Ryan and Chauvel, 2014). **Boninites are produced during subduction initiation, an environment that is totally different from a mature arc, which at least in its initial stages involves a hotter, shallower slab than any modern**

arc. The correlations between ϵ_{Hf} , Ba/Nb and $\delta^{11}\text{B}$ clearly show that B, Ba and Hf moved off the slab into the LSB mantle source. It clearly indicates a high temperature, amphibolite facies melt released from the slab triggered LSB generation, a melt which based on its B and Sr isotope signature had to come from Layer 3 gabbros. There are no clear Hf isotope correlations evident in the HSB, while by contrast there are clear B-Sr-Nd isotopic co-variations during HSB generation. These observations point to the dehydration of slab sediments and basalts during HSB generation.

The slab fluid inputs to both hot (e.g., Cascades) and cold (e.g., Izu-Bonin) arc lavas are necessarily “deep fluids” as compared to the fluids fluxing boninite sources. Early dehydration (before reaching the depth under the volcanic front) affect the slab B and B isotope composition significantly (Ryan and Chauvel, 2014). We compared our boninite data with that of the mature arc in the main text.

3. There is no quantitative attempt on characterized the alleged change in slab T. It's all based on relative observation and inferences. There are no constraints. What is the temperature significance of the B and B isotope signature? Your conclusions are based on comparative behavior and distribution of B and B isotope during subduction, and ideas that have been forward on how to explain diversity in different arcs (not shown). It is a long leap to recast these observations (not even quantitative models) into inferring slab T variation during early subduction.

Our reply to the main concern explains how and why our data can provide information about slab temperature variation during subduction initiation.

Trying to quantitatively calculate slab melting and/or dehydration temperatures from B and B isotope data would involve a myriad of assumptions: slab B and B isotope compositions, the slab mineral assemblages during melting versus dehydration, how the slab mineral assemblage might have changed from LSB to HSB genesis, the partitioning behavior of B in all the likely metamorphic minerals and others. The B and B isotope composition of ocean crust are highly variable, and the partition coefficients of B between amphibolite mineral assemblages and possible hydrous fluids or slab

melts have not been examined, and cannot be reliably constrained. While software packages such as Perplex or Thermocalc can generate likely mineral assemblages for particular protoliths under particular P-T conditions, these packages do not generate data for mobile phases, be they fluids or the kinds of melt phases that were likely liberated from the shallow Izu-Bonin slab during subduction initiation. Therefore, working from B abundances and B isotope results to calculate slab temperatures is impossible at present.

To try and constrain conditions on the slab, we have worked from what the geochemistry of Izu-Bonin boninites tells us. The Li et al. (2019 EPSL) paper on the radiogenic isotopes of the Expedition 352 volcanic suites infer that slab melting was under amphibolite facies conditions during LSB generation. Temperature constraints on this slab melting were estimated to be between 900°C and 950°C based on Li et al. (2019), and Pearce et al. (1992), consistent with existing experimental results on amphibolite facies basaltic systems. Based on our constraints from B-Sr isotopes, we know that the first melted slab component is the lower gabbroic crust, which shows little variability in Ba/Nb. So, in this study we use Ba/Nb to calculate the slab melting temperature. Batch melting of amphibolite at 1.4 GPa and < 975°C produces a slab melt with Ba/Nb values consistent with those measured in the LSB (yellow cycle; Figure 7). Our inferences here are similar to those drawn from Sm/Hf systematics in boninites by Pearce et al. (1992) and Li et al. (2019).

For the HSB, fluid rather than melt inputs point to slab temperatures lower than those required for amphibolite melting, and the compositional differences between the slab that is sampled to produce LSB (i.e., just lower-crustal gabbros) versus that which must be sampled to produce HSB (slab sediments and basalt) pushes the temperatures down further. Johnson and Plank, (1999) place the solidus for marine sediments at $775\pm 25^\circ\text{C}$ at 2 GPa and $810\pm 15^\circ\text{C}$ at 3 GPa. Based on these results, we estimate that the slab surface temperature had to be lower than 800°C at 2 GPa during HSB

generation. The only amphibole/fluid partitioning data available for Ba and Nb are from Brennan et al. (1995, 1998), based on a single experiment at 900°C and 2 GPa. Therefore a more precise calculation is impossible at present.

4. The time-progressive models is not really shown. The time interval studied with boninites (1.5 Myr duration) is short compared to the overall length of the arc. Why do you expect/explain drastic slab T change within this short period that produced slab-influenced boninite series? It would be more instructive to have added B and d11F from FAB and post-boninite series, and see where the alleged slab T changes correlate with chemistry.

Besides, do you have evidence that the HSB are really significantly younger than the LSB? Is there are time lapse between them relevant to the time scales needed to transmit slab processes to arcs?

Moving from two oceanic plates at the Earth's surface to a scenario in which one is subducting beneath the other will necessarily produce dramatic temperature changes in the downgoing plate on such a short timestep. The mantle upwelling associated with early plate foundering brings hot mantle into contact with the leading end of the downgoing plate, initially heating it up. Continued subduction ultimately brings that same mantle into extensive contact with cooler downgoing crust and lithosphere. The timing of LSB versus HSB magmatism constrained by the Ar-Ar and U-Pb results in Reagan et al. (2019), and supported by previous work on Chichijima boninites (which, per Pearce and Reagan 2019, are HSB), confirms that HSB are younger than LSB, and that the timeline for this magmatic transition is comparatively quick. While the timing of processes on the slab versus those during boninite magmatism might be sortable using U-series disequilibrium isotopes or like tools in zero-age rocks, such detail cannot be obtained on 50-52 Myr old submarine volcanic sequences.

As noted above, the slab temperature variations we discuss are inferred from relationships among ϵHf , ϵNd Ba/Nb and $\delta^{11}\text{B}$, specifically the differences in these relationships between the LSB and HSB.

Comments linked to the text.

L67: first you say that B in LSB and HSB are distinct by a factor of 3, and then you speak of overlap? Confusing.

The LSB are, on average, lower in boron than the HSB, though both grouping show considerable variability. While ranges for B in HSB and LSB have some abundance overlap, clear distinctions are evident with respect to MgO, TiO₂, and Ba, suggesting different magmatic sub-suites with distinct boron abundances. We've revised the text (line 78-80) to make these points clear.

L70-72: are you suggesting that the high MgO, low-TiO₂ (younger) HSB are produced by higher extent of melting despite their higher B/Nb relative to the LSB? Or why do you discuss the extent of melting here?

This was not our intent – in fact it's probable that melt fractions in the HSB are overall smaller. This sentence has been deleted.

L82: here you speak of slab melts - why then of fluids in the abstract?

Both melts and fluids are involved with the 352 boninites. The description in the abstract has been corrected.

L87: Reference missing after period.

Reference added.

L88: in Fig. 2, AOC is at +4 d11B, here you write d11B = -7.

$\delta^{11}\text{B} = -7\text{‰}$ is for fresh MORB or depleted mantle, per Marschall et al 2017. Average AOC values were estimated by Smith et al. (1995) at $\approx +3.7\text{‰}$, but with substantial variability, especially in the lower ocean crust.

L101: Here you do have slab melts. Does that means that the Ba and B are transported

in different slab fluids? Confusing.

Our point is that as Hf is immobile in hydrous fluids, correlations between these three tracers means B-Ba-Hf must be added to LSB sources together via (probably hydrous) melt inputs. We've corrected this statement to be clearer.

L111: You introduce here serpentinite a B source, which were never mentioned in the abstract.

L113: Does that mean, that gabbros could also be a B source? Or are they excluded?

L132: since the subducting slab loses B from the start, there is no wonder that B is lost from slab? How does the estimate of 60% B loss compare to other studies?

L129-139: confusing, incomprehensible -- how does the low $\delta^{11}\text{B}$ HSB fit into this modeling? What do you mean by 'our data' in L136? The LSB?

L140: I assume that this is the model to explain the LSB? Or both LSB and HSB?

As noted above, the B-Sr isotope correlations in Figure 4 point to that first slab melt component coming from lower crust gabbros. Serpentinite is precluded as the B source for LSB. Oceanic gabbros have a wide range of $\delta^{11}\text{B}$ (Figure 4), so 'early dehydration' is not necessary. This section of the text has been extensively modified, and the isotopic modeling is much simpler and clearer now.

L150: but the eHf of the HSB is very low. You explained earlier that this low eHf was a melt?

Per Li et al. (2019) the LSB mantle source is metasomatized by a fluid component to produce the HSB mantle source. It accounts the low and uniform eHf and variable eNd of the HSB. Our data are consistent with these contentions, and further constrain that the source of this fluid is subducted sediment and altered basalt.

L161-163: I think you need to be clear whether you have one or several components from slab.

For both LSB and HSB, B and B isotopes speak to the addition of a single slab input: a melt phase for the LSB, and a fluid phase for the HSB. The parts of the slab being

sampled by these mobile phases are different: for LSB it is only the lower crust, while for the HSB it is predominantly the sediments and upper crust. Volcanic arc lava B isotope systematics indicate at least two distinguishable slab-originating inputs, one consistent geochemically with serpentinite (which may come from the slab itself, or the subduction channel, or both), and one that is most consistent with a deep-sourced sediment melt (see Ryan and Chauvel, 2014).

L218-220: That you should show by a global compilation.

Several authors have produced global compilations (e.g. Ryan and Chauvel, 2014; Li et al., 2016; deHoog and Savov, 2018). The perspective of this paper is the difference between subduction initiation and the mature arc (IBM), specifically about how B abundance and isotopic systematics evolve from subduction initiation to a mature arc. We modified our text to be clearer on this point (line 241-270).

L232: Reference missing after period.

We accordingly added references.

Figure 4: why there are two panels? What is the difference?

We show two families of mixing models in the revision.

(a) depicts mixing between the depleted mantle (DM) and gabbroic slab melts, while (b) depicts a DM that has been metasomatized by an 'Early Fluid' (M1; $\delta^{11}\text{B} = -4\%$, B/Nb is 8 times of the DM) mixing with gabbroic slab melts. We have added this information to the Figure caption and text.

Figure S1: Legend/caption missing - what are the symbols?

Legend added [now changed to be B versus MgO, TiO₂ and Ba (Figure 2) in the text].

Figure S2: what are the crosses?

The crosses are FAB (now Figure 6 in the text), legend added.

Reviewers' comments:

Reviewer #1 (Remarks to the Author):

Review of 'Boron isotope evidence for upper crust accretion and a cooling slab during subduction initiation'

Since the original submission, the majority of the reviewer's comments have been suitably addressed and appropriate changes to the manuscript have been made. These changes have significantly improved the manuscript. In particular, the new introduction has improved the clarity of the paper.

There are, however, a number of paragraphs which do not appear to relate to the story and/or are difficult to follow. This may be a consequence of original text not being updated during the revision process. These minor changes should be address before publication.

L111: add 'substantially' before 'change'. The mixing with serpentine does appear to shift Ba/Nb a little.

L112-115: If gabbros have a highly variable B/Nb then why wouldn't inputs from gabbros have the potential to change the B/Nb and Ba/Nb? Did you mix up d11B, B/Ng and Ba/Nb here (see comments below)? If the B/Nb is highly variable, then it would be good to plot this range on Fig. 3.

L136-143 and Fig. 5: Here you suggest that the source of B and Ba in LSB is the downgoing Pacific Plate and that these elements are added by a 'non-fluid slab input mechanism' (these are labelled as Hydrous Melts on Fig. 5). However, in the following section (L176-201) you suggest that the slab input for HSB is a high temperature fluid phase. Why then do the HSB fall within the 'Hydrous Melts' trend in Fig.5a. It is difficult to follow the logic here when labelling of the figure is at odds with the discussion. This should be made clearer.

L275-286: I am not sure how this relates to the data you are presenting. It would seem more logical to end the manuscript with the major findings of the study rather than introducing background.

L157-174: This discussion on Fig.7 (old Fig. 4) has been greatly improved, but it is still hard to follow in places, examples:

- L171 'elevated B/Nb' – where is B/Nb plotted? Do you mean d11B?
- L171 'before Layer 3 gabbros' – this is the first time Layer 3 gabbros have been mentioned, what are they?

I also still believe (see original comments) that labelling panels (a) and (b) of Fig. 7 would be beneficial.

Fig.5 caption: add something along the lines of colours of dashed lines indicate different slab input events.

L90: two distinct, lower B/Nb LSB groups. Evidence for groups comes from Ba/Nb, so very confusing to describe discuss B/Nb and then go on to describe the d11B ranges. This apparent mix-up appears more times in the text. Check that B/Nb, Ba/Nb and d11B are correctly stated within the text.

Fig. 1 and response to comment regarding sampling: It would be good to plot the positions of the samples onto the Boninite Hole logs so that we can see that the chosen samples cover the full stratigraphic range and are representative.

Reviewer #2 (Remarks to the Author):

The manuscript is much more focused than the prior version. A major new finding is presented – that mixing of melts of oceanic gabbro to the mantle wedge produce the sources of the LSB and that overlying sediment and upper oceanic crust must be accreted rather than subducted, as shown in Fig. 9, for this to happen. This interpretation is reasonably well supported by Sr-isotope data, in addition to other elemental and isotopic evidence. Evidence from B and B-isotopes mainly precludes the involvement of serpentinite but is consistent with input from gabbro melts. Melting of gabbro is evidence for initial high temperatures in the downgoing slab during subduction and cooler temperatures are inferred later when HSB are formed with hydrous fluids (not melt) from sediment and oceanic crust entering the mantle wedge. Overall, it is an interesting and reasonably well-supported story, worthy of publication in Nature Communications. However, it is important to note that the interpretation does not rest exclusively or even largely on the B and B-isotope results.

Some other issues:

1) There is an overriding problem with the definition of the components added to the mantle to form appropriate sources for the LSB and HSB. Other reviewers commented on this. I strongly recommend that the term “hydrous fluid” is used throughout for non-melt components. Sometimes just “fluids” is used, and this is ambiguous; melt is fluid.

2) Although the involvement of serpentinite and hydrous fluids from serpentinite is ruled out, the author should still explain why this is so. I was a bit confused about the kind of serpentinite they are referring to. Serpentinite that forms in the lower oceanic crust (abyssal serpentinites discussed on line 107-111 and plotted in Fig. 3) would be likely to dewater if the overlying gabbro is hot enough to melt – why does this not occur? In contrast, the serpentinite that forms in the mantle wedge as the result of interaction with hydrous fluids from the slab may not have yet formed at the time that LSBs form. The authors should explain these two distinct serpentinite sources of B and include them in their model.

3) Among the figures, I think that Fig. 5 does not add much to the paper, whereas Fig. 6 is important.

Details:

Line 157-174: There is not much improvement in the fit to the data between the mixing models shown in Figs. 7a and 7b. The data fall within the range of amphibolite melts.

In both cases, it does look like the HSB source has incorporated some amphibolite melt and diverges from the LSB field toward the AOC + sediment mixture. If that is the case, the original mantle composition for the HSB should be mentioned in the explanation of Fig. 9.

Line 179 – Why does the fluid phase have to be high temperature? This is the stage that the slab is cooling, right?

Line 196 – 198: The geochemical evidence suggests that serpentinite dewatering does not affect the HSB, but the serpentinite must be there, below the gabbro layers, so what happens to it? Considering this might lead to another constraint on temperature.

Reviewer #3 (Remarks to the Author):

The authors provide an interesting data set on early-arc boninitic rocks. However, despite the revision, the authors failed in providing a solid and coherent interpretation on what these data could mean. There is much speculation, and some strange data interpretation. For example, the evidence for the gabbroic source does not pass scrutiny at all - and this is key to the paper. The component of accretion and subduction erosion this model requires is a complete speculation. Key aspects of subduction-related element recycling seem not to be understood. The manuscript remains difficult to read and is not always clearly written. The increased length has not added to clarity, but increased the complexity.

Reference style: citation in the text cannot be linked by number to the references list. Figures should be numbered for review.

Comments linked to the text:

L22: What are 'phases evolved off subducting plates' please specify

L25: In L22 you talk of phases (which I assume are fluid, partial melts etc), but now you mention sources

L30: accretion of Layer 1+2 of the Pacific plate - what is the evidence? Or is this a speculation of how subduction erosion vs. acc

L32: So here you are suggesting that two parameters change within 1 my: (i) subduction of Layer 1+2 and (ii) change to fluid-dominated material transfer from slab

L35-75: This is a very broad introduction. It would be better if you outlined very specifically which kind of problem was addressed with B and d11B (e.g. understanding melts vs fluid dominated material transfer and temporal change).

L78: Something needs to be said in the text why B and d11B are not affected by seawater alteration.

L114-115: In order to affect d11B with B from a given source, there needs to be a B flux from that source. So how can B/Nb NOT be affected? This does not make sense.

L120-121: The significant observation is that the boninites still have elevated 87/86 relative to mantle. Probably, 87/86 of the boninites is also similar to the Izu B arc front magmas.

L123-127: This kind of source identification (by comparison) does not work in arcs and their slab sources, especially in such fluid mobile elements. The identification of the gabbro source for the LSB is not substantiated (nor the distinction to the source of the HSB).

L136: if you deleted the red arrows on Fig 5b, there is no longer a correlation. By adding the arrows, you force a trends (the one you wish to see).

L138 (around): you never explain that the difference in eHf is as much a consequence of mantle wedge heterogeneity as of the slab addition. This is extremely confusing. Especially since the cause of the eHf (LSB) is related to slab melts, while the even lower eHf (HSB) is not related to slab melt.

L159: here you melt the gabbro with a 5% melt. This must add B to the wedge. How can then the B/Nb of the arc magmas stay low? (see above).

L205/206: You cannot say the B/d11B shows the thermal slab evolution - you have merely indication that it may be so provide your models hold that are derived from other models. This is very speculative.

Also, how does the hydrous LSB component in Fig 5 fit in?

L213: unfortunately, you never detail what the indication for sediment input are. This should be summarized somewhere in the paper. Best in the introduction.

L232-253: It should be critical to given the age data here, especially as it is mentioned early on that the extrusion of the boninites lasted no longer than 1 myr

L283-284: Where are you locating the subduction channel? In the forearc mantle? This is an unexpected place.

DETAILED RESPONSES TO THE REVIEWERS

Overall response to reviewers:

The major criticisms from three referees can be categorized as follows: (a) A less than clear presentation, with some paragraphs that do not appear to relate to the story and/or are difficult to follow; (b) Arguments for accretion and subduction erosion that are claimed to be complete speculation; (c) Problems with subduction-related element recycling and constraints on gabbros are needed to be further enhanced; (d) questions related to serpentinite.

We accept these criticisms and address them in the text and below in detail. Responses to other specific comments can be found in Responses to individual reviewers.

(a) Problem with presentation

Toward generating a clearer presentation, we have extensively re-organized the Results and Discussion, clearly identifying them as such, and rewritten parts of the manuscript. In particular, the following major changes are made:

1. In the Introduction, it is clearly stated that this study will address the questions as to how the slab begins to devolatilize when subduction starts, and how and why slab inputs change as igneous activity evolves. Of particular importance is constraining the slab-to-mantle material exchanges (e.g., melts versus fluids as the fluxing agent for LSB and HSB magmatism, respectively) during subduction initiation, and the governing mechanisms behind them.

2. We merged the original Results and Discussion sections to one 'Results and Discussion' section so that subheadings are allowed. We've added a number of descriptive sub-headings to make it easier for readers to follow our arguments and reasoning.

As re-organized, the revised MS first presents our new data, followed by our efforts to identify the possible slab inputs, which are then assessed in detail for the LSB and HSB. The revised text now highlights the peculiarity of the Expedition 352 boninites

and the evidence for a rapid change in slab inputs in association with slab cooling. Finally we use these new findings to refine our understanding of slab-mantle interactions during subduction initiation as compared to mature arcs, and to describe the likely physical phenomena of subduction initiation, as characterized by our new results.

3. As we re-organized the text, we removed extraneous sections of text, and we condensed our figures down to seven.

We hope these changes make the revised text easier to follow.

(b) Speculation on crustal accretion

Our new B abundance and isotopic data demonstrate that the first likely fluxing slab inputs to the boninite source mantle come from oceanic gabbros, located relatively deep in the downgoing plate. No sensible mixture of upper crustal materials (basalts, sediments) can explain these B signatures or their correlations with Sr isotopes and other elemental and isotopic tracers, even if one tries to special-plead for some involvement of serpentinite (which is likely precluded as a major contributor in this setting for straightforward thermal reasons- see comments below). The lack of a sedimentary signature in the LSB is supported by the broader radiogenic isotopic study of Li et al (2019: EPSL), and Pearce and others have long contended that boninite slab inputs have to include some kind of melt phase in order to mobilize species such as Zr and Hf. Our new contribution here is that based on the overall higher values AND on the variability of $\delta^{11}\text{B}$ seen in the LSB, the only reasonable source for this slab melt is Layer 3 in the oceanic crust, which shows comparable variability in $\delta^{11}\text{B}$ at all sites where it's been examined; is on average higher in $\delta^{11}\text{B}$ than altered basalts; and is comparably lower and more uniform in $^{87}\text{Sr}/^{86}\text{Sr}$ than altered basalts or sediments. Both the $\delta^{11}\text{B}$ values AND the shape of the LSB $\delta^{11}\text{B} - ^{87}\text{Sr}/^{86}\text{Sr}$ data array are suggestive of inputs from oceanic gabbros such as would be deep within the Pacific plate crust.

If one accepts, as our data indicate, that neither altered oceanic basalts nor sediments are contributing to the slab inputs that flux LSB melting, the first order inference is that these materials are not getting subducted to depths where they could

metamorphose and liberate a slab melt or fluid. Thus have we suggested in the paper that these materials are initially accreted before they later begin to subduct, dehydrate, and flux the melting of the HSB.

Given our statement above, we disagree with the reviewer claim that accretion of the uppermost portion of the downgoing Pacific plate is a matter of pure speculation – properly, it's an inference arising from our geochemical findings. However, it is not the key point of this paper, which is about characterizing the rapid evolution of slab outfluxes as subduction starts. Accordingly, we have changed the title to the more descriptive '**Boron isotopes in boninites document rapid changes in slab inputs during subduction initiation**'. We now discuss accretion only as an earliest-stage phenomenon in our description of the dynamic model for Izu-Bonin subduction initiation.

(c) Involvement of gabbros in the LSB

As noted above, we think that our evidence for the involvement of gabbros in the LSB is solid. We think the ambiguity arises from our presentation of these findings, so we have extensively rewritten this section. As well, we have performed new calculations involving Sr/Nd ratios, which show that the high Sr/Nd in the LSB (~ 80; Inset in Fig. 4c) can be plausibly accounted for via by melting of the Layer 3 amphibolitized gabbro source (~ 53; Holm, 2002), but far less so via melting of amphibolitized Layer 2A basalts, given their comparatively low Sr/Nd. We have added this information to Fig. 4c and the text describing it.

We want to emphasize the importance of understanding the comparative mobility of B and other incompatible elements in assessing slab inputs. For instance, the argument that the slab input to the LSB source is likely via a melt is based on the coherent behavior for B, Ba, Nd, Sr and Hf during the formation of the LSB. Since Hf is immobile in hydrous fluids, the correlations between these tracers mean that B, Ba, Sr, Nd and Hf must have been added to LSB sources through the same, non-fluid slab input mechanism. By contrast, Hf in the HSB seems not behave coherently with boron and other lithophile elements, given low and uniform ϵ_{Hf} in the HSB, irrespective of

their higher and more variable B/Nb, Ba/Nb, and ϵNd_i , consistent with a fluid phase adding boron and other species, while the Hf signature arises from the melt-modified, LSB residual mantle.

We hope our changes have clarified and strengthened these major points of our paper.

(d) The serpentinite issue.

Serpentinite is not the focus of this study, in particular because given the likely hot thermal structure of subduction initiation, the involvement of serpentinite or hydrous fluids from serpentinite can be ruled out. Nevertheless we agree with the referee that the implications of the absence of serpentinite contributions to the Expedition 352 boninites is important and should be discussed. In response to the comments of reviewer #2, we have added statements focusing on the serpentinite question in the revised text, documenting the unlikelihood of it playing an important role in boninite petrogenesis.

The referee also raised the question as to whether abyssal serpentinites formed in the lithospheric mantle immediately below the lower oceanic crust could melt and potentially contribute to the slab inputs to LSB sources. A significant problem such a model faces is the thermal stability limitations on serpentine minerals, which break down at temperatures $\leq 700^\circ\text{C}$ at the kinds of pressures required to produce amphibolite facies metamorphism of slab basalts and gabbros. Per the available experimental constraints, melts of slab amphibolites would occur at temperatures approaching 1000°C , so the likelihood of serpentinite preservation immediately beneath this horizon that might be available for melting is poor. It is also worth noting that any serpentinite that had formed beneath Layer 3 gabbros would likely be quite low in B, as most of the B in seawater-derived metamorphic fluids percolating through the oceanic crust would be very effectively scavenged by the gabbros, as documented by McCaig et al. (2018). As such, this kind of abyssal serpentinite likely cannot contribute much to the B enrichment or $\delta^{11}\text{B}$ signatures of arc volcanics (McCaig et al., 2018). (see **Line 197-206**)

Reviewer #1: (Remarks to the Author):

Since the original submission, the majority of the reviewer's comments have been suitably addressed and appropriate changes to the manuscript have been made. These changes have significantly improved the manuscript. In particular, the new introduction has improved the clarity of the paper. There are, however, a number of paragraphs which do not appear to relate to the story and/or are difficult to follow. This may be a consequence of original text not being updated during the revision process. These minor changes should be address before publication.

Please see the overall response to reviewers.

L111: add 'substantially' before 'change'. The mixing with serpentine does appear to shift Ba/Nb a little.

We have added 'substantially' before 'change' in this statement, now in **Line 122**.

L112-115: If gabbros have a highly variable B/Nb then why wouldn't inputs from gabbros have the potential to change the B/Nb and Ba/Nb? Did you mix up d11B, B/Nb and Ba/Nb here (see comments below)? If the B/Nb is highly variable, then it would be good to plot this range on Fig. 3.

We intended to say that additions of Layer 3 gabbro to Layer 2A basalt would elevate $\delta^{11}\text{B}$ significantly without large changes in B/Nb and Ba/Nb. This section has been extensively rewritten (see **Lines 125-134**).

Bulk mixing curves between average serpentinite, Layer 3 gabbro, marine sediment and Layer 2A basalt are shown on the diagrams. The ranges for B/Nb and Ba/Nb for gabbros have been added in Figures 3a and 3b of the revision.

L136-143 and Fig. 5: Here you suggest that the source of B and Ba in LSB is the downgoing Pacific Plate and that these elements are added by a 'non-fluid slab input mechanism' (these are labelled as Hydrous Melts on Fig. 5). However, in the following section (L176-201) you suggest that the slab input for HSB is a high temperature fluid

phase. Why then do the HSB fall within the 'Hydrous Melts' trend in Fig.5a. It is difficult to follow the logic here when labelling of the figure is at odds with the discussion. This should be made clearer.

We have corrected the original Figure 5 (Figure 4 now) and labeled the 'Hydrous melts' and 'Hydrous fluids' trends more clearly. Both higher and lower T hydrous melts are suggested, constrained primarily by Ba/Nb systematics, which show a strong temperature dependency during amphibolite melting. While the HSB appear to fall along an extension of the 'Hydrous melts' LSB trend in terms of Ba/Nb, this is not the case in the other Figure 4 panels, in particular as regards their $\delta^{11}\text{B}$ signatures. The low and very uniform HSB ϵHf_i is in contrast with its variability in the LSB, where its covariation with B/Nb, Ba/Nb and $\delta^{11}\text{B}$ points to all of these species being added to boninite sources via a melt phase. The decoupling of ϵHf_i from the mobile elements is indicative of a hydrous fluid input, as a melt is necessary to mobilize Hf. Nevertheless, the progressive ϵHf_i evolution of FAB, LSB and HSB suggest its inheritance from the mantle source: that is, the HSB are derived from a slab melt-modified mantle residual to LSB generation, that was subsequently fluxed by a hydrous fluid.

L275-286: I am not sure how this relates to the data you are presenting. It would seem more logical to end the manuscript with the major findings of the study rather than introducing background.

Please see the overall response to reviewers. This section is now incorporated into the new section titled: **A refined model for slab-mantle interaction during subduction initiation** (see Line 309-324).

L157-174: This discussion on Fig.7 (old Fig. 4) has been greatly improved, but it is still hard to follow in places, examples:

- L171 'elevated B/Nb' – where is B/Nb plotted? Do you mean $d^{11}\text{B}$?

Mixing on the plot of Ba/Nb and $\delta^{11}\text{B}$ is controlled by the B, Ba and Nb abundance in the mixing endmembers, and the effects of B abundance differences in the endmembers

are implicit in the $\delta^{11}\text{B}$ variation. We already rephrased the words in the corresponding place:

‘Our modeling results thus suggest the possibility that a small slab fluid input that elevated B with respect to Ba and Nb occurred before Layer 3 gabbros metamorphosed and melted to modify the LSB source mantle.’ (L192-195)

- L171 ‘before Layer 3 gabbros’ – this is the first time Layer 3 gabbros have been mentioned, what are they?

Corrected. The Layer 3 is first mentioned just after altered ocean crustal gabbroic rocks in the revision (L99-100).

I also still believe (see original comments) that labelling panels (a) and (b) of Fig. 7 would be beneficial.

We have labeled Panels (a) and (b) in the revision. (a) LSB: DM + Gabbro Melting; (b) LSB: M1 + Gabbro Melting. The M1 is now explained in the text and the Figure caption.

Fig.5 caption: add something along the lines of colours of dashed lines indicate different slab input events.

We have modified the original Figure 5 (now Figure 4). The ‘Hydrous melts’ and ‘Hydrous fluids’ trends are more clearly labeled.

L90: two distinct, lower B/Nb LSB groups. Evidence for groups comes from Ba/Nb, so very confusing to describe discuss B/Nb and then go on to describe the $\delta^{11}\text{B}$ ranges. This apparent mix-up appears more times in the text. Check that B/Nb, Ba/Nb and $\delta^{11}\text{B}$ are correctly stated within the text.

We have cleaned up the text in this regard. We first show the B/Nb, Ba/Nb and $\delta^{11}\text{B}$ differences between the LSB and HSB and then introduce the Ba/Nb and $\delta^{11}\text{B}$ differences among the LSB subgroups to avoid confusion. (L87-91).

Fig. 1 and response to comment regarding sampling: It would be good to plot the

positions of the samples onto the Boninite Hole logs so that we can see that the chosen samples cover the full stratigraphic range and are representative.

We have added sample positions in Fig. 1-c as well as can be done given the specifics of their sampling.

Reviewer #2 (Remarks to the Author):

The manuscript is much more focused than the prior version. A major new finding is presented – that mixing of melts of oceanic gabbro to the mantle wedge produce the sources of the LSB and that overlying sediment and upper oceanic crust must be accreted rather than subducted, as shown in Fig. 9, for this to happen. This interpretation is reasonably well supported by Sr-isotope data, in addition to other elemental and isotopic evidence. Evidence from B and B-isotopes mainly precludes the involvement of serpentinite but is consistent with input from gabbro melts. Melting of gabbro is evidence for initial high temperatures in the downgoing slab during subduction and cooler temperatures are inferred later when HSB are formed with hydrous fluids (not melt) from sediment and oceanic crust entering the mantle wedge. Overall, it is an interesting and reasonably well-supported story, worthy of publication in Nature Communications. However, it is important to note that the interpretation does not rest exclusively or even largely on the B and B-isotope results.

(1) We have changed the title, as noted above.

(2) We disagree with the contention that these results do not significantly rest on the B isotopes. As with any new elemental or isotopic tracer, it is through comparisons to other existing elemental and isotopic data, building on past interpretations, that new insights are gained. Specifically, absent the distinctions in the B abundance and isotopic signatures and in the data patterns among the boninite subtypes and among the potential slab components, we could not have resolved either the differing sources of the slab contribution (gabbros versus basalt/sediments) or their different physical characteristics (melt vs fluid). We try to emphasize these points in both the Introduction and in the main text.

Some other issues:

1) There is an overriding problem with the definition of the components added to the mantle to form appropriate sources for the LSB and HSB. Other reviewers commented on this. I strongly recommend that the term “hydrous fluid” is used throughout for non-melt components. Sometimes just “fluids” is used, and this is ambiguous; melt is fluid. Agreed. ‘Hydrous melts’ and ‘Hydrous fluids’ are now used in the whole text to distinguish the different types of subduction inputs.

2) Although the involvement of serpentinite and hydrous fluids from serpentinite is ruled out, the author should still explain why this is so. I was a bit confused about the kind of serpentinite they are referring to. Serpentinite that forms in the lower oceanic crust (abyssal serpentinites discussed on line 107-111 and plotted in Fig. 3) would be likely to dewater if the overlying gabbro is hot enough to melt – why does this not occur? In contrast, the serpentinite that forms in the mantle wedge as the result of interaction with hydrous fluids from the slab may not have yet formed at the time that LSBs form. The authors should explain these two distinct serpentinite sources of B and include them in their model.

We agree with the reviewer’s arguments regarding the likely viability of serpentinite as a component in boninite systems: abyssal serpentinites should break down well before any kind of amphibolite melting could occur, and the generation of serpentinite via hydration of mantle wedge materials in the subduction channel is highly unlikely under the temperature conditions of subduction initiation. We have expanded our discussion related to the different kinds of serpentinites to try and make this clearer. Both abyssal serpentinites (on the sea floor) and forearc mantle serpentinites are enriched in H₂O and B (e.g., Savov et al., 2005, 2007; Boschi et al., 2013), and are touted as possible sources for boron and high $\delta^{11}\text{B}$ signatures in mature arc lavas (Benton et al, 2001; Straub and Layne 2002; Savov et al., 2005, 2007; Tonarini et al 2007). However, the relatively low B concentrations and consistently low $^{87}\text{Sr}/^{86}\text{Sr}$ in the LSB argues against either kind of serpentinite as a significant source component (L122-124; L197-206; L253-258).

As well, if serpentinite forms in/below the lower oceanic crust, it would likely be poor in B as a result of altered gabbro sequestering most of the boron out of any circulating metasomatic fluids. McCaig et al. (2018, Nature Communications) in a B and B isotope study of the Hess Deep gabbros (IODP Hole U1415), demonstrated that one-pass hydration of the upper mantle, such as has been as proposed for near trench serpentinitization, will not get significant B into the hydrated slab mantle. They thus conclude that hydrated mantle in subducting slabs will only rarely contribute to boron enrichment in arc volcanics, or to deep mantle boron recycling. We make note of these new findings in the text (L197-206).

We agree with the reviewer that the serpentinite in the forearc mantle likely could not form at the time of LSB generation. At temperatures higher than 700°C, serpentine minerals are not stable. During LSB generation, the slab (900-1000°C) and nascent mantle wedge are both at much higher temperatures, which we note in the text (L255-258).

3) Among the figures, I think that Fig. 5 does not add much to the paper, whereas Fig. 6 is important.

We have merged the original Figures 5 and 6 and added a plot of Sr/Nd versus ϵ_{Hf} to the new Figure 4 in our revision. The Ba/Nb, Nb/Ti and Sr/Nd versus ϵ_{Hf} diagrams document that the subduction inputs to the LSB source are hydrous melts of the slab crust. The Sr/Nd versus ϵ_{Hf} and $^{87}\text{Sr}/^{86}\text{Sr}$ versus $\delta^{11}\text{B}$ diagrams serve to identify the melted crust component as Layer 3 gabbros, while the Ba/Nb versus $\delta^{11}\text{B}$ diagram explain the LSB petrogenesis: almost dry DM + Gabbro melting. These diagrams together show that B, Ba, Sr, Nb, and Hf must be added to LSB sources by the same, slab melt input mechanism.

Details:

Line 157-174: There is not much improvement in the fit to the data between the mixing models shown in Figs. 7a and 7b. The data fall within the range of amphibolite melts.

In both cases, it does look like the HSB source has incorporated some amphibolite melt and diverges from the LSB field toward the AOC + sediment mixture. If that is the case, the original mantle composition for the HSB should be mentioned in the explanation of Fig. 9.

The pattern of the array of melt models in Figure 5b (original 7b) is a better fit to the data for M1+Gabbro melt mixing and M1+measured sample mixing. The mixing lines in Figure 5a (with DM at $\delta^{11}\text{B}$ of -7‰ and very low B contents as a starting point) are strongly hyperbolic and do not fit our data as well. Our thinking is that some kind of pre-contaminated mantle, residual to the generation of the FABs and reflecting some kind of early slab influence, as seen in the later-stage FABs (e.g., Coulthard, 2018) is the probable source of the LSB

We have added information about the HSB original mantle composition in the text (L212-222) and in the caption to Figure 4.

Line 179 – Why does the fluid phase have to be high temperature? This is the stage that the slab is cooling, right?

The reviewer is correct that the only constraint we have on the temperature of this fluid is that it's not a melt of sediment or basalt. So we've not tried to define the fluid temperature in our revision to avoid any confusion. Nd is clearly mobile (Figure 6) in the 'Hydrous fluids', but Hf is immobile (Figure 4). This 'Hydrous fluid' is similar as the deep subduction input endmember for mature arc mantle (Pearce et al., 2007 EPSL; Woodhead et al., 2012 Geology). That is why we called it 'high temperature' in the last version of submission.

Line 196 – 198: The geochemical evidence suggests that serpentinite dewatering does not affect the HSB, but the serpentinite must be there, below the gabbro layers, so what happens to it? Considering this might lead to another constraint on temperature.

Please see reply to the main issues (2). If temperatures are such that the gabbros are melting as amphibolites at 975-1015°C, then any serpentine minerals that might underlie the gabbros would have already broken down. And serpentinite that forms

in/below the lower oceanic crust likely does not contribute significantly to the boron budget in this case (McCaig et al., 2018, Nature Communications).

Reviewer #3 (Remarks to the Author):

The authors provide an interesting data set on early-arc boninitic rocks. However, despite the revision, the authors failed in providing a solid and coherent interpretation on what these data could mean. There is much speculation, and some strange data interpretation. For example, the evidence for the gabbroic source does not pass scrutiny at all - and this is key to the paper. The component of accretion and subduction erosion this model requires is a complete speculation. Key aspects of subduction-related element recycling seem not to be understood. The manuscript remains difficult to read and is not always clearly written. The increased length has not added to clarity, but increased the complexity.

We disagree with this reviewer in that we think our evidence for the Layer 3 gabbro sourcing of the slab melts that flux LSB magmatism does withstand scrutiny: the combined B-Sr-Ba-Nb abundance and Hf isotopic systematics provide a solid case for melts of an oceanic gabbroic source generated under amphibolite facies conditions as the most likely fluxing agent for the LSB. We also disagree with the reviewer's claim that the initial accretion of Pacific Plate basalts and sediments is "complete speculation"; the idea of initial accretion is a reasoned inference based on the lack of evidence for a slab signature in the LSB that can be clearly derived from altered basalts or sediments. However, as we've noted extensively above, we accept that the clarity of our presentation made it difficult for readers to follow our reasoning. We have thus extensively rewritten significant parts of the manuscript, with the following major changes:

(1) The title is changed as noted above.

(2) In the Introduction, we now clearly state that this study will address questions as to how the slab begins to devolatilize when subduction starts, and how and why slab inputs appear to change as igneous activity evolves. Of particular importance is

constraining the slab-to-mantle material exchanges (e.g., melt versus fluid fluxed the LSB and HSB mantle source) during subduction initiation and the governing mechanisms behind this.

(3) In the Discussion, we re-organized the text under a series of subheadings for ease of reading:

a) Possible slab components involved in the source of Izu-Bonin boninite

b) Lower crust hydrous melt inputs to the LSB mantle source.

c) Upper crust hydrous fluid inputs to the HSB mantle source

d) Slab-derived melt vs. fluid additions to Izu-Bonin boninites differ from what is seen in arcs.

e) Rapid change in slab flux from LSB to HSB reflects a cooling slab.

f) A refined geodynamic model for slab-mantle interaction during subduction initiation.

We merged the original Results and Discussion sections to one 'Results and Discussion' section so that subheadings are allowed.

(4) Our evidence for the gabbroic source arises from the similarity in both isotopic signatures and data patterns for the boron and strontium isotopes in the LSB and in published data for oceanic gabbros, and from modeling that shows that a high Sr/Nd component similar to oceanic gabbros is required to produce the high Sr/Nd ratios of the LSB. Importantly, the identification of a gabbroic melt input to the LSB mantle source is based on the observed systematics of both fluid-mobile and fluid-immobile elements, specifically the coherent behavior for B, Ba, Nb, Sr and Hf, during the formation of the LSB. As Hf is immobile in hydrous fluids, the correlations between these tracers mean that B, Ba, Sr, Nb and Hf must have simultaneously been added to the LSB sources through the same, non-fluid slab input mechanism.

(5) As noted above, several figures were revised and merged, and confusing and/or irrelevant passages in the text were heavily revised or deleted.

Reference style: citation in the text cannot be linked by number to the references list.

Figures should be numbered for review.

The references have been linked by number and the Figures are now re-numbered and identified.

Comments linked to the text:

L22: What are 'phases evolved off subducting plates' please specify

The abstract has been re-written, and this phrase has been replaced with more concretely descriptive language (L22-24)

L25: In L22 you talk of phases (which I assume are fluid, partial melts etc), but now you mention sources

Replaced with more concrete and descriptive language (L22-24)

L30: accretion of Layer 1+2 of the Pacific plate - what is the evidence? Or is this a speculation of how subduction erosion vs. acc

The evidence supporting this inference is now clearly stated in the abstract.

L32: So here you are suggesting that two parameters change within 1 my: (i) subduction of Layer 1+2 and (ii) change to fluid-dominated material transfer from slab

Yes, the sources and phases both changed within ~1 Ma. This may be a special feature of subduction initiation.

L35-75: This is a very broad introduction. It would be better if you outlined very specifically which kind of problem was addressed with B and d11B (e.g. understanding melts vs fluid dominated material transfer and temporal change.

In the Introduction, it is now clearly stated that this study will address the questions as to how the slab begins to devolatilize when subduction starts, and how and why slab inputs change as igneous activity evolves. Of particular importance is to constrain the slab-to-mantle material exchanges (e.g., melt versus fluid fluxed the LSB and HSB mantle source) during subduction initiation and the governing mechanism behind

L78: Something needs to be said in the text why B and $\delta^{11}\text{B}$ are not affected by seawater alteration.

This information is laid out in detail in the Methods section. The samples include fresh boninites with unaltered glassy matrices that were selected specifically for this work (boninites can frequently be very fresh, even when old), and a subset of the large-volume 'POOL' samples examined for radiogenic isotopes (Li et al., 2019a), that have been subject to coordinated post-cruise measurement. HCl leaching procedures documented as effective in removing seawater boron contamination were applied to show that the B and $\delta^{11}\text{B}$ of the fresh samples were not affected by seawater alteration (Li et al., 2019b; Supplementary information). All the 'POOL' samples were subject to leaching to remove seawater effects, so we only report their measured $\delta^{11}\text{B}$ and not their boron contents (L341-344).

L114-115: In order to affect $\delta^{11}\text{B}$ with B from a given source, there needs to be a B flux from that source. So how can B/Nb NOT be affected? This does not make sense. We intended to say that additions of Layer 3 gabbro to Layer 2A basalt would elevate $\delta^{11}\text{B}$ significantly without large changes in B/Nb and Ba/Nb. This section has been extensively rewritten (see Lines 125-134). As well, given that the slab flux to the LSB source is a melt phase, both boron and niobium are being added.

L120-121: The significant observation is that the boninites still have elevated $^{87}\text{Sr}/^{86}\text{Sr}$ relative to mantle. Probably, $^{87}\text{Sr}/^{86}\text{Sr}$ of the boninites is also similar to the Izu B arc front magmas.

The LSB have modestly elevated $^{87}\text{Sr}/^{86}\text{Sr}$ that are similar to Izu-Bonin volcanic front lavas, but the HSB range to much higher values, suggesting differences in the sources of Sr between the two boninite types (Figure 3c). The systematics of B, Sr, Nd, and Hf are decoupled in mature subduction zones (Ryan and Chauvel, 2014; Pearce et al., 2005): the paths of these elements from the slab to the sub-arc mantle, in terms of the phases that transport them and the timing of their mobilization, are markedly different: Hf is not soluble in any kind of hydrous fluid phase, Nd is moderately soluble

in high T fluids, Sr is more soluble than Nd, while boron has much higher $D_{\text{fluid/solid}}$ values than either Sr or Nd, and it is strongly soluble and mobile at lower temperatures. B/Nb correlates strongly and positively with $\delta^{11}\text{B}$ in all arc lavas, while Ba/Nb correlates broadly with $\delta^{11}\text{B}$ along arc volcanic fronts, but much less well across arcs (e.g., Ryan et al., 1995; 1996). No clear correlations are evident between $\delta^{11}\text{B}$ and ϵHf in arcs, or for that matter between $\delta^{11}\text{B}$ and ϵNd . $\delta^{11}\text{B}$ has been found to correlate with $^{87}\text{Sr}/^{86}\text{Sr}$ and with Pb isotopes within arcs (Ishikawa and Nakamura 1994; Ishikawa and Tera 1997) but no straightforward global relationships among these isotopic systems in arc lavas are evident.

The fact that the LSB show correlated B, Ba, Sr, Nd, and Hf systematics means that all of these elements were transported in the same way from the slab into the LSB mantle source. This is something that absolutely does not occur in modern volcanic arcs. As such, we can use these elemental and isotopic relationships to constrain the source for boron. (L149-153).

L123-127: This kind of source identification (by comparison) does not work in arcs and their slab sources, especially in such fluid mobile elements. The identification of the gabbro source for the LSB is not substantiated (nor the distinction to the source of the HSB).

As noted above, the relationships among these tracers for the different boninite types from Expedition 352 (and for other Izu-Bonin boninites) are NOT arc-like, and we have added statements to the text to emphasize this (L248-250). As such we are able to use the first-order relationships among these different elemental and isotopic tracers to define the likely sources.

L136: if you deleted the red arrows on Fig 5b, there is no longer a correlation. By adding the arrows, you force a trends (the one you wish to see).

The arrows in Figure 5 relate back to the recognized isotopic relationships among the Expedition 352 rock types and the nature of this igneous setting, as constrained by the local geology: magmatism during subduction initiation begins as forearc basalts, and

then transitions into low-silica boninites followed by high silica boninites, both of which must be derived from a mantle source depleted by the previous basaltic magmatism. The radiogenic isotopes record a progressive change from Indian Ocean to Pacific Ocean mantle provenance as magmatic products change through this sequence, related to progressively greater inputs of Pacific-derived crustal and sedimentary materials (see Li et al 2019, Reagan et al 2017), pointing to boninite petrogenesis from the mantle residual to FAB melting. So, per their isotopic trajectory, the patterns in Figure 5 should show curvilinear arrays suggestive of a similar progressive change, from no slab signature in the FABs to some slab signature in the LSBs to the strongest slab effect in the HSB – and, if you take the arrows off of Figures 5b and 5c, they kind of do show that sort of array. However, for Ba/Nb and especially for $\delta^{11}\text{B}$ this is clearly NOT what we see, which necessarily means that the sourcing of the most strongly mobile elements in the boninites differ between the LSB and HSB, and that there are additional complexities related to the Ba/Nb systematics of the LSB.

The combined Ba/Nb, Nb/Ti and Sr/Nd versus ϵHf relationships (Figure 4) and the Ba/Nb versus $\delta^{11}\text{B}$ relations (Figure 5) together show that B, Ba, Sr, Nb, and Hf have to be added to LSB sources via the same slab input mechanism, which the extreme hydrous fluid immobility of Hf constrains to be a melt phase. The actual trends for progressive additions of a slab melt would necessarily be curved, but the real point of Figure 4 is to note the Hf isotope variability of the LSB, and how that variability correlates (or doesn't) with those of other key tracers. The insets to Figure 4a and 4c highlight some of the unexpected geochemical outcomes of melting amphibolitized gabbros, by way of offering an explanation for the Ba/Nb variation of the LSB, and their unusually high Sr/Nd.

L138 (around): you never explain that the difference in ϵHf is as much a consequence of mantle wedge heterogeneity as of the slab addition. This is extremely confusing. Especially since the cause of the ϵHf (LSB) is related to slab melts, while the even lower ϵHf (HSB) is not related to slab melt.

We don't talk about ϵHf mantle heterogeneity as a possible origin for the variation

observed in the ϵ_{Hf} of our samples because it's immaterial in this case, as other recent work on these and related samples has documented. Li et al (2019a; EPSL) demonstrated that the ϵ_{Hf} signatures of our boninites relate to amphibolite facies melting of subducting ocean crust (L144-146). The observed ϵ_{Hf} versus Sm/Zr and Ti/Zr correlations are consistent with the high-T high-P experimental results (Tiepolo et al., 2007) and early interpretation for Zr-Hf anomalies for the boninites and TTG (Pearce et al., 1992; Foley et al., 2002 Nature). As well, our samples are part of a stratigraphic suite of volcanic rocks from four IODP holes drilled within 8 km of one another in the Izu-Bonin forearc, which show a regular variation in Hf isotopes, from FAB to HSB, that precludes any significant role for existent in-mantle ϵ_{Hf} heterogeneity. Li et al., (2019a; EPSL) showed that the lowest LSB ϵ_{Hf} values are comparable with the ϵ_{Hf} seen in the HSB (Fig. 7 of Li et al., 2019a). So, there are no ϵ_{Hf} changes from LSB to HSB that might justify another explanation.

L159: here you melt the gabbro with a 5% melt. This must add B to the wedge. How can then the B/Nb of the arc magmas stay low? (see above).

The reason that B/Nb stays low is that Layer 3 gabbro B contents are low as compared to average Layer 2A basalts, as has been seen in past studies on samples from several DSDP and ODP drilling sites (DSDP Holes 417A, 417D and 418A and ODP Hole 735B; Smith et al., 1995) as well as in the Oman ophiolite (Yamaoka et al., 2012 GCA). In addition, this process will add slab-derived Nb to the wedge (Fig. 4b). Melting to produce low-silica boninites is not at all like the melting that produces arc basalts, either in terms of the slab input or of the mantle substrate that the slab is fluxing.

L205/206: You cannot say the B/d11B shows the thermal slab evolution - you have merely indication that is may be so provide your models hold that are derived from other models. This is very speculative. Also, how does the hydrous LSB component in Fig 5 fit in?

We address this in greater detail above. Our constraints on slab temperatures relate

primarily to the initial generation of melts of slab gabbros to flux LSB magmatism, followed by fluids from slab sediments and basalts to produce HSB. Our modeling based on experimental results suggests temperatures approaching 1000°C to melt amphibolitized slab gabbros, while slab temperatures later have to be low enough to not produce melts of subducting sediments during HSB petrogenesis. As noted above, our constraints are thus reasonably limited, but they are consistent with the inferred compositional changes in slab inputs inferred from the boron and radiogenic isotopic compositions of our two boninite types, as well as from their trace element systematics. The 'Hydrous fluids' signature shows similarities to what others have described as the deep subduction input endmember in mature volcanic arcs (Pearce et al., 2007 EPSL; Woodhead et al., 2012 Geology).

L213: unfortunately, you never detail what the indication for sediment input are. This should be summarized somewhere in the paper. Best in the introduction.

We now explain these specifics in the abstract, introduction, and discussion (L27-33; L62-65; L295-309) and in subduction initiation cartoon (Now Figure 7). We have written these sections to distinguish between our observations our model in the text and the abstract.

L232-253: It should be critical to given the age data here, especially as it is mentioned early on that the extrusion of the boninites lasted no longer than 1 myr

We have added the age information to the text.

L283-284: Where are you locating the subduction channel? In the forearc mantle? This is an unexpected place.

The subduction channel in mature arcs develops at the plate interface. However, the primary sources for the materials in the subduction channel, in particular the abundance of serpentine and like hydrated magnesian minerals that comprise much of the mélange matrix in this setting, tend to be derived from the overriding plate. In our case the subduction channel as it's commonly recognized is relevant only in that it appears to be

absent in the earliest stages of subduction, thanks to the high temperatures encountered near the plate interface during subduction initiation.

Reviewers' comments:

Reviewer #1 (Remarks to the Author):

Comments on 'Boron isotopes in boninites document rapid changes in slab inputs during subduction initiation'

Main comments:

In light of the second round of reviews, the manuscript has been significantly restructured again. In addition, the text and figures that needed further clarification have been edited. These changes, particularly using subheadings, have significantly improved the clarity and reasoning of the manuscript. I still agree with the other reviewer comments, that the evidence used to trace the slab inputs largely relies on data the previous studies (e.g 87Sr/86Sr & Hf) and that the new data presented has been fitted into an existing model.

One major concern I now have is in regard to the stratigraphic succession. This was not evident in earlier versions of the manuscript that didn't include the stratigraphic sampling locations. It does therefore appear that the distinction between the LSB and HSB is geochemical, rather than temporal. This means that the 'rapid geochemical changes' discussed, may not exactly hold up. These rapid temporal changes are critical to the tectonic interpretations in the paper. Therefore, if no temporal change is present, then the conclusions need to be significantly revised. I believe more transparency in the wording of the manuscript, to put less weight onto the temporal changes would be needed before publication.

In terms of transparency, some more detail as to why these particular samples were chosen for this study, and some explanation as to why there appears to be a temporal overlap between LSB and HSB would be beneficial. For example, It seems like the 2 HSB samples (and a purple LSB) from U14442A are within a sheet flow (rather than pillow/hyaloclastites). Could there be a reason for this? It is also not easily apparent as to the purple/yellow distinction for LSB (until Fig. 4). So again, with more sampling detail and context up-front (prior to results and discussion), the authors could make a much stronger case for the geochemical differences they observe.

What is interesting, is that the HSB+LSB is different from modern arcs. Therefore, I still believe it is a valuable contribution to discuss the different processes that might produce LSB+HSB during subduction initiation and why this is not seen in modern arcs.

Additional Comments:

In the response to reviewers, the authors use the study of McCaig et al. (2018) as the sole evidence that serpentinite in and below the oceanic crust does not contribute significantly to the boron budget. However, this is one interpretation based on fast spreading oceanic crust. In slower spreading settings, there is evidence that serpentinization extends deeper into the slab and can influence B enrichment in arc volcanics e.g. Cooper et al. (2020, Nature). It would benefit the discussion to mention other scenarios and not simply rule out this B source, particularly when the range in δB of the Layer 3 gabbro is so large, and entirely overlaps the B range of serpentinites (Fig.3). On that point, how was the average δB of Layer 3 gabbro calculated? it appears to be at the lower end of the range plotted in Fig. 3c. I question whether it is meaningful to use an average value when the range is so large. I do, however, agree with the authors that that the source of B in LSB or HSB is unlikely to be dominantly from serpentinite, but that this signature is picked up later within the Izu Mariana arc volcanics (a point worth mentioning in the discussion).

Line 150: I am not convinced that there is a positive correlation of Ba/Nb and δB (Fig. 5). They cover a broad range in δB , with little variation in Ba/Nb. But the two different LSB samples do have distinctive Ba/Nb. Myself and other reviewers previously mentioned that there is little to distinguish the fit of the two mixing models in this Figure, and this is still the case. I don't think that this figure is particularly useful.

In the section 'A refined model for slab mantle interaction during subduction initiation'. You still do not mention what new evidence (from δB) this contribution brings to refining this model. It still

appears to be interesting geochemical data that is fitted into a pre-conceived model.

Reviewer #2 (Remarks to the Author):

In this version, the authors have done much work to identify and focus on the major findings of their research and to support the findings with lucid explanations and data. The title is changed to show that the B isotopes and abundances are interpreted in the context of other results and that they document changes in slab outputs during subduction initiation.

As I said in my review of version 2, the story (Fig. 7) is well knit, coherent and reasonable, worthy of publication in Nature Communications.

With respect to the comments of reviewer 3, my opinion is that clarifications and answers to individual questions have been made by the authors.

I do not agree with reviewer 3's initial statement that "The component of accretion and subduction erosion this model requires is a complete speculation." This interpretation arises directly from interpretation of the geochemical results and age data. Given that IODP Expedition 352 sampled an in-situ subduction initiation sequence, a rarity in itself, unexpected results should not be discounted unless there are obvious alternatives. I do not see that.

Reviewer #3 (Remarks to the Author):

I much appreciate the detailed revisions, clarifications and responses. It's now clear what model the authors are proposing. However, despite all clarification, substantial problems remain.

Major comments

(1) I welcome the stratigraphic columns present in Fig. 1 that also show the location of the LSB and HSB samples. I feel confounded, however, that the green (HSB) samples are found in both the LSB and HSB sections. In Fig. 1, I see green HSB in both your ochre LSB column. You have no green samples in the HSB of 1442A. Only half of your HSB samples actually comes from the HSB part, your other half comes from the LSB part. This generates the impression that the separation of HSB vs. LSB is a geochemical one, and that the stratigraphic+age aspect was only attributed in retrospect. This would definitely need clarification, since it does not appear that the trace element and isotopes follow the groups that are defined by major elements.

Question: Would this up-core stratigraphic 'trend' hold of fall apart once you took more samples? How can you document 'rapid geochemical changes' (L74) if you have no solid temporal succession? Where is the stratigraphy or the different in physical ages?

(2) The problem of not having a clear time+ stratigraphic trend transmits to not not having clear-cut chemically distinction among HSB and LSB. Yes, they are respective end members of a group, but what really seems to matter that they are as group (HSB+LSB) different from modern arcs.

This is an interesting and noteworthy result. Frankly, the within LSB and HSB variation could as well be considered to be as 'inherent' and as to be expected from such package.

Note, for example that Ba (one of your key elements in distinguishing LSB and HSB) actually overlaps in both series, and is not different as stated in L82.

(3) Overall, I remain unconvinced about the ability of these data to identify a slab gabbroic source. In general, the compositional variations within the LSB+HSB that can be expected from AOC-fluid reaching the mantle wedge. Yes, the d_{11B} of the LSB is a bit lower, but what about B isotope fractionation during subduction? The residual slab (after defluidization) would have lower d_{11B} . I think the 'low 87/86' is overrated in its importance to deduce convincingly a gabbroic slab source. See for example the studies in the western Aleutians where low 87/86 was attributed to AOC, instead of gabbro.

(4) Without a clear temporal progression based on the rock series, any inferences on the geometry

of the early subduction zone, including the notion of sediment accretion, remains speculative.

Comments linked to the text:

L55: I agree that a lower Ti points to mantle source depletion. But why does a high Si indicate also mantle source depletion?

L62-64: Here you present the conclusion of your paper (LSB as slab melts, HSB as fluids) as starting point. This is really odd - do you start with a preconceived model? What you need to expound is instead whether there is a striking, durable and true time-progressive change within the SB series.

L68 and L71: why 'exchange'? there is transfer from slab to wedge, but none from wedge to slab?

L82: there is quite some overlap in MgO and Ba between LSB and HSB, the only true difference is in B and TiO₂

L86: In Fig. 3 (Ba/Nb vs. d_{11B}) the LSB and HSB overlap in d_{11B} with only the yellow LSB extending to higher d_{11B}. This makes me wonder that the clear d_{11B} differences in the other panels are largely due to missing data points. Moreover, to what extent are the differences in Ba/Nb and B/Nb a function of the Nb (and not Ba), given that Nb likely follows Ti in its behavior? This also shows in Figure S1.

L101: what about Layer 2B?

L118: give number for 87/86.

L134: what do you mean quantitatively? That you have a mixture of 70:30 of gabbro to 2A basalts as slab component that is added to mantle wedge? Where are the 'bulk mixing calculations'?

L137: which processes? This is very vague.

L157: Why could the LSB not be a layer 2 basalts component with a lower 87/86? The LSB are still within the range of layer 2 basalts.

And why do you not add Nd isotopes to these data? Combined Nd-Hf isotopes would much better constrain (or discard) the interpretation of gabbroic slab melt?

Why is eHf lower in the LSB (Pacific-MORB-type crust?) than in the FAB (Indian-MORB-type mantle)? Generally, IO has higher eHf than PO. That's one problem. And why does the HSB even have lower eHf than the LSB (sediment)?

L159: Again, the 87/86 argument is diffuse - it could as well be a basalt with lower 87/86. Moreover, Ba/Nb and B/Nb are considered to fractionate during subduction zone processing -- how can you infer a slab source from these element ratios?

L162: are these altered or fresh gabbros?

L167: based on your stratigraphy in Fig 1, your green HSB sample could just represent such a different 'fluxing event' as well?

L172: It suggests that some Nb may have been added to slab (nothing else)

L173: How do you explain Ti retention in slab if rutile and other titanate phases are not stable?

L174: the notation of the partition coefficient is unusual

L188: these conditions could also generate the HSB, according to the inset in Fig 4a

L200: preceding discussion: Why do you not make an attempt fitting the HSB in? With some

'twisting' (similar to that discussed) should be possible.

L201: It would be a great result, if you could show that there is no serpentinite involvement in the LSB/HSB relative to mature arcs.

L216: Again, here is the problem that your HSB samples (according to your Fig. 1) are not correlated with the HSB stratigraphic group - thus, to which extent are the HSB only 'inherent variability' of the LSB/HSB package that was emplaced at the same time?

L216: Which 'other tracers'? Neither in B/Nb nor Ba/Nb, and not even in 87/86 the HSB are drastically different from LSB. Again, note that your HSB samples, are mixing with the LSB stratigraphic column.

L224: I think what you want to say here that the HSB have more fluid input than the LSB. I agree, maybe a little more. But why does this 'little more' justify a separate group given the fact that you do not have a succession in time (according to Fig. 1)?

The important thing here is to consider the LSB/HSB package relative to other active settings that have also been analysed for the same elements. Maybe from that you can build a case that the early LSB/HBS sequence is unique to subduction initiation. But I do not think that the data are strong enough to argue for a stratigraphic/temporal trend + change of source + process.

L255: In fact, in Figure 3a and 3b you show that the HSB component is different from other arcs. You do not show (just discuss) that it is similar. And you contradict yourself in the next sentence...

L269ff: The thermal evolution aspect is more wishful thinking than hard data. Why is the compositional evolution also a thermal slab evolution? How to you connect this

L271: repetition

L273: Reference for the statement on temperature?

L281: now you speak from lack of sediment melting, yet the HSB have both lower eNd and eHf than the LSBhow to you explain this without sediment melting?

L301: accretion would remove only some of the sediment (that's normal), and besides how do you explain the progressively lower eNd and eHf from FAB to LSB and HSB?

L314: but you do NOT show this age gap at all in the data you present; instead there is temporal overlap

L323: the term 'subduction channel' should be used as defined. This 'new' definition of subduction channel in the context of this paper does not make sense.

DETAILED RESPONSES TO THE REVIEWERS

Overall response to reviewers:

A major new criticism from the referees is the stratigraphic succession issue, i.e., whether the HSB were erupted later than the LSB. The reviewers seem to be questioning the idea that the HSB could post-date LSB magmatism if a few HSB samples were found intercalated within in the LSB core section, when geologically speaking this is pretty much exactly what one would find were one to drill a comparable section of similarly erupted lavas on land. In their detailed chronological study of the Expedition 352 suite (Reagan et al., 2019 EPSL), Ar-Ar and U/Pb-based dates on forearc basalts and boninites point to a very limited timespan of eruption, and a clear time gap between the LSB and HSB is not solidly constrained: they inferred the LSB were erupting at ~ 51.3 Ma, while HSB appear at ~ 50 Ma (Reagan et al., 2019 EPSL; Figure 5). The ‘relative ages’ of the HSB and LSB are constrained by their stratigraphy.

Since our boron isotope manuscript has been in review, two comprehensive major and trace element studies on the Expedition 352 boninites were submitted and are now accepted for publication (attached as **Related Manuscript Files**). The Shervais et al (2020, G-cubed; doi: [10.1029/2020GC009093](https://doi.org/10.1029/2020GC009093)) paper focuses on the petrology and stratigraphy of the Expedition 352 boninite suite. On the *JOIDES Resolution*, the Expedition 352 Science Team used a hand-held portable XRF (pXRF) instrument to track chemical variations in recovered basalts and boninites as part of the core logging process, compiling over 2000 individual measurements (Ryan et al., 2017). The data in the Shervais et al. paper, and in our Expedition summary publications (Reagan et al., 2015; 2017), both of which include the pXRF shipboard data, all clearly show that the HSB overlie the LSB, with uncommon appearances of HSB deeper in the section, consistent with intrusions through LSB strata.

Another paper, focused on analyses of forearc basalt and boninite glasses, has also been accepted (Coulthard et al., 2020, G-cubed; doi:[10.1029/2020GC009054](https://doi.org/10.1029/2020GC009054)). As the HSB have higher K/Ti and La/Sm than the LSB, Coulthard et al (2020) used the

K/Ti ratios of their highly evolved high-Mg andesite glass samples (< 8wt% MgO) to identify their boninite parental magma types, i.e. LSB or HSB. Generally, glass samples from the upper part of the boninite holes (< ~250 mbsf) have higher K/Ti and La/Sm than those from the lower parts of the holes, consistent with the findings from the Shervais et al (2020) petrology and stratigraphy paper. Coulthard et al. (2020) also found a few anomalous high K/Ti glass samples in Hole U1442A (at ~310 and 450-500 mbsf), and described these as high-K LSB. Our deeper seated HSB (305-325 mbsf) from Hole U1442A are from the same depth interval as one of the high-K LSB samples of Coulthard et al. (2020). One of these unusual Hole U1442A samples has MgO > 11 wt%, indicating that these “high-K LSB” samples should in fact be classified as HSB (Source data). Major element data for the two deep HSB from Hole U1439C (289.8-290.1 mbsf; analyzed only in this study) have MgO > 11 wt% and thus also support this classification. As our data demonstrate, and the larger boninite datasets confirm, while HSB samples are occasionally encountered within LSB sequences, no LSB lavas have been found in the HSB sequence, which is good geologic evidence that the HSB eruptions began after LSB magmatism.

Ar-Ar dating of HSB from Bonin islands to the west of the Expedition 352 drill sites show that HSB magmatism persisted until ≈ 46 Ma (Ishizuka et al., 2020 JPet), some ~ 4 Ma later than our uppermost HSB boninites (Reagan et al., 2019 EPSL). These boninites show similar geochemical characteristics to the Expedition 352 HSB, e.g., limited Hf isotope variation at the similar value of $\epsilon_{\text{Hf}} \approx 13$ (Figure 6f of Ishizuka et al., 2020 JPet) and correlated Cs/La with ϵ_{Nd} variations, indicating sediment-derived fluid contributions (Figure 6 of this MS). Thus the locus of HSB volcanism appears to have moved westward, i.e., away from the trench, after ~ 50 Ma (Ishizuka et al., 2020 JPet). These observations are consistent with our conclusions, e.g., early sediment accretion and later erosion, and gradual cooling of the forearc mantle wedge after subduction initiation.

Our boron isotope dataset is necessarily smaller than those presented in these recent G-cubed papers and is much smaller than that represented by our shipboard

pXRF results. Our samples are necessarily biased toward highly glassy materials suitable for B isotopic study. Our addition of the depth plot at reviewer request appears to have muddied the waters for referees, who questioned whether the HSB were in fact generated later than LSB, a point which was clear from the shipboard data and is implicit in the Reagan et al (2017) overview paper on our Expedition results, but has only now been explicitly stated in the journal literature. We appear to need to clarify the stratigraphy as Reviewer #1 and #3 suggested, so we now cite the results of the Shervais et al (2020) paper. Intrusions of younger lavas through older lava sequences frequently occur in stratigraphic volcanic successions. In the Izu-Bonin subduction system, where early eruptions of LSB lavas transitioned upsection into comparatively long-lived eruptions of HSB, the likelihood of such crosscutting relationships is high.

This information has been added to the text (Line 48-57).

We have also modified the depth of Figure 1c as meters below the seafloor (mbsf) to facilitate comparison with the study of Shervais et al. (2020) and Coulthard et al. (2020). The HSB from Li et al. (2019 EPSL) are also plotted in the revised diagram.

Reviewer #1: (Remarks to the Author):

Comments on ‘Boron isotopes in boninites document rapid changes in slab inputs during subduction initiation’

Main comments:

In light of the second round of reviews, the manuscript has been significantly restructured again. In addition, the text and figures that needed further clarification have been edited. These changes, particularly using subheadings, have significantly improved the clarity and reasoning of the manuscript. I still agree with the other reviewer comments, that the evidence used to trace the slab inputs largely relies on data the previous studies (e.g $^{87}\text{Sr}/^{86}\text{Sr}$ & Hf) and that the new data presented has been fitted into an existing model.

The implication of the last of this reviewer’s comments, that B isotopes are

somehow not making that significant a contribution to our understanding of boninite genesis and subduction initiation, is something we clearly disagree with, and we've worked in our revisions to try and highlight what this unusual element and its associated isotopic system show us.

As an extreme fluid-mobile species that is particularly sensitive to subduction phenomena, boron provides insights into subduction-related magmatic and metamorphic phenomena that other tracers simply don't. Boninite petrogenesis has historically been treated in much the same way as arc basalt petrogenesis (i.e., a "slab fluid" added to, in the case of boninites, an extremely depleted mantle source). If this "slab fluid" was in fact a C-O-H dominated fluid phase, then we would see highly decoupled systematics in both B abundances and B isotopes relative to large-ion-lithophile elements and radiogenic isotopes, such as is observed routinely in mature volcanic arcs. The fact that we don't see this – and that B isotopes appear to correlate broadly with Hf isotopes in the LSB sub-suite (i.e., an extreme fluid-mobile tracer correlating with an extreme fluid-immobile one) – rules out a significant role for any kind of hydrous fluid, and supports the idea of an ocean crustal melt input to LSB mantle sources. Boron and boron isotopes further tell us is that this first slab input to LSB mantle sources was most likely derived from lower crustal gabbros, and not altered basalts, as LSB boron abundances are too low, and their $\delta^{11}\text{B}$ values range too high. This unique pattern of higher $\delta^{11}\text{B}$ with low boron concentrations is evident only in the LSB, i.e., B/Nb and Ba/Nb are not strictly positively correlated with $\delta^{11}\text{B}$. In all other volcanic arc lavas worldwide, $\delta^{11}\text{B}$ and B/Nb show clear positive correlations (see Ryan and Chauvel, 2014; deHoog and Savov, 2018). This geochemical finding is totally new; it provides a reason for differences in the $^{87}\text{Sr}/^{86}\text{Sr}$ signatures of the HSB and LSB, and it places interesting physical constraints on the tectonic specifics of starting a subduction zone.

Our radiogenic isotope paper (Li et al., 2019 EPSL), the G-cubed petrology and stratigraphy paper (Shervais et al. 2020), and the G-cubed glass paper (Coulthard et al 2020) all point to subducted sediment inputs to the HSB mantle source. It is the relationships among boron and other fluid mobile and immobile elements that

constrain this contribution from sediments to being via high-T hydrous fluids. This is our second important geochemical finding and it puts constraints on the temperature evolution of the slab and mantle wedge through the period of boninitic magmatism. Our conclusions are consistent with the observation that the locus of HSB volcanism appears to have moved westward, away from the trench, after ~ 50 Ma (e.g., Ishizuka et al., 2020 JPet). As reviewer #2 indicated, our model arises directly from this geochemical finding, as no other interpretation can explain all the available data.

We have added a sentence to our introduction to further clarify the motivations of our study. (Line 70-74).

One major concern I now have is in regard to the stratigraphic succession. This was not evident in earlier versions of the manuscript that didn't include the stratigraphic sampling locations. It does therefore appear that the distinction between the LSB and HSB is geochemical, rather than temporal. This means that the 'rapid geochemical changes' discussed, may not exactly hold up. These rapid temporal changes are critical to the tectonic interpretations in the paper. Therefore, if no temporal change is present, then the conclusions need to be significantly revised. I believe more transparency in the wording of the manuscript, to put less weight onto the temporal changes would be needed before publication.

Please see the overall response to reviewers. This concern is basically moot, as noted.

In terms of transparency, some more detail as to why these particular samples were chosen for this study, and some explanation as to why there appears to be a temporal overlap between LSB and HSB would be beneficial. For example, It seems like the 2 HSB samples (and a purple LSB) from U14442A are within a sheet flow (rather than pillow/hyaloclastites). Could there be a reason for this? It is also not easily apparent as to the purple/yellow distinction for LSB (until Fig. 4). So again, with more sampling detail and context up-front (prior to results and discussion), the authors could make a much stronger case for the geochemical differences they observe.

As we replied in the **Overall response to reviewers**, there are uncommon appearances of HSB lavas in the deeper LSB section, though no LSB lavas have been recognized in the upper HSB sections at either Site U1439 or Site U1442, suggesting that the HSB lavas encountered among the LSB reflect younger, cross-cutting units. Our reason for selecting some deeper HSB samples for B isotope study was to discern whether their subduction signatures were similar or different from those of HSB at the top of the hole. Our results reveal that all the Exp. 352 HSB are very similar, in particular in their B contents and $\delta^{11}\text{B}$. Additional fluid inputs to a highly depleted harzburgitic mantle source (e.g., a mantle source residual to LSB magmatism) is required for HSB generation. Our study as well as the recent paper by Ishizuka and coworkers in *Journal of Petrology* (2020) both find that the kinds of subduction inputs are highly correlated with the type of the magmatism during subduction initiation. We have added this information to the text (Line 378-381).

What is interesting, is that the HSB+LSB is different from modern arcs. Therefore, I still believe it is a valuable contribution to discuss the different processes that might produce LSB+HSB during subduction initiation and why this is not seen in modern arcs.

We agree with the reviewer that subduction related magmatism and inputs from the downgoing plate are different during subduction initiation as compared to mature arcs. As noted above our findings allow us to put unique physical and thermal constraints on slab-mantle exchange phenomena involved in boninite petrogenesis, building on past work.

Additional Comments:

In the response to reviewers, the authors use the study of McCaig et al. (2018) as the sole evidence that serpentinite in and below the oceanic crust does not contribute significantly to the boron budget. However, this is one interpretation based on fast spreading oceanic crust. In slower spreading settings, there is evidence that serpentinization extends deeper into the slab and can influence B enrichment in arc

volcanics e.g. Cooper et al. (2020, Nature). It would benefit the discussion to mention other scenarios and not simply rule out this B source, particularly when the range in δB of the Layer 3 gabbro is so large, and entirely overlaps the B range of serpentinites (Fig.3). On that point, how was the average δB of Layer 3 gabbro calculated? it appears to be at the lower end of the range plotted in Fig. 3c. I question whether it is meaningful to use an average value when the range is so large. I do, however, agree with the authors that that the source of B in LSB or HSB is unlikely to be dominantly from serpentinite, but that this signature is picked up later within the Izu Mariana arc volcanics (a point worth mentioning in the discussion).

In arcs where slow spreading oceanic crust is subducting, or the subducting crust has large-scale fracture zones, serpentinite may play an important role in the arc boron budget. The Lesser Antilles (Cooper et al., 2020 Nature) and the South Sandwich arcs (Tonarini et al., 2011 EPSL) are good cases. Common features of these two island arcs is that they include some of the highest $\delta^{11}B$ and B/Nb lavas thus far measured, and they both subduct slow spreading oceanic crust. However, these kinds of $\delta^{11}B$ and B/Nb signatures are not observed in the arcs surrounding the Pacific plate. This may be related to the fact that Pacific plate crust is generated largely at fast spreading ocean ridges. Our boninites have much lower B/Nb and $\delta^{11}B$ than either Lesser Antilles (Cooper et al., 2020 Nature) or South Sandwich (Tonarini et al., 2011 EPSL) samples, ruling out a dominant role for subducting plate serpentinite as a control on the B budgets of Izu-Bonin boninites. The low $^{87}Sr/^{86}Sr$ signature of our LSB relative to Lesser Antilles arc lavas (at ~ 0.7055 ; Freymuth et al., 2016 Geology; Macdonald et al., 2000 ESR) also support our interpretation.

Klaver et al. (2020 GCA 288 101-119) proposed a model wherein the breakdown of serpentinite in the slab mantle releases a pulse of fluid at sub-arc depths. These fluids travel through and equilibrate with the overlying oceanic crust and sediments and induce wet partial melting of the upper altered crust and sediments. This hydrous melt is purported to have low $^{87}Sr/^{86}Sr$, as in their model $> 70\%$ Sr contribution from the slab lithosphere. This hydrous melt is then delivered to the mantle source of arc magmas as a single metasomatic component. This model was

used to explain the Sr isotopic signatures of the Mariana arc basalts, though its viability for explaining the B abundances and B isotopic signatures of Mariana arc basalts is questionable, given the comparatively much higher concentrations of boron and very high $\delta^{11}\text{B}$ of abyssally formed serpentinites. Their serpentinite breakdown model is also ineffective at explaining the HSB, which have both high $^{87}\text{Sr}/^{86}\text{Sr}$ and low $\delta^{11}\text{B}$ relative to mature arc lavas, and it fails to explain the low Ba/Nb signatures of the LSB. Thus, break down of serpentinite in the slab is not a big factor affecting the Sr isotopes of the boninites, as we can't imagine how it plays the crucial role during LSB generation, but then disappears later when the HSB erupted.

We have added information to the text regarding this recent study (Line 215-230).

The $\delta^{11}\text{B}$ of Layer 3 gabbro is based on data from Smith et al. (1995, Chemical Geology), for gabbros from ODP Hole 735B. ODP Hole 735B is the only source of drilled samples taken directly from intact oceanic crustal Layer 3, particularly for Boron isotope data. The low B/Nb and high $\delta^{11}\text{B}$ signatures of the ODP Hole 735B gabbros are consistent with data from ophiolites, where $\delta^{11}\text{B}$ increases while B/Nb decreases with depth (Yamaoka et al., 2012 GCA; 2015 Chemical Geology). Therefore, we think using the Smith et al. (1995, Chemical Geology) gabbroic data is reasonably representative of the Layer 3 oceanic crustal signature. Some high $\delta^{11}\text{B}$ data for Layer 3 gabbros come from IODP Hole U1415 (McCaig et al., 2018) at Hess Deep. McCaig et al. (2018) thought all their collected samples were likely from the damage zone of a multistrand fault, and that the alteration intensity is thus unlikely to be typical of fast-spreading lower oceanic crust away from faults, which are still analytically inaccessible. In the revised submission, the McCaig et al. (2018) data is distinguished from the Smith et al. (1995, Chemical Geology) data. This information have been added to the text (Line 134-136).

For the mature Izu-Bonin-Mariana arc volcanics, forearc mantle serpentinite is a possible Boron source (e.g., Ryan and Chauvel, 2014). This information has been added the text (Line 359-365).

Line 150: I am not convinced that there is a positive correlation of Ba/Nb and δB (Fig. 5). They cover a broad range in δB , with little variation in Ba/Nb. But the two different LSB samples do have distinctive Ba/Nb. Myself and other reviewers previously mentioned that there is little to distinguish the fit of the two mixing models in this Figure, and this is still the case. I don't think that this figure is particularly useful.

Rewording: In addition, the data array on a plot of Ba/Nb versus $\delta^{11}B$ shows a funnel-like shape, converging toward a low Ba/Nb low $\delta^{11}B$, depleted mantle like value (Fig. 5). (line 160-162)

In the manuscript we obviously make note of the endmember calculations, but what these diagrams show are a family of hyperbolic mixing models, between a depleted, mantle-like low B, low Ba, low $\delta^{11}B$ endmember and a high B, higher Ba endmember with variably elevated $\delta^{11}B$, consistent with the observed variability in Layer 3 ocean crust. These diagrams highlight that the $\delta^{11}B$ heterogeneity of the gabbro exert a primary control over the $\delta^{11}B$ of the LSB. It would have perhaps been better to plot B/Nb versus $\delta^{11}B$, but unfortunately it was impossible to obtain a large number of LSB samples that were fresh enough for boron concentration determinations (as noted in the paper, our B isotopic data for the LSB include leached samples, which we've shown through past studies can provide intrinsic $\delta^{11}B$ measurements, but not B concentration measurements). The Ba/Nb versus $\delta^{11}B$ diagram show most clearly the data patterns for slab-mantle mixing that are important to our interpretation. As such, we hope to keep these two diagrams in the paper.

In the section 'A refined model for slab mantle interaction during subduction initiation'. You still do not mention what new evidence (from δB) this contribution brings to refining this model. It still appears to be interesting geochemical data that is fitted into a pre-conceived model.

We have added our main geochemical findings to this section to show the new evidence this contribution brings to refining this model (see our response to the reviewer's Main comments above) (Line 324-332).

Reviewer #2 (Remarks to the Author):

In this version, the authors have done much work to identify and focus on the major findings of their research and to support the findings with lucid explanations and data. The title is changed to show that the B isotopes and abundances are interpreted in the context of other results and that they document changes in slab outputs during subduction initiation. As I said in my review of version 2, the story (Fig. 7) is well knit, coherent and reasonable, worthy of publication in Nature Communications. With respect to the comments of reviewer 3, my opinion is that clarifications and answers to individual questions have been made by the authors. I do not agree with reviewer 3's initial statement that "The component of accretion and subduction erosion this model requires is a complete speculation." This interpretation arises directly from interpretation of the geochemical results and age data. Given that IODP Expedition 352 sampled an in-situ subduction initiation sequence, a rarity in itself, unexpected results should not be discounted unless there are obvious alternatives. I do not see that.

Reviewer #3 (Remarks to the Author):

I much appreciate the detailed revisions, clarifications and responses. It's now clear what model the authors are proposing. However, despite all clarification, substantial problems remain.

Major comments

(1) I welcome the stratigraphic columns present in Fig. 1 that also show the location of the LSB and HSB samples. I feel confounded, however, that the green (HSB) samples are found in both the LSB and HSB sections. In Fig. 1, I see green HSB in both your ochre LSB column. You have no green samples in the HSB of 1442A. Only half of your HSB samples actually comes from the HSB part, your other half comes from the LSB part. This generates the impression that the separation of HSB vs. LSB

is a geochemical one, and that the stratigraphic+age aspect was only attributed in retrospect. This would definitely need clarification, since it does not appear that the trace element and isotopes follow the groups that are defined by major elements.

Question: Would this up-core stratigraphic 'trend' hold or fall apart once you took more samples?

How can you document 'rapid geochemical changes' (L74) if you have no solid temporal succession? Where is the stratigraphy or the different in physical ages?

Please see reply to the "**Overall response to reviewers**" for the reason we can conclude the HSB was erupted later than the LSB, with uncommon appearances of HSB deeper in the section. We have also modified the depth of Figure 1c as meters below the seafloor (mbsf) to facilitate comparison with the studies of Shervais et al (2020) and Coulthard et al (2020). The HSB from Li et al. (2019 EPSL), which were analyzed for radiogenic isotopes are also included in the revised diagram.

Some further specifics:

Samples examined in this study were selected from the working halves of cores for Holes U1439A, U1439C and U1442A, to represent both the natural chemical variability of the IODP Expedition 352 boninites and to reflect overall core stratigraphy. Hole 1439C documents the lithologic change of boninite and its differentiates to the fullest extent, comprising nine volcanic units and a basal intrusive unit. The stratigraphy of Hole U1442A is broadly similar to Hole U1439C but is by comparison more simple, consisting of only four volcanic units. One volcanic unit (all HSB) with three cores was recovered from U1439A (Shervais et al., 2020). Therefore, Hole U1439C was investigated in detail in this study. HSB lavas from < ~250 mbsf from these three holes are all relatively fresh and show largely similar geochemical variations (Li et al., 2019; Shervais et al., 2020). Two HSB samples from U1439A that reflect the upmost part of the section were selected to supplement HSB samples < 250 mbsf from Hole U1439C. Four HSB samples identified from > 250 mbsf in Holes U1439C and U1442A were also selected based on our shipboard pXRF logging

results (Ryan et al., 2017; Shervais et al., 2019; 2020). We chose these samples to compare their geochemistry with the HSB from < 250 mbsf in view of their uncommon occurrences deeper in these cores. Fresh LSB were rarely recovered in Expedition 352. LSB from the thick lower units in Hole U1442A were heavily sampled to supplement the LSB available from Hole U1439C. We have added information to clarify our sampling strategy (Line 368-383).

(2) The problem of not having a clear time+ stratigraphic trend transmits to not not having clear-cut chemically distinction among HSB and LSB. Yes, they are respective end members of a group, but what really seems to matter that they are as group (HSB+LSB) different from modern arcs. This is an interesting and noteworthy result. Frankly, the within LSB and HSB variation could as well be considered to be as 'inherent' and as to be expected from such package. Note, for example that Ba (one of your key elements in distinguishing LSB and HSB) actually overlaps in both series, and is not different as stated in L82.

As has been evident from the shipboard pXRF and ICP-OES results that are the basis of results presented in Reagan et al (2017), and has now been confirmed by the more detailed results presented in Shervais et al (2020), the HSB and LSB are geochemically distinct, and HSB eruptive activity followed that of the LSB. Please see reply to the “**Overall response to reviewers**” for the detailed reasons why we can draw these conclusions.

The Ba content of any specific sample is going to be controlled by the degree of mantle source depletion, the magnitude of the fluxing subduction input, the extent of mantle melting in response to that fluxing, the extent of fractional crystallization, and also crystal accumulation in the magma. Ba/Nb can effectively discriminate the HSB from the LSB as this ratio is little affected by melt extent or by fractional crystallization/crystal accumulation of the magma. Our HSB all have higher Ba/Nb than any of the LSB (Figures 3 and 5). This is also evident on a Ba versus Nb

correlation plot (Supplementary Figure S2).

Rewording: Different B versus Ba correlations within the LSB and HSB suites also suggests their source difference (Line 92-93).

(3) Overall, I remain unconvinced about the ability of these data to identify a slab gabbroic source. In general, the compositional variations within the LSB+HSB that can be expected from AOC-fluid reaching the mantle wedge. Yes, the d11B of the LSB is a bit lower, but what about B isotope fractionation during subduction? The residual slab (after defluidization) would have lower d11B.

The reviewer seems to be framing this critique in the context of a normal arc setting, where early, low-temperature losses of boron shallow in the system leave behind both an isotopically lighter slab and a reservoir/return flux of isotopically heavy boron – Ryan and Chauvel (2014) and others have advocated for such a model. The serpentinite mud volcanoes of the Mariana forearc are a type example of this kind of high B, high $\delta^{11}\text{B}$, shallow subduction channel reservoir/outflux, one that Savov et al (2005; 2007) have suggested may account for >70% of the downgoing slab's initial boron inventory.

The problem with this kind of thinking with regard to boninite petrogenesis is that everything we know about the subduction initiation setting is at odds with this picture. Initial slab foundering leads to mantle upwelling, melting, and the eruption of tholeiitic “forearc” basalts with no measurable slab signature (see Reagan et al 2010; 2017; Shervais et al 2019; Coulthard et al 2020). These basalts create the early crust that fills the gap between the plates. Slab heating in this scenario should happen initially at very shallow depths through slab contact with this same upwelling, partially melting upper mantle. Temperatures would thus be high (our estimates for the slab during LSB petrogenesis are between 975°C-1015°C), which would preclude any significant chemical fractionation of the boron isotopes in any slab-derived mobile phase that might be generated. Thus, in the exotic and hot circumstances of subduction initiation, the boron isotopes

serve as a “fingerprint” for different pieces of the slab, so inputs from marine sediments (at $\approx -1\%$) or altered basalts (at $\approx +1\%$) should be recognizable. And, as it turns out, they are – but only in the HSBs! By contrast, the LSB’s show highly variable but on average higher $\delta^{11}\text{B}$ ($> +4\%$), a range and variability that is the same in both glassy and leached LSB samples. The only crustal or mantle reservoir that looks at all like the B isotopes in the LSB’s is Layer 3 oceanic gabbros, the small dataset for which reflects an even larger variation in $\delta^{11}\text{B}$ (from -4% to $+26\%$, with a mean around $+6\%$). The large $\delta^{11}\text{B}$ range of Layer 3 gabbros is consistent with higher temperature, but less pervasive seawater/rock interactions at depth in the crust. As it turns out, Layer 3 gabbros show an unusually limited range in $^{87}\text{Sr}/^{86}\text{Sr}$, with a mean value similar to that of our LSBs, but lower than that of altered basalts or HSBs; and they also match up well with LSBs in terms of parameters such as $\delta^{18}\text{O}$ (e.g., Coulthard et al 2020). **So, if one presumes that B isotopes cannot be chemically fractionated under conditions consistent with boninite petrogenesis (which we would claim to be a given, from considerations of slab and mantle temperatures during subduction initiation), then of the possible slab reservoirs that one could point to as explaining the B isotope signature in LSBs, Layer 3 gabbros are easily the most sensible option.** That the range in LSB $\delta^{11}\text{B}$ is considerably less than is observed in oceanic gabbros is consistent with the extraction of small degree, amphibolite-facies slab melts, that might serve to average out some local B isotopic variability, but not pervasively re-equilibrate the downgoing plate.

I think the ‘low $^{87}/^{86}$ ’ is overrated in its importance to deduce convincingly a gabbroic slab source. See for example the studies in the western Aleutians where low $^{87}/^{86}$ was attributed to AOC, instead of gabbro.

It’s worth noting that we are not using the Sr isotopes to argue for a Layer 3 gabbro slab source – we’re making that argument from the boron isotopes, and the existing Sr and O isotope data appear to be consistent with the idea. As regards the use of Sr isotopes in the Aleutians, Yogodzinski and coworkers seek to distinguish

between eastern and central Aleutian lavas, where sediment inputs are of high importance, and the western Aleutians, where sediment inputs are minimal. They argue for a basaltic melt, i.e., adakitic component as a primary slab input in the Western lavas. Eastern and Central Aleutian lavas have more radiogenic Sr isotopic fingerprints than do Western Aleutian lavas, which are MORB-like overall (<0.7029). They attribute this very low $^{87}\text{Sr}/^{86}\text{Sr}$ in the Western lavas to melting of an eclogitic slab at depth. While this is entirely reasonable in the context of modern Aleutian magmatism, it is outside the realm of the possible during subduction initiation magmatism in the Izu-Bonin forearc. That Layer 3 gabbros have lower and less variable $^{87}\text{Sr}/^{86}\text{Sr}$ than Layer 2 basalts is a reflection of the less pervasive nature of alteration processes as one gets deeper in the ocean crust.

[Full disclosure: co-author Ryan is working with Yogodzinski on conducting a B and B isotope survey of the Aleutians, toward trying to tease out the roles of serpentinite, crust, and sediment in slab inputs along this arc]

(4) Without a clear temporal progression based on the rock series, any inferences on the geometry of the early subduction zone, including the notion of sediment accretion, remains speculative.

There is a clear temporal progression. Please see reply to the “**Overall response to reviewers**” for the reason why we can conclude the HSB was erupted later than the LSB.

Comments linked to the text:

L55: I agree that a lower Ti points to mantle source depletion. But why does a high Si indicate also mantle source depletion?

Boninites form via melting of a CPX-poor mantle source, and are in equilibrium with a harzburgitic residuum. During harzburgite melting, OPX (which has high SiO_2) is the primary melting phase. At lower pressures OPX melts incongruently to yield a siliceous melt and residual olivine, driving melt silica contents up (Pearce and Reagan,

2019 Lithosphere; Falloon and Danyushevsky, 2000 JP).

L62-64: Here you present the conclusion of your paper (LSB as slab melts, HSB as fluids) as starting point. This is really odd - do you start with a preconceived model? What you need to expound is instead whether there a striking, durable and true time-progressive change within the SB series.

Thanks for pointing this out. Now corrected.

We have added a sentence in the introduction to clarify the motivation for our study: “However, which part of the oceanic crust, upper crust, lower crust or serpentinite fluxed the LSB mantle source, and what phase (fluid versus melts) from the sediments fluxed the HSB mantle source are not well constrained yet. These questions are directly relevant to how these igneous activities reflect the geometry of the slab and its thermal evolution during the subduction initiation.” (Line 70-74).

L68 and L71: why ‘exchange’ ? there is transfer from slab to wedge, but none from wedge to slab?

Reworded: inputs.

L82: there is quite some overlap in MgO and Ba between LSB and HSB, the only true difference is in B and TiO₂

The point of this statement is to note that there are multiple fractionation suites among the boninites reflecting primary magmas with different major element compositions, consistent with melting along the olivine-enstatite cotectic.

Rewording:

“While the B concentration ranges of HSB and LSB show overlap, there are clear distinctions with respect to MgO and TiO₂, suggesting different magmatic sub-suites with distinct B abundances. Different B versus Ba correlations in the LSB and HSB suites also suggests source differences (Fig. 2)”

L86: In Fig. 3 (Ba/Nb vs. $\delta^{11}\text{B}$) the LSB and HSB overlap in $\delta^{11}\text{B}$ with only the yellow LSB extending to higher $\delta^{11}\text{B}$. This makes me wonder that the clear $\delta^{11}\text{B}$ differences in the other panels are largely due to missing data points. Moreover, to what extent are the difference in Ba/Nb and B/Nb a function of the Nb (and not Ba), given that Nb likely follows Ti in its behavior? This also shows in Figure S1.

Reworded: $\delta^{11}\text{B}$ in Expedition 352 boninites ranges from -1.6‰ to +8.3‰, with clear differences between sub-suites when linked to B/Nb and Ba/Nb (Figs. 3a, b).

Our interpretations are based on the various correlations among $\delta^{11}\text{B}$, B/Nb, Ba/Nb, $^{87}\text{Sr}/^{86}\text{Sr}$ and ϵHf . Please see the interpretation in the following sections.

From higher to lower, Ba/Nb and B/Nb variations run from: HSB > High Ba/Nb LSB > Low Ba/Nb LSB, asid from one HSB sample that has B/Nb like the LSB (lowest B/Nb and Cs/Nb, indicating the lowest degree of sediment fluids input).

As Nb has a much smaller partition coefficient into amphibolite than Ti (Tiepolo et al., 2007), Nb is mobile during amphibolite melting (Figure 4b).

L101: what about Layer 2B?

Only three B isotope data exist for Layer 2B dolerites (Ishikawa and Nakamura (1992; DSDP/ODP Hole 504B). These have much lower $\delta^{11}\text{B}$ (-0.1‰ to 1.0‰) than Layer 3 gabbros (\approx +6‰ ODP Hole 735B), and are indistinguishable from the Layer 2A basalts (0.8‰; DSDP Holes 417A, 417D and 418A). We cannot find supporting geochemical data for these three dolerite samples so as to include them in our diagrams. But we added their B isotope information in our text (Line 118-120).

L118: give number for 87/86.

Added, 0.7032-0.7038

L134: what do you mean quantitatively? That you have a mixture of 70:30 of gabbro to 2A basalts as slab component that is added to mantle wedge? Where are the 'bulk

mixing calculations' ?

Reworded: Bulk mixing calculations indicate that a mixture of 70:30 of Layer 3 gabbro to Layer 2A basalt could yield $\delta^{11}\text{B}$ and $^{87}\text{Sr}/^{86}\text{Sr}$ signatures comparable to those of the LSB (Fig. 3c).

The bulk mixing result is plotted in Figure 3c and explained in the Figure Caption.

L137: which processes? This is very vague.

Reworded: slab dehydration/melting process

L157: Why could the LSB not be a layer 2 basalts component with a lower 87/86? The LSB are still within the range of layer 2 basalts.

And why do you not add Nd isotopes to these data? Combined Nd-Hf isotopes would much better constrain (or discard) the interpretation of gabbroic slab melt?

Why is ϵHf lower in the LSB (Pacific-MORB-type crust?) than in the FAB (Indian-MORB-type mantle)? Generally, IO has higher ϵHf than PO. That's one problem. And why does the HSB even have lower ϵHf than the LSB (sediment)?

The problem with Layer 2 basalts is their uniformly lower $\delta^{11}\text{B}$. And as noted above, we see this Layer 2 signal clearly in the HSBs, so we know what that should look like. We've built our interpretation on a comprehensive foundation of the mobility of the different trace elements, on both B and Sr isotopes in the LSB, and available data for the composition of the oceanic crust. Our explanations are not based on a single element or isotope.

The Nd-Hf isotope correlations are published in our 2019 EPSL paper and this information is extensively cited in the text. As Hf and Nd are both highly insensitive to the effects of seafloor alteration, it's unlikely that any Nd-Hf isotopic correlations can help to distinguish a melt from the Layer 2 basalt or Layer 3 gabbro – in fact no such distinctions are evident in Li et al (2019). The reason that LSB have lower ϵHf than the FAB is that they contain a melt contribution from the subducted Pacific ocean

crust. This information is in the original text, on Line 156-158. This explanation is in our EPSL paper and cited in the text.

The HSB has the uniformly low ϵ_{Hf} , similar to the lowest ϵ_{Hf} seen in the LSB (Li et al., 2019 EPSL). It's an inherited signature of oceanic crust melting during LSB generation.

L159: Again, the 87/86 argument is diffuse - it could as well be a basalt with lower 87/86. Moreover, Ba/Nb and B/Nb are considered to fractionate during subduction zone processing -- how can you infer a slab source from these element ratios?

Our explanation is based on current observations as to the chemical composition of the oceanic crust. Layer 2A basalts with relatively low $^{87}\text{Sr}/^{86}\text{Sr}$ do not have high $\delta^{11}\text{B}$, as documented in existing data on DSDP/ODP/IODP samples.

We agree with B/Nb and Ba/Nb fractionation, the sentence regarding B/Nb and Ba/Nb have been deleted from the revision.

L162: are these altered or fresh gabbros?

The gabbro data is from ODP Hole 735B, the only source of drilled samples taken directly from intact oceanic crustal Layer 3. These gabbros have experienced some degree of alteration, as they include samples with higher $\delta^{11}\text{B}$ and lower $\delta^{18}\text{O}$ than fresh MORB. We see this variation as the natural signature of subducting gabbroic crust.

L167: based on your stratigraphy in Fig 1, your green HSB sample could just represent such a different 'fluxing event' as well?

Please see reply to the "**Overall response to reviewers**" for the reason why we can conclude the HSB was erupted later than the LSB.

L172: It suggest that some Nb may added to slab (nothing else)

If rutile is stable in the slab, we won't see Nb inputs to the mantle source. Thus, Ba/Nb fractionation cannot be controlled by rutile.

Reworded: Thus, residual titanate phases such as rutile, which would increase the bulk slab-liquid partition coefficient of Nb (D_{Nb}), appear not to have impacted boninite Ba/Nb ratios.

L173: How do you explain Ti retention in slab if rutile and other titanate phases are not stable?

The Ti partition coefficient in amphibole is much higher than that of Nb (Tiepolo et al., 2007).

L174: the notation of the partition coefficient is unusual

However, the ratio of amphibole-liquid partition coefficient between Ba and Nb ($D_{Ba}^{Amph/L}/D_{Nb}^{Amph/L}$) appears to increase with increasing temperature during amphibolite facies melting

L188: these condition could also generate the HSB, according to the inset in Fig 4a

One could conclude this, but based only on this diagram. And this conclusion would contradict findings from ϵNd , $^{87}Sr/^{86}Sr$ and Pb isotopes. A detailed interpretation of the HSB data is in the next section.

L200: preceding discussion: Why do you not make an attempt fitting the HSB in? With some 'twisting' (similar to that discussed) should be possible.

The HSB show completely different geochemical signatures than the LSB, in ϵNd , $^{87}Sr/^{86}Sr$ and Pb isotopes. Detailed interpretation for the HSB is in next section.

L201: It would be a great result, if you could show that there is no serpentinite involvement in the LSB/HSB relative to mature arcs.

Yes, and there is none. Please see the discussion in the text (Lines: 215-230).

L216: Again, here is the problem that your HSB samples (according to your Fig. 1) are not correlated with the HSB stratigraphic group - thus, to which extent are the HSB only 'inherent variability' of the LSB/HSB package that was emplaced at the same time?

Please see reply to the “**Overall response to reviewers**” for the reason why we can conclude the HSB was erupted later than the LSB.

L216: Which 'other tracers'? Neither in B/Nb nor Ba/Nb, and not even in $^{87}\text{Sr}/^{86}\text{Sr}$ the HSB are drastically different from LSB. Again, note that your HSB samples, are mixing with the LSB stratigraphic column.

Please see reply to the “**Overall response to reviewers**” for the reason why we can conclude the HSB was erupted later than the LSB.

The interpretations for other tracers are in the following sentences. Our explanation is based on fully understanding the mobility of the elements, and of the B-Sr-Pb-Nd-Hf isotopes of the boninites. There are clear $^{87}\text{Sr}/^{86}\text{Sr}$ and $\delta^{11}\text{B}$ differences between the LSB and HSB (Figure 3c), especially when they are linked to B/Nb and Ba/Nb. Please note that there is some inherited signature in HSB from the residual LSB mantle source, so some transitional character between these two suites of boninites is natural.

L224: I think what you want to say here that the HSB have more fluid input than the LSB. I agree, maybe a little more. But why does this 'little more' justify a separate group given the fact that you do not have a succession in time (according to Fig. 1)?

The important thing here is to consider the LSB/HSB package relative to other active settings that have also been analysed for the same elements. Maybe from that you can build a case that the early LSB/HBS sequence is unique to subduction initiation. But I do not think that the data are strong enough to argue for a stratigraphic/temporal trend + change of source + process.

Please see our reply in the “**Overall response to reviewers**” for the reason why we can conclude the HSB was erupted later than the LSB. The HSB form via melting

of mantle at least in part residual to LSB melting, so a greater slab input is required to flux HSB, and this comes from sediments.

The Expedition 352 suite is the most extensive, well-constrained suite of volcanic rocks ever obtained from *in situ* forearc crust (Shervais et al., 2020 G3; Ishizuka et al., 2020 JPet). There are no other B isotope papers related to subduction initiation.

L255: In fact, in Figure 3a and 3b you show that the HSB component is different from other arcs. You do not show (just discuss) that it is similar. And you contradict yourself in the next sentence...

As we illustrated in the text, the similarity relates to source (sediments + basalt) and the apparent mobility of the elements, not specifically the B/Nb and Ba/Nb ratios. Differences between the HSB slab input and the inferred high-T input for the mature arcs (i.e., low B/Nb low $\delta^{11}\text{B}$ endmember of Ryan and Chauvel 2014) reflect the differences in thermal structures and slab dehydration phenomena during subduction initiation as compared to a mature arc.

L269ff: The thermal evolution aspect is more wishful thinking than hard data. Why is the compositional evolution also a thermal slab evolution? How to you connect this

The subduction input to the LSB are melts from gabbros. HSB subduction inputs are fluids from (dominantly) sediments. The interpretation is in the following sentences.

L271: repetition

This and the later sentences answer your above question.

L273: Reference for the statement on temperature

This temperature is calculated according the Ba/Nb ((Line 293-294)).

L281: now you speak from lack of sediment melting, yet the HSB have both lower ϵ_{Nd} and ϵ_{Hf} than the LSBhow to you explain this without sediment melting?

The low ϵ_{Hf} signature of the HSB is inherited from the LSB residual source (Line 245-247). Oceanic gabbros can also have low ϵ_{Hf} . Our EPSL paper makes similar claims, for undifferentiated ocean crust. In this submission it is for gabbros. Hf-Nd isotopes in MORB show a wide range of variation.

The new Figure 6 shows that the HSB of Exp. 352 and Bonin Island all have much higher Cs/La than either sediments or Layer 2 basalt, indicating a lower-T subduction input during HSB generation ($< 840^\circ\text{C}$). The uniform Hf isotope signature of the HSB also preclude a sediment melt contribution.

L301: accretion would remove only some of the sediment (that's normal), and besides how do you explain the progressively lower ϵ_{Nd} and ϵ_{Hf} from FAB to LSB and HSB?

Accretion in a normal arc setting will only remove some of the sediment. Subduction initiation is not "normal", as described above. Given voluminous initial magmatism during initial plate foundering it may be difficult to get the whole plate to subduct, so only deeper parts of the subducting crust may be accessible to melting. We have explained this in the text. It is a combined effect of restricted mantle melting and time-changed subduction inputs.

L314: but you do NOT show this age gap at all in the data you present; instead there is temporal overlap

It's likely that there isn't a readily measurable age gap, given that dating results suggest that all these magmatic transitions may occur within $< 1\text{Ma}$. Reheating and alteration affect the boninite $^{40}\text{Ar}/^{39}\text{Ar}$ ages significantly (Reagan et al., 2019 EPSL), so age distinctions between the LSB and HSB are not yet defined. Reagan et al (2019) concluded the LSB erupted at $\sim 51.3\text{ Ma}$, while HSB erupted at $\sim 50\text{ Ma}$ (Reagan et al., 2019 EPSL; Figure 5). The stratigraphic information presented in the two new manuscripts show that HSB erupt later than the LSB. HSB eruptions on the Bonin

Islands began at 49.4 Ma (Ishizuka et al., 2006; Table S2), implying an initial westward migration of the embryonic arc. All these have documented that the HSB should erupt later than the LSB. Please see reply to the “**Overall response to reviewers**” for detailed explanation.

L323: the term ‘subduction channel’ should be used as defined. This ‘new’ definition of subduction channel in the context of this paper does not make sense.

The term “subduction channel” is widely used in studies of subduction zone metamorphism as the interface between the plates, where mélange assemblages of upper and lower plate materials develop. That’s the definition we’re using. We’re not sure what other definition the reviewer is referring to.

REVIEWER COMMENTS

Reviewer #1 (Remarks to the Author):

Comments on: Boron isotopes in boninites document rapid changes in slab inputs during subduction initiation

The authors have again revised the manuscript considerably since the last iteration. The introduction section now covers more information on the stratigraphy of the boninite holes, providing more confidence in the rapid geochemical changes discussed. In addition, more context on how the new B isotope data fits within the already extensively published radiogenic isotope data from the same holes is provided in both the introduction and discussion of the refined model. These improvements do a great job of clarifying the overall motivations of the study. More explanation of the figures has now been included in the main text which further clarifies the logic of the paper throughout. Therefore, much of the concerns that myself and the other reviewers had, have now been addressed. I now believe that the manuscript is ready for publication within Nature Communications.

I only have a few very minor comments relating to the Figures:

Figure 1: Explanation of this stratigraphy is now discussed in the introduction, providing greater clarity for Fig. 1. The yellow circles (LSB) and green squares (HSB) should be included within the key/ Currently, the reader needs to refer to Figure 2 to understand what these symbols represent.

New Figure 6: The red dashed lines (temperature estimates?) need to be explained within the figure caption. Chichijima and Mukojima HSB are not referred to anywhere else in the manuscript, so I suggest grouping these as Bonin Island HSB within the figure for consistency with the text.

I look forward to seeing this manuscript in its published form.

Sincerely,

George Cooper

Reviewer #3 (Remarks to the Author):

I have seen this manuscript the third time. There are a few interesting points in this manuscript, but the problem remains that the main points are poorly carved out, and that - while many things are said and discussed - those that would be important lack depth. The text says "...allow us to document rapid temporal changes in the slab inputs to the sources of IBM boninitic magmas, from lower crustal melts to upper crustal fluids, and to begin to reconstruct the thermal evolution of the slab in the earliest stages of subduction".

So I looked for discussion on the time frame (in the responses but not in the text), for link to the slab input (poorly constrained and speculative) and how the thermal evolution was reconstructed (should be much better evolved). Again, I found the manuscript unsatisfactory. There are questions about the B isotope fraction (now well documented during shallow subduction) that are completely ignored. The text is long winded, difficult to follow sometimes, repetitive in some parts, and often full of speculation where there should be more facts.

Reviewer #4 (Remarks to the Author):

This manuscript presents an interesting B isotope data set as well as supporting trace element and radiogenic isotope data from boninites from IODP Hole 352, which sampled the Izu-Bonin fore-arc to investigate subduction initiation. The data is used to determine the source of slab-derived material added to the magma sources, and temporal changes therein. Two types of boninite are

distinguished, LSB, which erupted first and shows evidence of melt derived from gabbroic subducted crust, and HSB, which shows the signature of fluids (and melts?) from basaltic crust + marine sediments. This sequence sets constraints on the thermal and physical evolution of the nascent subduction system.

This is a revised manuscript that appears to have gone through review at least twice. The manuscript also includes the rebuttal of the authors to the comments of the previous round of reviews. I read those before the actual manuscript, and found that the authors did an excellent job rebutting the comments which appear to have centred on the eruptive sequence. However, I had several issues with the manuscript itself, which raised a lot of questions and therefore leaves the authors' interpretations open to criticism. I am well aware that new rounds of reviews can be frustrating particularly if they contradict earlier reviews or argue about other aspects that weren't commented on before. I agree with the authors' model in broad terms, but I think there are several aspects that need clarifying and perhaps some re-interpretation.

I've summarised my main comments in the attached review, whereas detailed comments are made in an annotated version of the manuscript word document.

Review of Li et al.

This manuscript presents an interesting B isotope data set as well as supporting trace element and radiogenic isotope data from boninites from IODP Hole 352, which sampled the Izu-Bonin fore-arc to investigate subduction initiation. The data is used to determine the source of slab-derived material added to the magma sources, and temporal changes therein. Two types of boninite are distinguished, LSB, which erupted first and shows evidence of melt derived from gabbroic subducted crust, and HSB, which shows the signature of fluids (and melts?) from basaltic crust + marine sediments. This sequence sets constraints on the thermal and physical evolution of the nascent subduction system.

This is a revised manuscript that appears to have gone through review at least twice. The manuscript also includes the rebuttal of the authors to the comments of the previous round of reviews. I read those before the actual manuscript, and found that the authors did an excellent job rebutting the comments which appear to have centred on the eruptive sequence. However, I had several issues with the manuscript itself, which raised a lot of questions and therefore leaves the authors' interpretations open to criticism. I am well aware that new rounds of reviews can be frustrating particularly if they contradict earlier reviews or argue about other aspects that weren't commented on before. I agree with the authors' model in broad terms, but I think there are several aspects that need clarifying and perhaps some re-interpretation. In addition, I think the structure is letting it down, making it repetitive and sometimes contradictory.

At the bottom of this review is a list of detailed comments I made whilst going through the manuscript. I'll summarize my main comments here:

Assumption that heterogeneity of mantle source is transferred to arc magma source and even persists after melting.

An recurring argument by the authors is the similarity of 'data patterns' of their samples with certain slab components (meaning large or small spread in B isotopes). By doing this the authors assume that heterogeneity of a slab component would result in a large variation in basalt compositions that are metasomatised by fluids of melts from such a source. This assumes heterogeneity persists during melting/dehydration (processes that will strongly reduce heterogeneity, as heterogeneity is sample volume related, and melts/fluids will represent completely different volumes than the cited literature core samples), then persist during mantle source metasomatism and partial melting (possible but not likely).

More likely, large variations in samples represents metasomatism of the source by multiple sources and a large sensitivity to small variations therein. In the case of LSB, this could potentially be due to small but variable amounts of serpentinite-derived fluid (see detailed comments, supported by correlation between Sr and B isotopes for these samples, which is not properly evaluated by the authors).

Boron isotope fractionation during dehydration has been poorly taken into account.

Boron isotope fractionate strongly during dehydration, as $^{11}\text{B}/^{10}\text{B}$ in the fluids is higher than in the residue, and note that this fractionation is temperature dependent. This is completely ignored in the first section where the authors compare slab component with magma

compositions, and therefore these comparisons are incomplete. This won't matter much for melting or dehydration of serpentinite, but it does for dehydration of altered crust and sediments (as proposed for HSB). Thus, in the mixing models, fluids from each source should also be evaluated to rule them out (or in) as a potential source. Finally, bulk mixing as the authors adopt in their models also assumes no fractionation during melting of trace elements. This is technically incorrect; this fractionation is indeed limited but cannot be completely ignored. Therefore, using DM as an endmember is also problematic, if the source has undergone previous melt extraction. The modeling would be more authoritative if the authors used a more realistic model (adding melts and fluids derived from various subducting reservoirs to a strongly depleted mantle source, then extract partial melts from this metasomatised source), especially if they want to go in such detail as in Fig. 5, where they aim to demonstrate a prior B enrichment event.

Furthermore, the effect of metamorphism (i.e., dehydration) prior to melting of crust is not discussed. Rocks don't just heat up and then melt, they undergo dehydration reactions first. Fluid will be extracted and this will fractionate B isotopes, leaving a residue with a lighter B isotope signature. These fluids may induce melting elsewhere in the slab depending on the thermal profile. They may mix with slab-derived melts. They may escape first. We know very little about this process, but it is very unlikely that slab melts will have a B isotope composition identical to the low-T protolith. This needs discussing.

Incomplete and contradicting explanation of origin of HSB.

The authors are rather vague about HSB, 'the upper sections of ocean crust' (l. 255) without specifying which parts and mentioning hydrous fluids elsewhere but without taking into account B isotope fractionation in the generation of these fluids. Also, eHf appears to indicate there needs to be a significant melt component (it's more extreme than LSB compared to front-arc basalts), but this doesn't appear to be part of the modelling. Then further in the manuscript there's a model that assumes 3-5% dehydration of 95% AOC + 5% sediment (l. 312), but later only mention additional contributions from subducting sediments (l. 328). It's all a bit vague. Both eHf and B isotopes need to be explained by the model (taking into account B isotope fractionation during dehydration), and the amount of melt vs. fluid and the source of these needs to be clarified.

This is important as HSB is actually more unique in terms of d_{11B} - B/Ce - B/Nb systematics compared to global arc volcanics than LSB, despite the authors suggesting LSB is more unique (it isn't, not even in their own Fig 3, where LSB partially overlaps with Mariana volcanic front, whereas HSB is uniquely low in d_{11B} for its B/Nb). A strong sediment component is indeed likely and in fact involvement of a Layer 2 basalt component seems hard to prove.

Structure can be improved.

The authors start the discussion with a section on possible slab components, but this whole section seems redundant as they explore this in more detail in the following sections, and also they assume whole-scale mixing and ignore B isotope fractionation during dehydration, so the discussion is incomplete. They even admit as much in their last two sentences.

The section on HSB is very short compared to the detailed discussion of LSB, but a

subsequent section (Rapid changes in slab flux from LSB to HSB reflects a cooling slab) is essentially about generation of HSB again. It would make more sense to integrate those sections. The cooling slab angle isn't very clear at all. I think rearranging and removing repetition would greatly improve the clarity of the manuscript.

In summary, I think the paper presents a really interesting high-quality dataset which puts some important constraints on the process of subduction initiation, which is worthy of publishing in Nature Comms. However, it needs some significant revisions, in response to my comments above and further detailed comments below, and probably some tweaking of the interpretations as a result.

Cees-Jan De Hoog

University of Edinburgh

Detailed comments (copied from annotated version of manuscript, where they can be found pointing to the relevant part of the manuscript)

- I think Gervais et al 2021 make some quite clear statements about this (amphibolite melt for LSB, hydrous sediment melt for HSB), so it's good to make clear what the current interpretation is, where it is still lacking, and if you disagree with previous findings.
- Not sure why in this figure some literature data is plotted as individual samples (MAR srp), others as (rather distracting) arrows (L3), others with an error bar in x but not y (avg sed), and not even consistent per panel. Fields would probably be better, one could still indicate range of d11B falling of the scale. Detailed Sr isotope inset is distracting and unnecessary .
- This is an artificially small variation, based on an integrated average of composite samples, so cannot be used to compare with the small range of HSB (as you do in l. 124). The actual range is -2.5 to +3.8, still not a big range but more than the samples. Same is true for sediments. Plotting averages here and saying the range is small, and use that as an argument, and plotting individual samples for other sources, and use that as an argument for a large range, seems biased. Moreover, it isn't even a valid argument. Surely large-scale extraction of material from the slab either as melt or fluid and subsequent metasomatism of the mantle source would eradicate the heterogeneity of the slab component.
- But range from -13 to +26. I don't see how this is consistent with the limited variation in HSB (l. 123-125). Also, how was this average derived? Tonga sediments are +2±4, Japan marine sediments are also positive (+4 but large range), only continental detritus has negative values (unlikely to be present during intra-oceanic subduction initiation), SSI is negative but highly variable. Your average and stdev in Fig 3 masks a lot of the true variability. Some evaluation of the type of sediment entering the subduction zone is needed.
- There's in fact a positive correlation between Sr and B isotopes for LSB, which is at least as 'strong' as the correlation with eHf you mention later (neither being particularly strong)

- In terms of d_{11B} , fluids from altered basalt would do the job perfectly. Why do you only consider whole-scale mixing? Any discussion of possible sources in relation to B isotopes is incomplete without considering fluids.
- On the Sr-B isotope diagram HSB to LSB is a continuum, and most of the LSB can be modeled as a mix of L2A and L3 gabbro considering the range in both parameters for both sources. So only **some** LSB would require those additions. You actually say this in l. 144 so you are contradicting yourself somewhat.
- Data for hole 1415 also in figure, so not just 735B? Also, how about Troodos and Oman data from the same paper?
- Sure, the pattern is similar, but that doesn't mean anything unless you have a model that explains how you can transfer the heterogeneity of the slab to the mantle source region and preserve it during melting
- And more importantly, its isotopic signature!
- Which then begs the question, what's the point of the previous discussion? Why not incorporate these considerations right away?
- It's a real pity that there is no B isotope data for these. It would go a long way in alleviating any worries about seawater alteration effects on d_{11B} (as one would expect them to be MORB-like), as well as provide a starting point for any mixing model involving B isotopes.
- Is that a thing? Isn't wet basalt melting curve at much higher T than amphibolite facies? Looking at the model, it appears the authors simply mean amphibolite melting, the residue being a granulite, and conditions in fact granulite facies. This may seem pedantic, but it's confusing terminology.
- It's unclear what the red arrows in the figure represent, melting models? Would we expect straight lines? And then why two lines from each colour, are these different source compositions? But then why do you have different sources for low and high-T melts? Please clarify and if these are just lines drawn by eye, please replace with melting/mixing models.
- Although I can see the argument for this, I do want to point out that the trace element modeling is highly dependent on the mineral assemblage of the source rock, which is assumed to be amphibolite with only plagioclase and amphibole, but this is not entirely consistent with the melting conditions, and ignores the possible presence of garnet and/or mica in the source. This needs some discussion.
In addition, the model is incomplete. You simply mix a melt with DM. But this gives you the source. Melting needs to be included, and also, DM is a poor choice if considerable melt extraction has occurred prior (as shown by Shervais et al for these rocks). These points may not make much difference but we can't evaluate that.
- amphibolite melting (see comment earlier, it's not amphibolite facies)
- where are the details of this model (mineralogy, D values, etc), and why can't these be

plotted on the figures instead of as insets?

- One would probably expect more of a continuum, the slab doesn't melt at two distinct temperatures, first of all there will be temperature differences within the slab and also, it's a dynamic process as the slab is heating. Also, the correlation between 'less altered' and 'low d11B' is tenuous, L2 basalts are highly altered and have low d11B. In fact, these samples look a lot like L2 basalts in terms of d11B and Ba/Nb. So one could equally argue these are melts from L2 basalts that were less altered, so their Sr isotopes are not as high as other L2 basalts.
- I think we're missing a step here, as the boninites are not simply a mixture between slab-derived melt and DM, but their source is. So any melting effects are ignored. This may be true for d11B but could affect Ba/Nb (especially if there's amphibole in the source, which seems likely when melts fertilize depleted mantle). If you want your model to be quantitative enough to draw conclusions about d11B of the source, as you do below, the model needs to be as accurate as possible (i.e., reflect the natural processes you think that are happening)
- This again assumes that the heterogeneity of the slab is reflected in the erupted melts. This is a huge leap of faith, if you think of all that happens inbetween.
- This all seems rather speculative and it's unclear how small changes in other parameters would affect this (Ba/Nb, melting T, also earlier comment about ignoring melting fractionation and possible amphibole in the source). Also, a fluid liberated from the crust would have a positive d11B value (e.g. King et al, 2007) as well as high B and completely overprint the DM B isotope value if it resulted in a 8x higher B content than DM. So I don't see how you can shift DM only a few permil this way.
- The effect of this on B isotopes is actually completely ignored. Rocks don't just heat up and then melt, they undergo dehydration reactions first. This will fractionate B isotopes. These fluids may induce melting elsewhere in the slab depending on the thermal profile. They may mix with melts. They may escape first. We know very little about this process. It may explain some or even much of the variability in B isotopes you are seeing, but there's little to no research done on this.
- Are you sure? Looking at your mixing lines between serpentinite and L2 basalt in Fig 3, it's clear you can increase d11B considerably before increasing $^{87}\text{Sr}/^{86}\text{Sr}$ and with only limited increase in B/Nb. As such, you could explain much of the variation in d11B in LSB by taking a low d11B - low $^{87}\text{Sr}/^{86}\text{Sr}$ LSB sample and add serpentinite fluid. Note that there's actually a correlation between B and Sr isotopes for LSB in support of such a scenario.
- I am not an expert in Hf isotopes. But can this not simply be the DM signature? Or does DM have a high epsilon? In that case I understand your dilemma, you need to explain why HSB has a stronger melt signature than LSB, yet also a fluid signature, which must eradicate some of the melt signature characteristics of LSB (e.g. reduce range of d11B). Please clarify how HSB gets such low eHf.

- So you are proposing a mixture of Layer 3A melts and Layer 2A fluids? The fluids must have had high Sr/Nd (as Sr/Nd is much lower than expected based on the eHf trend), no Ba (as Ba/Nb falls on eHf trend) and enough B to overprint the LSB melt B isotope signature. Thus the fluid (not the rock it came from) had d11B of 0-2 permil. Does this still fit the model? Note that fluids derived from a rock will have a different signature than the rock itself. This doesn't appear to be taken into account. The model could be correct, but it's not properly evaluated and d11B not properly discussed.
- I'd probably say fore arc mantle wedge, as you do in the next sentence. It may indeed end up in the subduction channel, but it doesn't have to (it can be brought beneath the arc by corner flow). Any serpentinites in the subduction channel will probably have interacted with sediments and have low d11B, as demonstrated by Martin et al 2020.
- By this point I got curious how different it was from the De Hoog and Savov database, and yes, they have rather low B, but in terms of B/Nb they aren't unique at all, in the Izu arc we find similar values (and note they overlap Mariana VF in your own plot). B/Ce is normal as well. So low B is simply due to high degrees of melting. They are rather unusual in have very low Ba/Nb though. Also, La/Sm is low, certainly for the low d11B ones. In fact, it appears to trend to MORB, perhaps again an indicator of high melt fractions compared to typical arc basalts.
The ones really standing out from the global arc database are HSB, which have high B/Nb as well as B/Ce for their (low) d11B.
So perhaps you want to modify your statements here. It appears for LSB the slab-derived component is different, but the actual melt composition not that much from at least some regular arc basalts (in terms of B/Nb and B isotopes at least), whereas HSB is quite different despite the involvement of similar components. Note that the low d11B of HSB was not properly explained in the previous section.
- This section is essentially part of generation of HSB. It would make sense to include it in the earlier section on that topic.
- than LSB?
- As far as I understood, HSB also has a strong slab melt component (evidenced by low eHf) with an additional fluid signature. Does this model reflect that? As this makes it sound like it is fluid only?
- Important point, which wasn't taken into account earlier during generation of LSB. More importantly, it will also fractionate B isotopes.
- Note that B/Nb is actually quite high for the low d11B values (compared to global arcs). The B isotope signature requires more explanation than B/Nb.
- I thought it was mostly Layer 2A basalts? Or do you refer to the 5% sediment in the model in the previous section? I am getting very confused. I agree that sediments are likely but your arguments or wording are inconsistent. Also, I'd add 'fluids from'
- I am not convinced the variation in d11B of LSB isn't the result of small amounts of

serpentinite fluid to another component with a value of about 0 to +2 (consistent with correlation with Sr isotopes and Ba/Nb and B/Nb systematics). It's certainly a valid alternative to your explanation based on the heterogeneity in d_{11B} of L3 gabbro. It doesn't make the rest of your story less valid, as you say earlier, abyssal serpentinites would dehydrate at lower T.

- Why are the two plates a different thickness? Aren't they essentially the same plate, which breaks and then one part sinks?
- What is the significance of these?
- I thought you didn't need fluids. You modeled melts alone specifically.
- The overlying L2 basalts are more altered and closer to the hot upwelling mantle. Why wouldn't these melt first? Or are you suggesting these were eroded in the proto-accretionary wedge? Some explanation is needed.
- Note again they have very anomalous low d_{11B} for their high B/Nb. They are not similar to other arcs.
- From when (do we observe typical arc basalts)? The timing of this is of interest.
- Do you mean melting reactions?
- I assume that's from Wolf and Wyllie?

DETAILED RESPONSES TO THE REVIEWERS

Reviewer #1: (Remarks to the Author):

Comments on: Boron isotopes in boninites document rapid changes in slab inputs during subduction initiation

The authors have again revised the manuscript considerably since the last iteration. The introduction section now covers more information on the stratigraphy of the boninite holes, providing more confidence in the rapid geochemical changes discussed. In addition, more context on how the new B isotope data fits within the already extensively published radiogenic isotope data from the same holes is provided in both the introduction and discussion of the refined model. These improvements do a great job of clarifying the overall motivations of the study. More explanation of the figures has now been included in the main text which further clarifies the logic of the paper throughout. Therefore, much of the concerns that myself and the other reviewers had, have now been addressed. I now believe that the manuscript is ready for publication within Nature Communications.

I only have a few very minor comments relating to the Figures:

Figure 1: Explanation of this stratigraphy is now discussed in the introduction, providing greater clarity for Fig. 1. The yellow circles (LSB) and green squares (HSB) should be included within the key/ Currently, the reader needs to refer to Figure 2 to understand what these symbols represent.

The key has been added in the revised Figure 1.

New Figure 6: The red dashed lines (temperature estimates?) need to be explained within the figure caption. Chichijima and Mukojima HSB are not referred to anywhere else in the manuscript, so I suggest grouping these as Bonin Island HSB within the figure for consistency with the text.

The explanations for the red dashed lines have been added in the Figure caption. The Chichijima and Mukojima HSB have been grouped as Bonin Island HSB in the

revised Figure 6.

I look forward to seeing this manuscript in its published form.

Sincerely,

George Cooper

Reviewer #3: (Remarks to the Author):

I have seen this manuscript the third time. There are a few interesting points in this manuscript, but the problem remains that the main points are poorly carved out, and that - while many things are said and discussed - those that would be important lack depth. The text says "...allow us to document rapid temporal changes in the slab inputs to the sources of IBM boninitic magmas, from lower crustal melts to upper crustal fluids, and to begin to reconstruct the thermal evolution of the slab in the earliest stages of subduction". So I looked for discussion on the time frame (in the responses but not in the text), for link to the slab input (poorly constrained and speculative) and how the thermal evolution was reconstructed (should be much better evolved). Again, I found the manuscript unsatisfactory. There are questions about the B isotope fraction (now well documented during shallow subduction) that are completely ignored. The text is long winded, difficult to follow sometimes, repetitive in some parts, and often full of speculation where there should be more facts.

We have made further detailed corrections that hopefully address these issues. B isotope fraction has been fully discussed in the revised text.

Please see the following sections in the revised text:

Boron systematics in subduction versus subduction initiation settings.

Slab sources for boninite boron in subduction initiation settings.

Reviewer #4: (Remarks to the Author):

This manuscript presents an interesting B isotope data set as well as supporting trace

element and radiogenic isotope data from boninites from IODP Hole 352, which sampled the Izu-Bonin fore-arc to investigate subduction initiation. The data is used to determine the source of slab-derived material added to the magma sources, and temporal changes therein. Two types of boninite are distinguished, LSB, which erupted first and shows evidence of melt derived from gabbroic subducted crust, and HSB, which shows the signature of fluids (and melts?) from basaltic crust + marine sediments. This sequence sets constraints on the thermal and physical evolution of the nascent subduction system.

This is a revised manuscript that appears to have gone through review at least twice. The manuscript also includes the rebuttal of the authors to the comments of the previous round of reviews. I read those before the actual manuscript, and found that the authors did an excellent job rebutting the comments which appear to have centred on the eruptive sequence. However, I had several issues with the manuscript itself, which raised a lot of questions and therefore leaves the authors' interpretations open to criticism. I am well aware that new rounds of reviews can be frustrating particularly if they contradict earlier reviews or argue about other aspects that weren't commented on before. I agree with the authors' model in broad terms, but I think there are several aspects that need clarifying and perhaps some re-interpretation. In addition, I think the structure is letting it down, making it repetitive and sometimes contradictory.

Main Reply #1: We sincerely thank the reviewer for his constructive comments and his concrete criticisms. We have reorganized the text so as to address the reviewer's concerns about repetitiveness, as well as to try frame out for the reader key aspects about the systematics of boron during subduction.

We now start the discussion with a short section entitled "Boron in Subduction versus Subduction Initiation settings", which seeks to frame out the important slab reservoirs for B as observed in volcanic arcs, highlighting the role of serpentinites in the subduction recycling of boron, be they sited in the slab lithosphere, the subduction, or as a constituent in subducting ocean crust (e.g., Cooper et al 2020). Subduction initiation settings, as outlined by Stern and Bloomer (1992) and Pearce et al (1992), complicate the involvement of serpentinites because of inherently high temperature,

low pressure conditions involving the upwelling of hot asthenospheric mantle in response to slab foundering. Boninite primary melts are at $>1200^{\circ}\text{C}$ at $< 1\text{GPa}$ (Whattam et al 2020; Shervais et al 2021), supporting unusually hot conditions near the plate interface, and the slab contributions to these magmas are different as well: Pearce et al. (1992) and Li et al (2019) call on melt additions from subducting ocean crust under hornblende-bearing granulite facies conditions ($900\text{-}950^{\circ}\text{C}$) to explain observed enrichments of Zr and Hf in boninites, as well as lower ϵHf_i signatures that correlate positively with measures of Hf enrichment (e.g., Sm/Hf and Ti/Hf). Serpentine minerals break down at $< 700^{\circ}\text{C}$ (Ulmer and Trommsdorf, 1995), so serpentinites are unlikely to be present along the plate interface, and given the likely temperatures in the downgoing plate ($>850^{\circ}\text{C}$, based on both recent modeling and on results from the metamorphic sole of the Oman ophiolite (e.g., Hacker and Mosenfelder 1996; Zhou and Wada, 2021)) serpentine in the slab may decompose well before the fluid/melt releases that appear to flux boninitic magmatism.

If serpentinites were a significant component in the slab source during subduction initiation, their decomposition near the slab-mantle interface or in the shallow slab lithosphere should result in the addition of a high B/Nb, Ba/Nb, Sr/Nd and $\delta^{11}\text{B}$ fluid component to the boninite mantle source before the slab becomes hot enough to melt. However, as shown in our Figure 4a-b, Ba/Nb and Sr/Nd in Expedition 352 low-silica boninites do not include an obvious aqueous fluid slab component, and no positive correlations are observed among B/Nb, Ba/Nb and $\delta^{11}\text{B}$ for the Expedition 352 LSB and boninite samples as a whole (Figs. 3a-b). High-silica boninites show high B/Nb and Ba/Nb, but relatively low $\delta^{11}\text{B}$ values, inconsistent with serpentinite dehydration as a dominant boron recycling mechanism, as is the case in mature subduction zones. These observations would all seem to indicate that serpentinite, be it from the slab lithosphere or crust, or the forearc mantle is unlikely to be major source for boron enrichment in the Expedition 352 boninites.

Figure A1. Ba/Nb (a) and Sr/Nd (b) versus ϵHf correlations for the Mariana arc basalts. Data according the compilation of Straub (2017). In (b), samples with MgO > 7% are selected. These diagrams are used to illustrate differences in elemental mobility between mature arc and subduction initiation (Fig. 4 a-b).

This said, the variability in the LSB data, and some of the LSB data patterns in Figure 3 are consistent with variably low level inputs from a high B and high $\delta^{11}\text{B}$ source, that in terms of its Sr isotopic signatures could be consistent with either abyssal serpentinites, or Marianas-like subduction channel serpentinites. While there is considerable B isotopic variability exhibited in the small body of data available for oceanic gabbros, we agree with the reviewer that it is unlikely that such variability could be carried from the trench into a mantle source, especially if the slab inputs responsible for low-silica boninite genesis are melts, as data for Zr and Hf isotopes suggest (see our response below on this topic). Understanding how this kind of variability might exist in the LSB is one of the peculiar challenges of this dataset, and prompted by the reviewer's comments we think we can now propose a sensible way to do this without the potential complications of serpentinite involvement.

At the bottom of this review is a list of detailed comments I made whilst going through the manuscript. I'll summarize my main comments here:

Assumption that heterogeneity of mantle source is transferred to arc magma source and even persists after melting.

An recurring argument by the authors is the similarity of 'data patterns' of their samples with certain slab components (meaning large or small spread in B isotopes). By doing this the authors assume that heterogeneity of a slab component would result in a large variation in basalt compositions that are metasomatised by fluids of melts from such a source. This assumes heterogeneity persists during melting/dehydration (processes that will strongly reduce heterogeneity, as heterogeneity is sample volume related, and melts/fluids will represent completely different volumes than the cited literature core samples), then persist during mantle source metasomatism and partial melting (possible but not likely).

More likely, large variations in samples represents metasomatism of the source by multiple sources and a large sensitivity to small variations therein. In the case of LSB, this could potentially be due to small but variable amounts of serpentinite-derived fluid (see detailed comments, supported by correlation between Sr and B isotopes for these samples, which is not properly evaluated by the authors).

Main Reply #2: We largely agree with the reviewer on this point, as the processes by which fluids or melts are extracted from slab materials should serve to produce a homogenized extract. Evidence in our data for multiple slab melting episodes (e.g., low and high Ba subsets of the LSB) might permit some source-related heterogeneity to carry through, but even this should be limited. In our revision we have separated our description of the likely slab sources from our discussion of the B isotope signatures of the different boninite types to try and improve clarity here.

The reviewer has suggested from our Figure 3 the idea of a low-level input from something like abyssal serpentinites to explain the LSB data patterns. We can't entirely rule this out, and now say as much in our revision, though the difficulties of preserving serpentinites in a setting as hot as the subduction initiation setting appears to be is problematic. However, our continued examination of the available literature raises the possibility that metasomatic redistribution of boron within the subducting crust may be capable of producing the $\delta^{11}\text{B}$ range that we see: Ishikawa et al (2005) and Prigent et al (2018) report on boron, mobile element, and $\delta^{11}\text{B}$ signatures in amphibolites of the metamorphic sole of the Oman ophiolite that show B isotope

diversity and a mean $\delta^{11}\text{B}$ signature similar to that of the LSB. As our data do not permit us to easily resolve between a lithospheric serpentinite or intra-crustal metasomatic origin for the $\delta^{11}\text{B}$ variation in the LSB, in our revision we offer both as possibilities, while preferring the crustal origin hypothesis.

Boron isotope fractionation during dehydration has been poorly taken into account.

Boron isotope fractionate strongly during dehydration, as $^{11}\text{B}/^{10}\text{B}$ in the fluids is higher than in the residue, and note that this fractionation is temperature dependent. This is completely ignored in the first section where the authors compare slab component with magma compositions, and therefore these comparisons are incomplete. This won't matter much for melting or dehydration of serpentinite, but it does for dehydration of altered crust and sediments (as proposed for HSB). Thus, in the mixing models, fluids from each source should also be evaluated to rule them out (or in) as a potential source. Finally, bulk mixing as the authors adopt in their models also assumes no fractionation during melting of trace elements. This is technically incorrect; this fractionation is indeed limited but cannot be completely ignored. Therefore, using DM as an endmember is also problematic, if the source has undergone previous melt extraction. The modeling would be more authoritative if the authors used a more realistic model (adding melts and fluids derived from various subducting reservoirs to a strongly depleted mantle source, then extract partial melts from this metasomatised source), especially if they want to go in such detail as in Fig. 5, where they aim to demonstrate a prior B enrichment event.

Main Reply #3: The effects on boron isotope fractionation during slab metamorphism and slab dehydration/melting are now more fully discussed. The B isotope system is unusual among stable isotopic systems in that fractionation of the isotopes not only vary with temperature, but also with system pH. The pH effect on B isotope signatures can be profound, in that in systems at pH values near that of seawater very large (>30‰) fractionations can occur at low temperatures, while at either very high or very low pH values, B isotopic fractionation is precluded, because all the boron in the system will be speciated as $\text{B}(\text{OH})_4^-$ or $\text{B}(\text{OH})_3$, respectively.

This phenomenon is seen in the Mariana forearc serpentinite seamounts, wherein serpentinite and porefluid pH's are essentially identical at fluid pH values of 11 to 12.5 (e.g., Benton et al 2001; Savov et al 2004). Under the much higher temperature conditions of subduction initiation we cannot easily assay system pH, though there is some experimental evidence for higher T systems biasing toward one B species (e.g., Hervig et al., 2002; Wunder et al., 2005; Kowalski and Wunder, 2018).

In our revision, we note the potential likely fractionations of B isotopes as constrained by the discussed temperatures of the system as potential maxima. B isotope fractionation during mantle melting will be minimal as the boninites are high T, low P melts of very depleted mantle (e.g., Pearce and Reagan, 2019). The estimated temperature for slab melting to induce LSB generation (900°C) could induce at most 5‰ of $\delta^{11}\text{B}$ fractionation between the melt and residual slab. Given mass balance constraints the melt would have a $\delta^{11}\text{B}$ at most ~2‰ higher than the slab material before melting based on our calculations. Thus, a slab component with $\delta^{11}\text{B}$ between -1.5‰ and +6.5‰ will fit the LSB data of this study, based on Figure 5a. The temperatures of slab dehydration to induce HSB generation (~800°C) could induce at most 6‰ of $\delta^{11}\text{B}$ fractionation between the fluid and residual slab. Given the lowest HSB $\delta^{11}\text{B}$ of ~ -0.2‰, the dehydrated slab could have a $\delta^{11}\text{B}$ as low as -6 ‰, which is in the range of a mixture of subducted sediments and basalts.

We recognize the model depicted in Figure 5 of mixing between gabbro-derived slab melts and the mantle was a weakness in our earlier manuscripts. This was partly due to trying to mix to a MORB-like depleted mantle source, which the reviewer correctly calls into question. It is clear from the compositions of the Izu-Bonin FABs documented in Shervais et al 2019 that the source mantle was not DM, and recent results on other mantle-derived oceanic rocks (e.g., Walowski et al 2019; 2021) demonstrate that mantle reservoirs exist that preserve both lower and higher $\delta^{11}\text{B}$ signatures than the average MORB signature of Marschall et al (2017). Also, as suggested by Shervais et al. (2021), adding 0.1% of slab melt to the FAB residual source will fully overprint its Hf-Nd isotopic signatures as this mantle source is so low in Hf, Nd and other immobile lithophiles that it becomes impossible to accurately

define the mantle elemental endmembers (e.g., Ba/Nb and Sr/Nd) in plots versus ϵ_{Hf} for the LSB. Given this, we think that using the DDM values of Workman and Hart (2005) as a starting point is reasonable, although the actual Izu-Bonin LSB mantle source may be more depleted. The original Figure 5 and related text have been modified to better depict the strengths and limitations of our LSB model.

Furthermore, the effect of metamorphism (i.e., dehydration) prior to melting of crust is not discussed. Rocks don't just heat up and then melt, they undergo dehydration reactions first. Fluid will be extracted and this will fractionate B isotopes, leaving a residue with a lighter B isotope signature. These fluids may induce melting elsewhere in the slab depending on the thermal profile. They may mix with slab-derived melts. They may escape first. We know very little about this process, but it is very unlikely that slab melts will have a B isotope composition identical to the low-T protolith. This needs discussing.

Main Reply #4: In the revision, we have more fully considered the effects of metamorphism on the B/Nb, Ba/Nb and Sr/Nd elemental ratios and on the $\delta^{11}\text{B}$ composition of the slab. In mature subduction zones, $\delta^{11}\text{B}$ in altered oceanic crust near the slab-mantle interface will significantly decrease due to the early loss of isotopically heavy B to the forearc mantle at shallow depths. This is directly documented in the low $\delta^{11}\text{B}$ values ($-6 \pm 4\%$) for entrained mafic blueschists recovered from serpentinite muds in the Mariana forearc, which reflect metamorphism at ~19 km depths and 200-350°C (Pabst et al., 2012). In subduction initiation, the slab thermal structure is very different, and the slab-mantle interface can reach temperatures of ~ 825°C or more at 15-20 km slab depths, with temperatures of ~ 500°C or more at 150 m beneath the interface, based on data from the Oman metamorphic sole (Hacker and Mosenfelder, 1996) and recent modeling by Zhou and Wada (2021). Thus, amphibolite-derived fluids can extract B from the slab from a few hundred of meters depth. Other fluid-mobile elements (e.g., B, Rb, K and Ba) are highly elevated in the Oman metamorphic sole amphibolites from 0 to 30 m below the

slab-mantle interface, suggesting equilibration with deeper derived, FME rich fluids during prograde metamorphism of the slab (Ishikawa et al., 2005). These mafic amphibolites have $\delta^{11}\text{B}$ from -2.3‰ to +10.8‰, with an average value of +3.75‰ (Prigent et al., 2018). Amphibolite fluids liberated from deeper in the crust at ~700°C can have $\delta^{11}\text{B}$ as high as +4.6‰ to +12.4‰, assuming an undehydrated amphibolite with $\delta^{11}\text{B}$ between -2.5 and +5.4‰ like upper crust basalts from DSDP Holes 417A, 417D and 418A (Smith et al., 1995; Hervig et al., 2002; Wunder et al., 2005). Infiltration by deeper-sourced amphibolite facies fluids explains the high $\delta^{11}\text{B}$ signatures in Oman metamorphic sole amphibolites. The high and variable $\delta^{11}\text{B}$ of the melted amphibolite documented in our LSB may be partly inherited from the gabbros before subduction (e.g., oceanic gabbros from ODP Hole 735B have high and variable $\delta^{11}\text{B}$ from -4.3‰ to +24.9‰; Smith et al., 1995; Hart et al., 1999) but will be strongly affected by fluid losses before melting, and subsequent interactions with amphibolite-derived fluids from deeper in the crust (see **Basalt/gabbro melt inputs to the LSB mantle source** section).

As the majority (> 70%) of slab boron is rapidly driven off from subducting sediments by mechanical compaction, diagenesis and prograde metamorphic dehydration at pressures < 1 GPa and temperature < 350°C in mature subduction systems (Savov et al., 2005, 2007; Bebout et al., 1993, 1999; Hulme et al., 2010), subducted sediments at ~ 0.5 GPa and ~ 300°C should have relatively low B contents (3.4 to 57.4 with average of 31.5 $\mu\text{g/g}$) and $\delta^{11}\text{B}$ (-9‰ to -11.9‰) (Nakano and Nakamura, 2001). Low B contents (3.6 to 24.1 $\mu\text{g/g}$) are also observed for subducted sediments in Oman metamorphic sole (Ishikawa et al., 2005). Bulk mixing calculations suggest the involvement of sediments in the slab must be less than 10 wt. % to be consistent with the highest $^{87}\text{Sr}/^{86}\text{Sr}_i$ of the HSB (Fig. 3c). Involvement of > 5 wt. % sediments in the slab will result in Pb isotopes being dominated by the sediment signature (Li et al., 2019a). These calculations also show that a mixture of 5-10 wt. % subducted sediments (B=31.5 $\mu\text{g/g}$; $\delta^{11}\text{B}$ =-11.2‰; Nakano and Nakamura, 2001) and 95-90 wt. % amphibolite such as seen at the slab-mantle interface of the Oman metamorphic sole (B=33.3 $\mu\text{g/g}$; $\delta^{11}\text{B}$ =+3.75‰; Ishikawa et al., 2005; Prigent et al.,

2018) will yield a $\delta^{11}\text{B}$ signature of +3.2% to +2.3%, much higher than our estimated slab input of $< -0.2\text{‰}$, or as low as -6‰ . This suggests that we may be overestimating the B and $\delta^{11}\text{B}$ in subducted basalts, and/or that they may not have been metasomatized by deeper source fluids before fluids were released to flux the HSB magmatism. This may reflect the cooling of the slab-mantle interface (see **Basalt and sediment fluid inputs to the high silica boninite mantle source** section).

Incomplete and contradicting explanation of origin of HSB.

The authors are rather vague about HSB, ‘the upper sections of ocean crust’ (l. 255) without specifying which parts and mentioning hydrous fluids elsewhere but without taking into account B isotope fractionation in the generation of these fluids. Also, eHf appears to indicate there needs to be a significant melt component (it’s more extreme than LSB compared to front-arc basalts), but this doesn’t appear to be part of the modelling. Then further in the manuscript there’s a model that assumes 3-5% dehydration of 95% AOC + 5% sediment (l. 312), but later only mention additional contributions from subducting sediments (l. 328). It’s all a bit vague. Both eHf and B isotopes need to be explained by the model (taking into account B isotope fractionation during dehydration), and the amount of melt vs. fluid and the source of these needs to be clarified.

Main Reply #5: We have revised our discussions of the HSB to try and clarify our model for their origins, which is founded in part on the results of Shervais et al (2021) and Li et al (2019). HSB magmas are derived from the melting of residual LSB source mantle, fluxed by fluid inputs from a mixture of the upper oceanic crust (slab basalts and diabases) and subducting sediments.

Higher B/Nb and Ba/Nb in the HSB are associated with lower $\delta^{11}\text{B}$ and ϵNd_i , and with elevated $^{87}\text{Sr}/^{86}\text{Sr}_i$, suggesting coherent behavior among B, Ba, Sr and Nd during HSB formation (Fig. 3; Supplementary Fig. S2). La/Sm ratios are near constant to slightly lower in the HSB, while Sr/Nd are higher (Figs. 4c-d), suggesting that the additional subduction additions required for the HSB are not melt dominated, as a

melt input should lead to higher La/Sm, as is seen in the LSB of this study and in mature arc lavas (e.g., Elliott et al., 1997). In addition, ϵ_{Hf_i} is uniformly low in the HSB, irrespective of their changing B/Nb, Ba/Nb, or Sr and Nd isotopes. This apparent decoupling of Hf isotopes from other tracers is different from what we see in the LSB, and suggests that B, Ba, Sr, and Nd inputs to HSB mantle sources may occur via hydrous fluids, in which Hf is immobile (Pearce et al., 2007; Woodhead et al., 2012). Hydrous fluid inputs should lower Hf/Nd, and greater inputs should result in higher degrees of melting, which are reflected in lower Hf/Ti and La/Yb (Supplementary Fig. S2).

A detailed discussion about the slab B isotopes during HSB generation is now included in the section **Basalt and sediment fluid inputs to the high silica boninite mantle source** section.

This is important as HSB is actually more unique in terms of $\delta^{11}\text{B}$ - B/Ce - B/Nb systematics compared to global arc volcanics than LSB, despite the authors suggesting LSB is more unique (it isn't, not even in their own Fig 3, where LSB partially overlaps with Mariana volcanic front, whereas HSB is uniquely low in $\delta^{11}\text{B}$ for its B/Nb). A strong sediment component is indeed likely and in fact involvement of a Layer 2 basalt component seems hard to prove.

The reviewer's argument that the HSB signature, i.e., high B/Nb, high Ba/Nb, and low $\delta^{11}\text{B}$, is distinct is true, and it is strong evidence for little to no involvement of a serpentinite contribution in their sources during subduction initiation. HSB are also distinctive in that their production requires the melting of harzburgite, as indicated by their major element chemistries, REE systematics, and phenocryst assemblages. LSB compositions, with their higher Ti contents and comparatively flat REE patterns point to the initial presence of clinopyroxene in the source when melting begins, though CPX appears to melt out early, leaving behind a harzburgitic residuum.

The LSB are unique in that they have not been identified as a distinct boninite

sub-type until our discoveries of extensive LSB eruptive sequences during Expedition 352. These discoveries have led to a re-assessment of boninites as a rock type (see Pearce and Reagan, 2019). This said, the LSB do most closely resemble FME-depleted arc lavas like those of the Cascades or Mexico in terms of their B systematics, possibly for similar reasons (i.e., extensive slab dehydration due to high slab temperatures), their oddly large range in $\delta^{11}\text{B}$ notwithstanding.

Structure can be improved.

The authors start the discussion with a section on possible slab components, but this whole section seems redundant as they explore this in more detail in the following sections, and also they assume whole-scale mixing and ignore B isotope fractionation during dehydration, so the discussion is incomplete. They even admit as much in their last two sentences.

The section on HSB is very short compared to the detailed discussion of LSB, but a subsequent section (Rapid changes in slab flux from LSB to HSB reflects a cooling slab) is essentially about generation of HSB again. It would make more sense to integrate those sections. The cooling slab angle isn't very clear at all. I think rearranging and removing repetition would greatly improve the clarity of the manuscript.

We have significantly modified the text to make the sequencing of topics more sensible, and to reduce repetition.

In summary, I think the paper presents a really interesting high-quality dataset which puts some important constraints on the process of subduction initiation, which is worthy of publishing in Nature Comms. However, it needs some significant revisions, in response to my comments above and further detailed comments below, and probably some tweaking of the interpretations as a result.

Cees-Jan De Hoog

University of Edinburgh

Detailed comments (copied from annotated version of manuscript, where they can be found pointing to the relevant part of the manuscript)

- I think Gervais et al 2021 make some quite clear statements about this (amphibolite melt for LSB, hydrous sediment melt for HSB), so it's good to make clear what the current interpretation is, where it is still lacking, and if you disagree with previous findings.

We have revised this explanation: HSB are melts of the LSB residual source fluxed by fluid inputs from a mixture of upper crust and sediments.

We (Li and Ryan) are co-authors of Shervais et al. (2021). The point of calling on hydrous sediment melts for the HSB in that paper is based on HSB and LSB having similarly high Zr/Sm (and Hf/Nd). Our interpretation favoring subduction-related fluid inputs to the HSB source, which is itself a source modified by LSB fluxing and melting, is new (see **Main Reply #5**). The Shervais et al. (2021) conclusions were primarily based on trace elemental modeling, while our model involves combined modeling of the trace elements and isotopes, which we think offers more solid constraints.

- Not sure why in this figure some literature data is plotted as individual samples (MAR srp), others as (rather distracting) arrows (L3), others with an error bar in x but not y (avg sed), and not even consistent per panel. Fields would probably be better, one could still indicate range of d11B falling of the scale. Detailed Sr isotope inset is distracting and unnecessary.

Limitations in the availability of some of the supporting trace element data were behind most of these choices in the diagrams. We've recently found an appropriately complete data set (Hart et al., 1999 GCA) that allows us to plot everything the same way, so we have revised the diagrams. The inset for Fig. 3c was deleted in the revision.

- This is an artificially small variation, based on an integrated average of composite samples, so cannot be used to compare with the small range of HSB (as you do in 1.

124). The actual range is -2.5 to +3.8, still not a big range but more than the samples. Same is true for sediments. Plotting averages here and saying the range is small, and use that as an argument, and plotting individual samples for other sources, and use that as an argument for a large range, seems biased. Moreover, it isn't even a valid argument. Surely large-scale extraction of material from the slab either as melt or fluid and subsequent metasomatism of the mantle source would eradicate the heterogeneity of the slab component.

- But range from -13 to +26. I don't see how this is consistent with the limited variation in HSB (1. 123-125). Also, how was this average derived? Tonga sediments are $+2\pm 4$, Japan marine sediments are also positive (+4 but large range), only continental detritus has negative values (unlikely to be present during intra-oceanic subduction initiation), SSI is negative but highly variable. Your average and stdev in Fig 3 masks a lot of the true variability. Some evaluation of the type of sediment entering the subduction zone is needed.

In our revision, we have chosen to use the data range rather than average values to be clear and consistent. The data ranges have been added in the text and updated in the diagram.

- There's in fact a positive correlation between Sr and B isotopes for LSB, which is at least as 'strong' as the correlation with eHf you mention later (neither being particularly strong)

Please see **Main Reply #2**. The Sr isotopic variation in the LSB is limited, and fully overlaps with the variation seen along the Izu-Bonin arc front. The $\delta^{11}\text{B}$ in the LSB span a far greater range than is seen in almost any arc at its volcanic front – and it's greater than the range seen in the Izu-Bonin arc, even including documentably behind-the front samples (e.g., Ishikawa and Nakamura 1994). Per the reviewer's comments re: our Figure 3, small additions of a high $\delta^{11}\text{B}$ enriched component that does not significantly perturb the $^{87}\text{Sr}/^{86}\text{Sr}$.

- In terms of $\delta^{11}\text{B}$, fluids from altered basalt would do the job perfectly. Why do

you only consider whole-scale mixing? Any discussion of possible sources in relation to B isotopes is incomplete without considering fluids.

Please see **Main Reply #1**. The effects of fluids have been extensively discussed in the revision.

- On the Sr-B isotope diagram HSB to LSB is a continuum, and most of the LSB can be modeled as a mix of L2A and L3 gabbro considering the range in both parameters for both sources. So only some LSB would require those additions. You actually say this in l. 144 so you are contradicting yourself somewhat.

We have changed this text and how discuss the source characteristics. These sentences have been deleted in the revision.

- Data for hole 1415 also in figure, so not just 735B? Also, how about Troodos and Oman data from the same paper?

Our $\delta^{11}\text{B}$ for Layer 3 gabbro is based on data from Smith et al. (1995, Chemical Geology) and Hart et al. (1999, GCA) for gabbros from ODP Hole 735B. ODP Hole 735B is the only source of drilled samples taken directly from intact oceanic crustal Layer 3, particularly as regards boron isotope data. The low B/Nb and high $\delta^{11}\text{B}$ signatures of the ODP Hole 735B gabbros are consistent with data from ophiolites such as Troodos and Oman, where $\delta^{11}\text{B}$ increases while B/Nb decreases with depth (Yamaoka et al., 2012 GCA; 2015 Chemical Geology). However, the tectonic histories of ophiolites are complicated, and as Expedition 352 results helped confirm, they are likely affected by subduction. So ophiolite data are not plotted on our diagrams, though we do make use of these data as appropriate. The Smith et al. (1995) and Hart et al. (1999) gabbroic data are reasonably representative of the Layer 3 oceanic crustal signature. Some high $\delta^{11}\text{B}$ data for Layer 3 gabbros come from IODP Hole U1415 (McCaig et al., 2018) in the Hess Deep. McCaig et al. (2018) thought all their collected samples were likely from the damage zone of a multistrand fault, and that the alteration intensity is thus unlikely to be typical of fast-spreading lower oceanic crust away from faults, which are still analytically inaccessible. In our revised

submission, the McCaig et al. (2018) data are removed from the diagram.

- Sure, the pattern is similar, but that doesn't mean anything unless you have a model that explains how you can transfer the heterogeneity of the slab to the mantle source region and preserve it during melting

Please see **Main Reply #1 and #4**. We have changed the structure of the text and the way we discuss source characteristics.

- And more importantly, its isotopic signature!

Rewording: Boron element and isotopes

- Which then begs the question, what's the point of the previous discussion? Why not incorporate these considerations right away?

We have tried to do so in the revised MS.

- It's a real pity that there is no B isotope data for these. It would go a long way in alleviating any worries about seawater alteration effects on $\delta^{11}\text{B}$ (as one would expect them to be MORB-like), as well as provide a starting point for any mixing model involving B isotopes.

Unfortunately the available FAB materials are significantly altered. These bulk rock samples are not suitable for B isotope analyses even using the leaching strategies of Li et al (2019; Chemical Geology). When analyzing Pb isotopes (Li et al., 2019 EPSL), it was difficult to obtain primary Pb isotope signatures even after heavily leaching the FAB powders.

- Is that a thing? Isn't wet basalt melting curve at much higher T than amphibolite facies?
- Looking at the model, it appears the authors simply mean amphibolite melting, the residue being a granulite, and conditions in fact granulite facies. This may seem pedantic, but it's confusing terminology.

Rewording: melting of the subducting ocean crust under hornblende-bearing granulite facies.

- It's unclear what the red arrows in the figure represent, melting models? Would we expect straight lines? And then why two lines from each colour, are these different source compositions? But then why do you have different sources for low and high-T melts? Please clarify and if these are just lines drawn by eye, please replace with melting/mixing models.

We have replaced these lines with curves based on melting/mixing models.

- Although I can see the argument for this, I do want to point out that the trace element modeling is highly dependent on the mineral assemblage of the source rock, which is assumed to be amphibolite with only plagioclase and amphibole, but this is not entirely consistent with the melting conditions, and ignores the possible presence of garnet and/or mica in the source. This needs some discussion.
- In addition, the model is incomplete. You simply mix a melt with DM. But this gives you the source. Melting needs to be included, and also, DM is a poor choice if considerable melt extraction has occurred prior (as shown by Shervais et al for these rocks). These points may not make much difference but we can't evaluate that.

Rewording: melting of the subducting ocean crust under hornblende-bearing granulite facies.

The calculations of Shervais et al. (2021), suggest there is no residual garnet in the melted slab. Based on observed mineral assemblages in amphibolites of the metamorphic sole of the Oman ophiolite, and on high temperature melting experiments, mica is also not residual. We use *Perple_X* calculations in our revision. According to the calculation result of Shervais et al. (2021), adding 0.1% of slab melt to the FAB residual source will totally overprint the Hf-Nd isotopes of the mantle as the FAB residual source is so low in Hf-Nd elements, then being impossible to see the elemental ratio (e.g., Ba/Nb and Sr/Nd) versus ϵ_{Hf} correlations for the LSB. Thus we think using the DDM (Workman and Hart, 2005) as a starting mantle is reasonable.

- amphibolite melting (see comment earlier, it's not amphibolite facies)

Rewording: hornblende-bearing granulite melting

- where are the details of this model (mineralogy, D values, etc), and why can't these be plotted on the figures instead of as insets?

The information about mineralogy and D values have been added in the text and Figure captions. The diagram has been modified to show mixing between the DDM and the slab melts.

- One would probably expect more of a continuum, the slab doesn't melt at two distinct temperatures, first of all there will be temperature differences within the slab and also, it's a dynamic process as the slab is heating. Also, the correlation between 'less altered' and 'low d11B' is tenuous, L2 basalts are highly altered and have low d11B. In fact, these samples look a lot like L2 basalts in terms of d11B and Ba/Nb. So one could equally argue these are melts from L2 basalts that were less altered, so their Sr isotopes are not as high as other L2 basalts.

Please see **Basalt/gabbro melt inputs to the low silica boninite mantle source.** section. We think the high Ba/Nb and low Ba/Nb signature are resulted from melting of the subducting ocean crust under hornblende-bearing granulite facies. The granulite suffered different degrees of modification by amphibolite fluids from the deep slab.

- I think we're missing a step here, as the boninites are not simply a mixture between slab-derived melt and DM, but their source is. So any melting effects are ignored. This may be true for d11B but could affect Ba/Nb (especially if there's amphibole in the source, which seems likely when melts fertilize depleted mantle). If you want your model to be quantitative enough to draw conclusions about d11B of the source, as you do below, the model needs to be as accurate as possible (i.e., reflect the natural processes you think that are happening)

The boninites reflect melting of a very depleted mantle such that the final residue is

harzburgite (e.g., Pearce and Reagan, 2019). Melting of such a mantle source will not significantly affect B/Nb, Ba/Nb, B/Ba or $\delta^{11}\text{B}$.

- This again assumes that the heterogeneity of the slab is reflected in the erupted melts. This is a huge leap of faith, if you think of all that happens inbetween.

Please see **Main Reply #2**. We have tried to address this issue more effectively.

- This all seems rather speculative and it's unclear how small changes in other parameters would affect this (Ba/Nb, melting T, also earlier comment about ignoring melting fractionation and possible amphibole in the source). Also, a fluid liberated from the crust would have a positive $\delta^{11}\text{B}$ value (e.g. King et al, 2007) as well as high B and completely overprint the DM B isotope value if it resulted in a 8x higher B content than DM. So I don't see how you can shift DM only a few permil this way.

Please see **Main Reply #3**. The mantle source is not DM, and the possible compositional variations cover a range of $\delta^{11}\text{B}$, Ba/Nb, and Sr/Nd.

- The effect of this on B isotopes is actually completely ignored. Rocks don't just heat up and then melt, they undergo dehydration reactions first. This will fractionate B isotopes. These fluids may induce melting elsewhere in the slab depending on the thermal profile. They may mix with melts. They may escape first. We know very little about this process. It may explain some or even much of the variability in B isotopes you are seeing, but there's little to no research done on this.

Please see **Main Reply #1 and #4**. The effects of metamorphism have been more completely discussed in our revised MS. It is worth noting, however, that the thermal structure and dynamics of the subduction initiation setting is markedly different from that of a mature subduction system. Adiabatic upwelling of hot mantle occurs to fill the gap created by the foundering of old Pacific crust and lithosphere. Because of this, initial slab heating is likely to occur from the bottom up as opposed to the top downward, especially along the leading edge of the slab, which is "bathed" in rising hot mantle material. Fluids liberated from the leading edge of the sinking slab

would likely percolate inward through the downgoing plate as opposed to upward, simply because in the earliest stages of subduction there isn't much of an upward option, as the slab is not deeply injected. As such, early dehydration of the slab edge will create a variably modified slab interior reflecting exchanges with fluids liberated from the slab near its sinking leading edge, and it is this interior slab that ultimately melts in the hornblende-bearing granulite facies and fluxes early boninite magmatism.

- Are you sure? Looking at your mixing lines between serpentinite and L2 basalt in Fig 3, it's clear you can increase d_{11B} considerably before increasing $^{87}\text{Sr}/^{86}\text{Sr}$ and with only limited increase in B/Nb. As such, you could explain much of the variation in d_{11B} in LSB by taking a low d_{11B} - low $^{87}\text{Sr}/^{86}\text{Sr}$ LSB sample and add serpentinite fluid. Note that there's actually a correlation between B and Sr isotopes for LSB in support of such a scenario.

Please see Main Reply #1. While this is possible, we think fluids generated in the slab crust by early dehydration reactions are more likely.

- I am not an expert in Hf isotopes. But can this not simply be the DM signature? Or does DM have a high epsilon? In that case I understand your dilemma, you need to explain why HSB has a stronger melt signature than LSB, yet also a fluid signature, which must eradicate some of the melt signature characteristics of LSB (e.g. reduce range of d_{11B}). Please clarify how HSB gets such low e_{Hf} .

The radiogenic isotopes are fully discussed in Li et al. (2019 EPSL). LSB generation: FAB residual mantle + slab melts; HSB: LSB residual mantle + additional fluid. The FAB mantle has higher ϵ_{Hf} than that of the LSB. The decrease of ϵ_{Hf} for the LSB is resulted from crust melting input. The low ϵ_{Hf} signature of the HSB is inherited from the crust melting input before HSB generation.

- So you are proposing a mixture of Layer 3A melts and Layer 2A fluids? The fluids must have had high Sr/Nd (as Sr/Nd is much lower than expected based on the e_{Hf} trend), no Ba (as Ba/Nb falls on e_{Hf} trend) and enough B to overprint the LSB

melt B isotope signature. Thus the fluid (not the rock it came from) had d_{11B} of 0-2 permil. Does this still fit the model? Note that fluids derived from a rock will have a different signature than the rock itself. This doesn't appear to be taken into account. The model could be correct, but it's not properly evaluated and d_{11B} not properly discussed.

In our revised Figure 4, we find that the lowest Sr/Nd in the HSB is near the median Sr/Nd value of the LSB, while the corresponding La/Sm ratio is slightly higher than the highest value of the LSB (Figs. 4c-d). This indicates that the mantle source may have experienced a low Sr/Nd slab melt input before the generation of the HSB. This event may not be clearly reflected in the chemistries of the Expedition 352 LSB. If such an event occurred, the melt may have come from subducted basalt.

I'd probably say fore arc mantle wedge, as you do in the next sentence. It may indeed end up in the subduction channel, but it doesn't have to (it can be brought beneath the arc by corner flow). Any serpentinites in the subduction channel will probably have interacted with sediments and have low d_{11B} , as demonstrated by Martin et al. 2020.

Rewording: Forearc mantle wedge

All the IBM forearc serpentinites (Benton et al., 2001 and our IODP 366 unpublished data) have high δ^{11B} , from +10‰ to +22‰. Serpentine formation processes in the Mariana forearc likely drive these signatures to such high values, because when the slab releases B-rich fluids these fluids will likely be heavier than the residual solids, and the serpentine formation process in the subduction channel raises the system pH to values >10, effectively preserving these initial high values. Guatemala and Nicaragua mélange serpentinites record much more variable δ^{11B} , from -14 to +10‰ (Martin et al., 2020 *Chemical Geology* 545 119637). These differences may relate to differences in the slab material (more or less continental sediments) but certainly relate to differences in their generation mechanisms and subsequent reactions related to their emplacement in the crust. A detailed discussion of this is beyond the scope of this study.

- By this point I got curious how different it was from the De Hoog and Savov database, and yes, they have rather low B, but in terms of B/Nb they aren't unique at all, in the Izu arc we find similar values (and note they overlap Mariana VF in your own plot). B/Ce is normal as well. So low B is simply due to high degrees of melting. They are rather unusual in have very low Ba/Nb though. Also, La/Sm is low, certainly for the low d11B ones. In fact, it appears to trend to MORB, perhaps again an indicator of high melt fractions compared to typical arc basalts.

The ones really standing out from the global arc database are HSB, which have high B/Nb as well as B/Ce for their (low) d11B.

So perhaps you want to modify your statements here. It appears for LSB the slab-derived component is different, but the actual melt composition not that much from at least some regular arc basalts (in terms of B/Nb and B isotopes at least), whereas HSB is quite different despite the involvement of similar components. Note that the low d11B of HSB was not properly explained in the previous section.

Please see Main Reply #1. The systematics of less fluid-mobile trace elements like the REE argue strongly against significant differences in melt extent between the LSB and HSB. While the shapes of their REE patterns are slightly different, overall their REE abundance levels are similar, indicating similar values of F during melting, given that they have similar mantle sources.

Comparisons of Izu-Bonin arc lavas and boninites in terms of melt extents are problematic because these lavas have very different source mantles: arc basalts form as eutectic melts of a fertile, lherzolitic mantle, while boninites are by definition melts formed in equilibrium with highly chemically depleted harzburgites. While similarities in B/Nb or B/Ce between Izu-Bonin arc lavas and boninites are interesting in terms of source fingerprints (though co-author Ryan reported data on Izu arc lavas with much higher B/Ce than any boninite in this study in his 1993 paper on B abundance systematics), one cannot reliably compare these two very differently sourced magma types with regard to melt extents merely in terms of their trace element ratios. As such, the boron concentration differences and differences in B/Nb

or B/Ce among the boninites must be related to difference in source, specifically differences in slab contributors, as the mantle contribution for B is effectively nonexistent.

Comparisons of Izu-Bonin and Marianas arc lavas in terms of boron and $\delta^{11}\text{B}$, and especially comparisons between Marianas lavas and Izu forearc lavas need to be considered in the context of the very large (per Savov et al 2005; 2007 upwards of 70% of the slab inventory) outfluxes of high $\delta^{11}\text{B}$ boron that occur all along the shallow Mariana forearc in the form of a mountain range of actively erupting serpentinite mud volcanoes. The overall lower B/Nb of Marianas arc lavas, which have similar Ba/Nb to Izu-Bonin lavas, is explicable in terms of this globally unique phenomenon. We've removed the Mariana arc data from Figure 3 to avoid confusion on this point.

This section is essentially part of generation of HSB. It would make sense to include it in the earlier section on that topic.

We have done so.

- than LSB?

Rewording: than LSB

- As far as I understood, HSB also has a strong slab melt component (evidenced by low ϵHf) with an additional fluid signature. Does this model reflect that? As this makes it sound like it is fluid only?

HSB: melting LSB residual mantle via fluxing of basalt+sediment derived fluids.

The FAB mantle has higher ϵHf than that of the LSB. The decrease of ϵHf for the LSB is resulted from crust melting input. The low ϵHf signature of the HSB is inherited from the crust melting input before HSB generation.

- Important point, which wasn't taken into account earlier during generation of LSB. More importantly, it will also fractionate B isotopes.

Please see **Main Reply #3**. The effects on boron isotope fractionation during slab

metamorphism and slab dehydration/melting have been discussed in the revision.

- Note that B/Nb is actually quite high for the low d_{11B} values (compared to global arcs). The B isotope signature requires more explanation than B/Nb.

We more extensively discuss the δ^{11B} of the HSB source in the revised MS. Dehydrated basalts+sediments are the likely source, and the relatively tight distribution of values is reflective of a uniformly sampled slab, such as would be the case with slab fluid releases.

- I thought it was mostly Layer 2A basalts? Or do you refer to the 5% sediment in the model in the previous section? I am getting very confused. I agree that sediments are likely but your arguments or wording are inconsistent. Also, I'd add 'fluids from' We added 'fluids from' in the revision.

The HSB are derived from the LSB residual mantle source fluxed by additional fluid inputs from the upper oceanic crust and sediments. This explanation is now consistent throughout the text.

- I am not convinced the variation in d_{11B} of LSB isn't the result of small amounts of serpentinite fluid to another component with a value of about 0 to +2 (consistent with correlation with Sr isotopes and Ba/Nb and B/Nb systematics). It's certainly a valid alternative to your explanation based on the heterogeneity in d_{11B} of L3 gabbro. It doesn't make the rest of your story less valid, as you say earlier, abyssal serpentinites would dehydrate at lower T.

Please see **Main Reply #1**. We agree that the pattern of the LSB data points to small and variable additions of some kind of δ^{11B} fluid. While serpentinite may be a source and we say as much, geodynamically it is probably more likely coming from fluids related to within-crust metasomatism in the foundering slab.

- Why are the two plates a different thickness? Aren't they essentially the same plate, which breaks and then one part sinks?

Our model picture is modified from Li et al. (2019), which is itself modified from past schematic models for the region. The two plates are separated by a transform fault. The proto-Philippine Sea Plate is younger and therefore presumed to have a thinner lithosphere than the Pacific Plate.

- What is the significance of these?

That is why sediments and the altered basalt contribution are not observed in the LSB. We have added clarifying information in the text.

- I thought you didn't need fluids. You modeled melts alone specifically.

Suggestions of fluid deleted from the revision.

- The overlying L2 basalts are more altered and closer to the hot upwelling mantle. Why wouldn't these melt first? Or are you suggesting these were eroded in the proto-accretionary wedge? Some explanation is needed.

That we don't see clear evidence for Layer 2 basalt involvement in the LSB argues that they may early on have been accreted to the proto-accretionary wedge. We have clarified our explanations of this in the text.

- Note again they have very anomalous low d_{11B} for their high B/Nb. They are not similar to other arcs.

We have rewritten this to make clear that the similarity is to slab fluid fluxed melting in the HSB, which is more like what is seen in mature arcs, than the slab-melt fluxing magmatism of the LSB.

- From when (do we observe typical arc basalts)? The timing of this is of interest.

After 44 Ma. We've added this information

- Do you mean melting reactions?
- I assume that's from Wolf and Wyllie?

We are using `Perple_X` for calculation in the revision.

REVIEWERS' COMMENTS

Reviewer #4 (Remarks to the Author):

This is a revised manuscript that I reviewed previously. The authors have done an excellent job addressing my concerns and I find the manuscript much improved. They present a coherent model documenting temporal changes during subduction initiation and how boron isotopes can be used to identify various sources as well as mode of transport to the developing mantle wedge. This is the first time subduction initiation has been documented in such detail and it is a significant contribution to our understanding of the process. I have no further comments and I look forward to seeing this manuscript published.

Cees-Jan de Hoog
University of Edinburgh

DETAILED RESPONSES TO THE REVIEWERS

Reviewer #4: (Remarks to the Author):

This is a revised manuscript that I reviewed previously. The authors have done an excellent job addressing my concerns and I find the manuscript much improved. The present a coherent model documenting temporal changes during subduction initiation and how boron isotopes can be used to identify various sources as well as mode of transport to the developing mantle wedge. This is the first time subduction initiation has been documented in such detail and it is a significant contribution to our understanding of the process. I have no further comments and I look forward to seeing this manuscript published.

Cees-Jan de Hoog

University of Edinburgh

We sincerely thank the reviewer for his constructive comments and his concrete criticisms that helped to improve the manuscript.